# Isoform-resolved mRNA profiling of ribosome load defines interplay of HIF and mTOR dysregulation in kidney cancer

Yoichiro Sugimoto[1] and Peter J. Ratcliffe [1,2] ✉

Hypoxia inducible factor (HIF) and mammalian target of rapamycin (mTOR) pathways orchestrate responses to oxygen and nutrient availability. These pathways are frequently dysregulated in cancer, but their interplay is poorly understood, in part because of difficulties in simultaneous measurement of global and mRNA-specific translation. Here, we describe a workflow for measurement of ribosome load of mRNAs resolved by their transcription start sites (TSSs). Its application to kidney cancer cells reveals extensive translational reprogramming by mTOR, strongly affecting many metabolic enzymes and pathways. By contrast, global effects of HIF on translation are limited, and we do not observe reported translational activation by HIF2A. In contrast, HIF-dependent alterations in TSS usage are associated with robust changes in translational efficiency in a subset of genes. Analyses of the interplay of HIF and mTOR reveal that specific classes of HIF1A and HIF2A transcriptional target gene manifest different sensitivity to mTOR, in a manner that supports combined use of HIF2A and mTOR inhibitors in treatment of kidney cancer.

Precise regulation of transcription and translation is required to define patterns of protein synthesis in healthy cells. Nevertheless, attempts to understand disease have often focused on a single pathway of transcriptional or translational control, despite their simultaneous dysregulation. For instance, two major pathways that link the cellular environment to gene expression, the HIF and mTOR pathways, are both dysregulated in many cancers. The most common kidney cancer, clear cell renal carcinoma, manifests upregulation of HIF owing to defective function of its E3 ubiquitin ligase, the von Hippel–Lindau tumor suppressor (VHL), and hyperactivation of mTOR[1,2]. In addition, microenvironmental tumor hypoxia increases the activity of HIF[3] and also acts on translation via mTOR and other pathways[4–8].

HIF mediates responses to hypoxia through a well-defined role in transcription, but recent studies also report a role for it in translation. In the presence of oxygen, two isoforms of HIFα (HIF1A and HIF2A) are ubiquitinated by VHL and degraded. This prevents the formation of transcriptionally active heterodimers with HIF1B[3]. In addition, HIF2A is reported to regulate translation via non-canonical cap-dependent translation, mediated by eukaryotic translation initiation factor 4E family member 2 (EIF4E2)[9]. It was further reported that a large subset of genes, including HIF transcription targets, are translationally upregulated by the HIF2A–EIF4E2 axis, resulting in induction of protein in hypoxic cells, even when HIF-dependent transcription was ablated by HIF1B knockdown[10]. Evaluation of this action of HIF is important given efforts to treat *VHL*-defective kidney cancer through HIF2A–HIF1B dimerization inhibitors[11,12], whose action to prevent transcription might be circumvented by effects of HIF2A on translation.

mTOR forms two different complexes, mTORC1 and mTORC2. mTORC1 controls translation via phosphorylation of EIF4E binding protein (EIF4EBP)[13,14]. When mTORC1 is inhibited, such as by nutrient deprivation, unphosphorylated EIF4EBP binds to EIF4E and this blocks the EIF4E-EIF4G1 interaction, which is necessary to form a canonical translation initiation complex[14]. In contrast, mTORC2 controls cell proliferation and migration by phosphorylating AKT serine/threonine kinase and other targets[13].

Comprehensive characterization of the regulation of gene expression by the HIF–VHL and mTOR pathways is crucial to understanding the biology of *VHL*-defective kidney cancer, particularly as agents targeting both these pathways are being deployed therapeutically[15,16]. Although mTOR has been reported to be inhibited by HIF under hypoxia[8,17], its interactions with the HIF system are poorly understood.

In part, this reflects the lack of efficient methods to measure translational efficiency and to interface such methods with transcriptional data. Most existing methods capable of pan-genomic analysis rely on one of two principles; assessment based on the position of ribosomes on mRNAs by ribosomal foot-printing (ribosome profiling), or assessment of the number of ribosomes on mRNAs by polysome profiling[18] (see also Supplementary Information). Such methods have provided valuable information on translational control. This has enabled the definition of mRNA features that regulate translational efficiency[19,20] and has facilitated analyses of interventions on pathways that regulate translation[14,21]. However, scaling these methods to permit multiple comparisons remains a challenge. Moreover, reliance on internal normalization, as used in the majority of studies, allows changes in global translation to confound the measurements of transcript-specific translational efficiency[22]. Furthermore, ribosomal profiling cannot readily distinguish the translational efficiency of overlapping transcripts such as those generated by alternate TSSs. Resolution of specific transcripts by their TSS provides important insights into the mode of translational regulation[19,23,24] and is particularly important when assessing translation in the setting of a large transcriptional change, as occurs in cancer[25,26].

Here we describe a new method, high-resolution polysome profiling followed by sequencing of the 5′ ends of mRNAs (HP5),

¹The Francis Crick Institute, London, UK. ²Ludwig Institute for Cancer Research, Nuffield Department of Clinical Medicine, University of Oxford, Oxford, UK. ✉e-mail: peter.ratcliffe@crick.ac.uk

that addresses these challenges, and demonstrate its use in defining the interplay between transcriptional and translational regulation by the HIF–VHL and mTOR signaling pathways in *VHL*-defective kidney cancer cells.

## Results

**Establishment of HP5 workflow.** HP5 encompasses two key features. First, through the use of external RNA standards, it robustly measures ribosome load of mRNAs. Second, by the exclusion of mRNA or cDNA purification steps before the first PCR amplification and multiplexing of samples at an early stage of the protocol, the method enables the processing of a large number of samples. (Fig. 1a and Extended Data Fig. 1).

We first evaluated the basic performance of HP5 using RCC4 VHL cells, in which constitutive upregulation of HIF in *VHL*-defective RCC4 cells is restored to normal by stable transfection of *VHL* (Extended Data Fig. 2). We obtained an average of 3.3 million reads per fraction, with ~80% of reads mapping to mRNA (Supplementary Data 1). Importantly, HP5 successfully generated each library from 100-fold less total RNA than a similar method (~30 ng compared with 3 μg)[19]. HP5 was highly reproducible: principal component analysis of mRNA abundance data demonstrated tight clustering of each polysome fraction, across three clones of RCC4 VHL cells (Fig. 1b). Furthermore, the 5′ terminus of HP5 reads precisely matched annotated TSSs in RefSeq or GENCODE at nucleotide resolution, confirming the accuracy of 5′ terminal mapping (Fig. 1c).

To further test the performance of HP5, we compared the polysome distribution of a set of TSS-defined mRNA isoforms analyzed by both HP5 and RT–qPCR. Very similar results were obtained, verifying that HP5 can accurately resolve the translation of these isoforms (Extended Data Fig. 3a). We then examined the overall relationships between translational efficiency and selected mRNA features, including those with known associations with translational control. Translational efficiency was calculated as the mean ribosome load for each of 12,459 mRNA isoforms resolved by their TSS from 7,815 genes. Using a generalized additive model, we found that the four most predictive features together explained around 36% of variance in mean ribosome load between mRNAs (Extended Data Fig. 3b). Notably, coding sequence (CDS) length showed the clearest association with mean ribosome load: values were greatest for mRNAs with a CDS length of around 1,000 nucleotides (nt) and declined progressively as the CDS became longer (Fig. 1d), probably owing to a lower likelihood of re-initiation of translation by mRNA circularization[27]. In agreement with previous studies[28–30], analysis of HP5 data identified the negative effect on translation of upstream open reading frames (uORFs) and RNA structures near the cap, as well as the positive effect of the Kozak sequence (Fig. 1e and Extended Data Fig. 3c,d). Importantly, the association of mean ribosome load with mRNA features that affect translation extended to comparisons between mRNA isoforms arising from alternative TSS usage (Fig. 1f). Overall, HP5 reproduced and extended known associations between mRNA features and translation, verifying its performance in the measurement of translational efficiency at transcript resolution (see Supplementary Information for further validation of the method).

**mTOR-dependent translational regulation greater than reported.** We next applied HP5 to the analysis of mTOR pathways, which are frequently dysregulated along with hypoxia signaling pathways in *VHL*-defective kidney cancer. To analyze translational changes that arise directly from mTOR inhibition, RCC4 VHL cells were treated for a short period (2 hours) with Torin 1, an ATP-competitive inhibitor of mTORC1 and mTORC2 (ref. [31]). mTOR inhibition globally suppressed translation, as shown by a marked reduction in polysome abundance (Fig. 2a). Measurements of changes in

translational efficiency were initially analyzed at the level of the gene. This provided the first direct display of both a general reduction in translation by mTOR inhibition and of its heterogeneous effects on individual genes across the genome (Fig. 2b). To assess the performance of HP5 against other methods, we next compared the HP5 data on translational responses to mTOR inhibition with data in four previous studies that reported mTOR hypersensitive genes[14,21,32,33]. Although the mTOR hypersensitive genes identified by these studies did not always strongly overlap, HP5 revealed the translational downregulation of mTOR targets identified in each of the four previous studies (Extended Data Fig. 4a,b). By contrast, at least within these studies, ribosome profiling appeared less powerful in identifying the mTOR hypersensitive genes defined by polysome profiling (Extended Data Fig. 4b). Note that one caveat to this is that ribosomal load is not a direct measure of translational efficiency, as translation can be regulated not only by initiation but also elongation[22].

mTOR has been reported to regulate a wide range of processes by different mechanisms[13], while the identification of the direct translational targets has been more limited, for instance, involving proteins that function in translation itself. Our data confirmed many of these known mTOR translational targets, as well as the previously described resistance of many transcription factors[14]. Importantly, our data also demonstrated directly that the translation of genes encoding proteins with many other functions, such as in different metabolic pathways, and in proteasomal degradation is hypersensitive to mTOR inhibition (Fig. 2c).

The accurate resolution of the TSS provided by HP5 also offered an opportunity to improve the understanding of transcript-specific mRNA features associated with mTOR hypersensitivity or resistance. mTOR has been shown to regulate mRNAs with a 5′ terminal oligopyrimidine (TOP) motif in a tract-length-dependent manner[34]. Our analysis confirmed this (Fig. 2d). By contrast, although it has been reported that TOP motifs starting between +2 and +4 nt downstream of the cap mediate mTOR control[14], the high-resolution analysis permitted by HP5 revealed that any such association with Torin 1 sensitivity was much weaker if the TOP motifs did not start immediately after the cap (Fig. 2d).

Although these data confirmed the importance of the TOP motif for translational regulation by mTOR, the proportion of mRNAs containing a TOP motif immediately after the cap was low (only 6% of mRNAs had a TOP motif of more than 2 nucleotides, Extended Data Fig. 5a) compared with the global extent of translational alteration by mTOR inhibition, suggesting that additional mechanisms contribute to the mTOR sensitivity[24]. To explore this, we examined the interaction of Torin 1-induced changes in translation with uORF frequency and CDS length, the two most important mRNA features affecting translational efficiency under mTOR-active conditions (Extended Data Fig. 3b). We observed that uORF number retained only a very weak association with mean ribosome load under mTOR inhibition (Fig. 2e). With respect to CDS length, the increased translational efficiency of mRNAs with a CDS of close to 1 kb was not observed upon mTOR inhibition (Fig. 2f and Extended Data Fig. 5b). Rather, there was a progressive increase in mean ribosome load with increasing CDS length, as might be expected if CDS length was not affecting translational initiation. These differences suggest that mTOR pathways also impinge on the translational effects of these mRNA features. For instance, EIF4EBP activation by mTOR inhibition might prevent mRNAs from forming a loop through blocking EIF4E and EIF4G1 interactions. Note that an association of mRNA length with mTOR sensitivity was also observed but was slightly weaker (Extended Data Fig. 5c). Interactions between the mTOR sensitivity of mRNAs and features such as the TOP motif or number of uORFs were also observed when comparing mRNA isoforms of the same gene (Extended Data Fig. 5d). Overall, the analyses revealed that the extent of translation regulation by

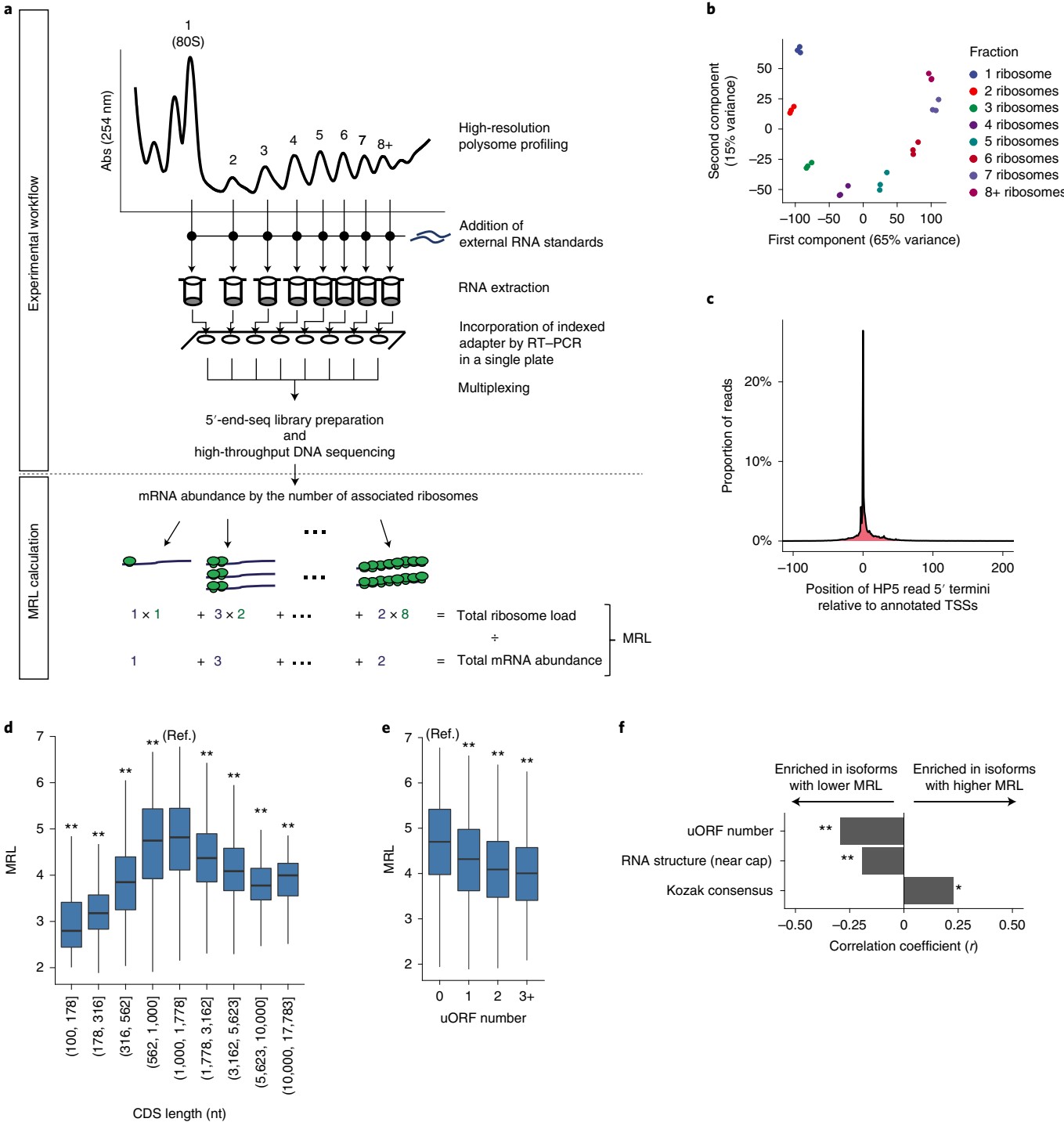

**Fig. 1 | HP5 reliably measured translational efficiency of mRNAs resolved by their TSS. a**, Schematic overview of HP5. Top panel shows the experimental workflow; bottom panel shows an example of the mean ribosome load (MRL) calculation. Abs (254 nm), absorbance at 254 nm. **b**, Principal component analysis of HP5 data by polysome fraction, for three independent RCC4 VHL clones. **c**, Position of identified 5′ termini relative to the closest annotated TSSs; data are the proportion of reads with the indicated 5′ terminus, relative to the total reads mapping to that gene locus. **d**, MRL as a function of CDS length. MRL for mRNAs with the indicated CDS length was compared to that of 1,000 to 1,778 nt (reference, indicated as Ref.), using the two-sided Mann–Whitney $U$ test. **e**, MRL as a function of uORF number. MRL for mRNAs with the indicated uORF number was compared with that without an uORF (reference, indicated as Ref.), using the two-sided Mann–Whitney $U$ test. **f**, mRNA features associated with higher or lower MRL in the two most differentially translated mRNA isoforms (FDR < 0.1) derived from the same gene, but differing by their TSS. Comparisons were performed for those pairs of isoforms differing in the relevant features using the two-sided Wilcoxon signed-rank test; the effect size of the association with each mRNA feature was measured using matched-pairs rank biserial correlation coefficients. RNA structure (near cap), is the stability of predicted RNA secondary structures (first 75 nt of mRNA); Kozak consensus, match score to the consensus sequence. Box plots show the median (horizontal lines), first to third quartile range (boxes), and 1.5× interquartile range from the box boundaries (whiskers). *$P < 0.05$, **$P < 0.005$. $P$ values were adjusted for multiple comparisons using Holm's method. Details of the sample sizes and exact $P$ values for **d**–**f** are summarized in the Supplementary Information.

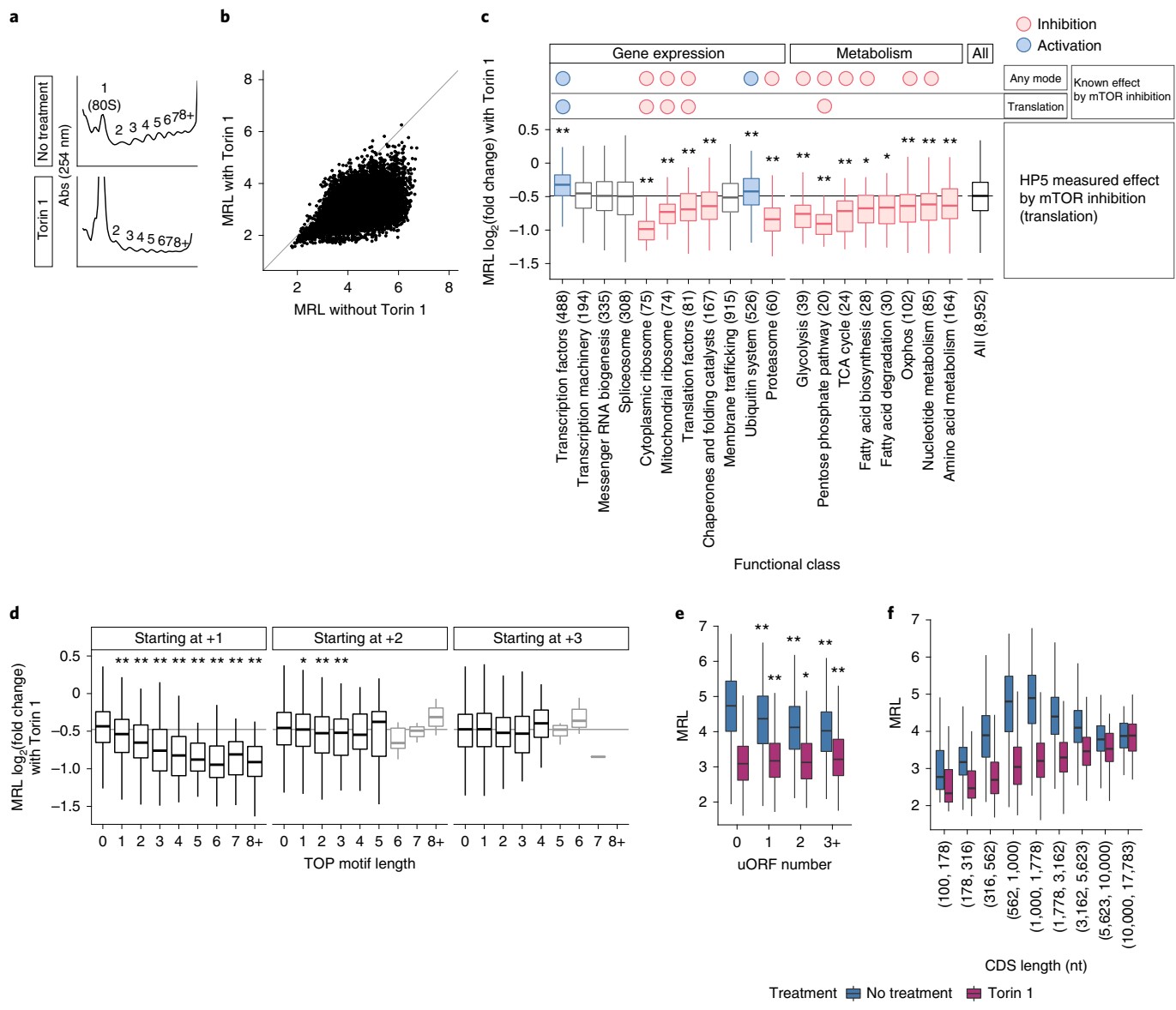

**Fig. 2 | Comprehensive analysis of mTOR-dependent translation regulation by HP5. a**, Polysome profiles of RCC4 VHL cells with and without Torin 1. Abs (254 nm), absorbance at 254 nm. **b**, Comparison of the MRL of genes with and without Torin 1 (data presented are the mean of 2 and 3 independent RCC4 VHL clones). **c**, Box plots showing changes in translational efficiency of genes (expressed as log₂(fold change) in MRL) with Torin 1, among different functional classes. Responses within a functional class were compared against responses for all other genes using the two-sided Mann–Whitney $U$ test; classes that are hypersensitive and resistant to mTOR inhibition are colored red and blue, respectively ($P < 0.05$); numbers of genes in each class are indicated in parentheses. Known mTOR regulation by any mechanism or by translation is indicated above the box plots. **d**, Changes in translational efficiency with Torin 1 as a function of TOP motif (pyrimidine tract) length and starting position with respect to the mRNA cap (individual panels). MRL for mRNAs with the indicated TOP motif length was compared to that without a TOP motif using the two-sided Mann–Whitney $U$ test; boxes representing fewer than ten mRNAs are faded. **e,f**, MRL as a function of uORF number (**e**) or CDS length (**f**) in the presence (purple) or absence (blue) of Torin 1. For **e**, MRL for mRNAs with the indicated uORFs number was compared with that of those without a uORF using the two-sided Mann–Whitney $U$ test. Box plots show the median (horizontal lines), first to third quartile range (boxes), and 1.5× interquartile range from the box boundaries (whiskers). *$P < 0.05$, **$P < 0.005$. $P$ values were adjusted for multiple comparisons using Holm's method. Details of the sample sizes and exact $P$ values for **c–f** are summarized in the Supplementary Information.

mTOR is greater than previously reported and refined the understanding of mRNA features that influence mTOR sensitivity.

**Limited role of HIF2A in regulating translation.** We next sought to examine translational regulation by HIF–VHL pathway by applying HP5 to *VHL*-defective RCC4 and 786-O cells re-expressing either wild-type VHL (RCC4 VHL and 786-O VHL) or empty vector alone. The two cell lines were chosen because RCC4 expresses

both HIF1A and HIF2A, whereas 786-O expresses only HIF2A (Extended Data Fig. 2), enabling us to distinguish roles of HIF1A and HIF2A. Furthermore, previous studies reporting the role of the HIF2A–EIF4E2 pathway were performed in part using 786-O cells[9,10]. Figure 3a shows the changes in translational efficiency associated with loss of *VHL* for RCC4 cells, or 786-O cells compared with the action of Torin 1 on RCC4 VHL cells. In both RCC4 and 786-O cells, *VHL*-defective status was associated with a small global

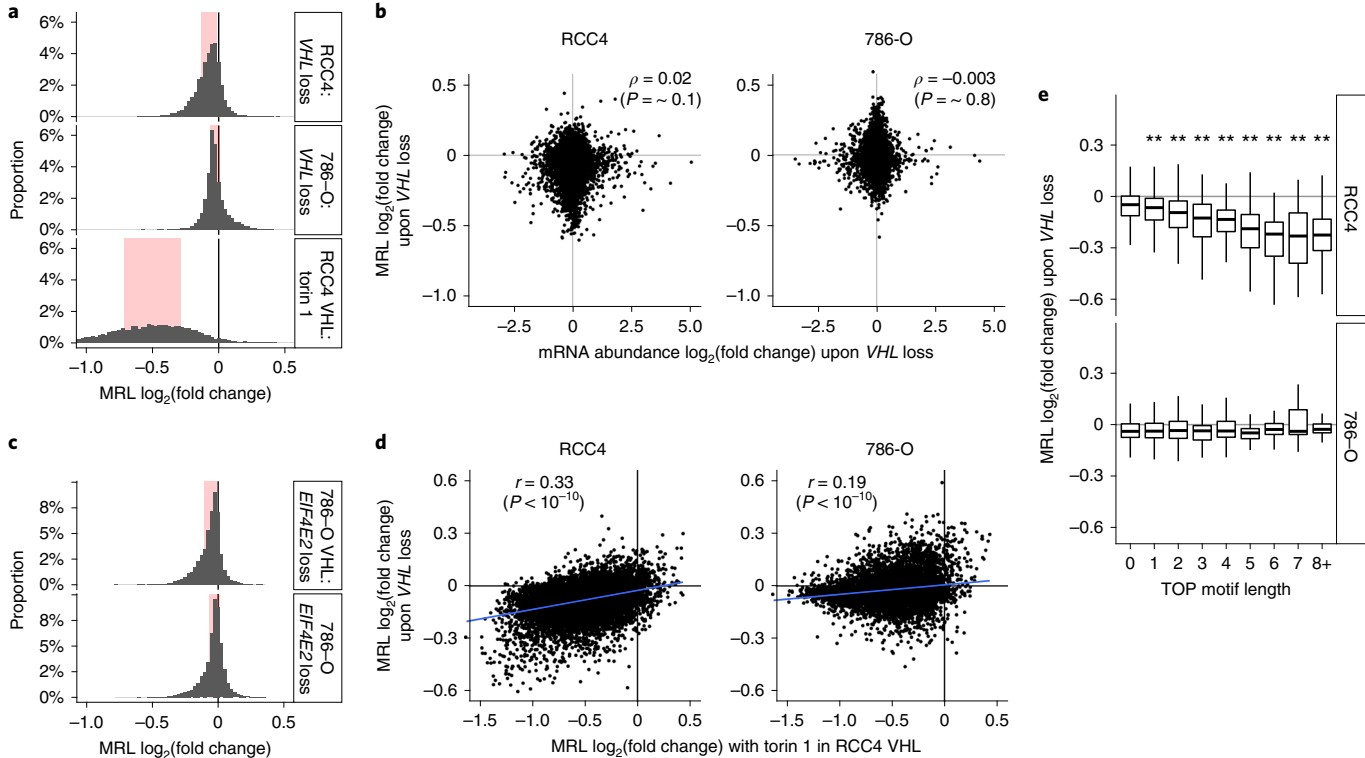

**Fig. 3 | Global view of HIF-dependent translational regulation. a**, Frequency histograms comparing changes in translational efficiency, expressed as $\log_2$(fold change) in MRL for each gene, for three interventions; *VHL* loss in RCC4 cells, *VHL* loss in 786-O cells, and Torin 1 treatment in RCC4 VHL cells. Interquartile range is highlighted in red; the width of histogram bin was set to 0.05. **b**, Scatter plots comparing changes in mRNA abundance of genes induced by *VHL* loss with the changes in translational efficiency in the respective cell type. Spearman's rank-order correlation coefficient was used to assess the association ($n = 9{,}318$ and 7,844 for RCC4 and 786-O respectively). **c**, Frequency histograms as in **a**, showing effects of *EIF4E2* inactivation in 786-O VHL cells and 786-O cells. **d**, Scatter plots comparing changes in translational efficiency of genes upon *VHL* loss in RCC4 or 786-O cells with those induced by Torin 1 treatment in RCC4 VHL cells. The blue line indicates the linear model fit by ordinary least squares. Pearson's product moment correlation coefficient was used to assess the association ($n = 8{,}829$ and 7,512 for RCC4 and 786-O, respectively). **e**, Box plots showing changes in translational efficiency of mRNAs upon *VHL* loss as a function of TOP motif length (*x* axis). Only TOP motifs starting immediately after cap were analyzed. MRL for mRNAs with indicated TOP motif length was compared to that without a TOP motif using the two-sided Mann–Whitney *U* test. Box plots show the median (horizontal lines), first to third quartile range (boxes), and 1.5× interquartile range from the box boundaries (whiskers). *$P < 0.05$, **$P < 0.005$. *P* values were adjusted for multiple comparisons using Holm's method. Details of the sample sizes and exact *P* values are summarized in Supplementary Information.

downregulation of translation, with more genes showing reduced translational efficiency in *VHL*-defective RCC4 cells.

Particularly striking, in view of the reported role of HIF2A in translational upregulation[9,10], was the absence of clear upregulation in translational efficiency in *VHL*-defective RCC4 and 786-O cells, either generally or for those genes reported to be translationally upregulated by HIF2A[9,10] (Fig. 3a,b and Extended Data Fig. 6a), although we confirmed strong induction of HIF2A in both of these cell lines (Extended Data Fig. 2). It is possible that HIF2A upregulates the translation of only a small number of mRNAs, for instance a subset of HIF-induced mRNAs. We therefore compared changes in mRNA abundance induced by *VHL* with changes in translational efficiency. However, we saw no correlation between regulation of transcript abundance and translation, as might have been anticipated if a set of HIF transcriptional targets were also regulated by translation (Spearman's $\rho = 0.02$ and $-0.003$, $P = {\sim}0.1$ and ~0.8 for changes in translational efficiency against changes in mRNA abundance in RCC4 and 786-O cells, respectively; Fig. 3b and Extended Data Fig. 6b).

Because HIF2A's ability to promote translation has been proposed to be mediated by EIF4E2 (ref. [9]), we engineered *EIF4E2*-defective 786-O and 786-O VHL cells by CRISPR–Cas9-mediated inactivation and examined the effects on translational efficiency. In both

786-O and 786-O VHL cells, *EIF4E2* inactivation weakly but globally downregulated the translational efficiency of genes (Fig. 3c). If co-operation of EIF4E2 and HIF2A had a major role in translation, it would be predicted that *EIF4E2* inactivation would have a larger effect in the absence of VHL. However, we observed no evidence of this, for either global translation or reported HIF2A–EIF4E2-target genes[9,10] (Fig. 3c, compare upper and lower panels, and Extended Data Fig. 6c). Finally, to exclude the possibility that HP5 analysis did not capture the effect of HIF2A–EIF4E2-dependent translational regulation, we used immunoblotting to examine changes in the abundance of proteins encoded by reported target genes of HIF2A–EIF4E2 (refs. [9,10]), as a function of *VHL* or *EIF4E2* status in 786-O cells. This further confirmed that the effect of the HIF2A–EIF4E2 pathway was considerably weaker than or undetectable compared with that of HIF2A–VHL-dependent transcriptional regulation (Extended Data Fig. 6d). Taken together, the data revealed little or no role for the HIF2A–EIF4E2 axis in regulation of translation under the analyzed conditions.

Although we did not observe systematic upregulation of translational efficiency, either of HIF transcriptional targets or other genes in *VHL*-defective cells, we did observe downregulation of translational efficiency, particularly in RCC4 cells. To examine whether this might reflect interaction of HIF and mTOR pathways, we first

compared the gene-specific effects on translation that are associated with *VHL*-defective status in RCC4 cells with those observed by inhibition of mTOR in RCC4 VHL cells. This revealed a moderate, but highly significant, correlation between responses to the two interventions in RCC4 cells (Pearson's $r = 0.33$, $P < 1 \times 10^{-10}$, Fig. 3d left panel). Furthermore, mRNAs with a longer TOP motif were more strongly repressed by *VHL* loss in RCC4 cells (Fig. 3e upper panel). Earlier work has suggested that induction of HIFα, particularly the HIF1A isoform, can suppress mTOR pathways[8,35]. Consistent with this, we observed that *VHL* loss in RCC4 cells was associated with a significant upregulation of mRNAs that encode negative regulators of mTOR (BNIP3 and DDIT4) or its target, the translational repressor EIF4EBP1 (Extended Data Fig. 6e). In contrast, in 786-O cells, which do not express HIF1A, we observed less downregulation of translation by *VHL* loss, less association of any gene-specific effects with mTOR targets (defined either by responsiveness to Torin 1, or the length of the TOP sequence) and weaker regulation by VHL of mRNAs that repress mTOR pathways (Fig. 3d right panel, Fig. 3e lower panel, and Extended Data Fig. 6e). Although VHL may have other effects on gene expression beyond regulation of HIF, the findings suggest that modest downregulation of translation occurs in RCC4 cells, most likely as a consequence of HIF1A-dependent actions on mTOR pathways.

**HIF promotes alternate TSS usage to regulate translation.** Although transcription may regulate translation by promoting alternative TSS usage and altering the regulatory features of the mRNA, the effects of HIF on this have not been studied systematically. To address this, we first compared 5′ end sequencing (5′ end-seq) reads from total (that is, unfractionated) mRNAs in RCC4 VHL versus RCC4 and identified 149 genes with a VHL-dependent change in TSS usage (false-discovery rate (FDR) < 0.1). For these genes, we defined a VHL-dependent alternative TSS (which showed the largest change in mRNA abundance with *VHL* loss). Discordant regulation of the alternative and other TSSs (that is, up versus down) was rare (9/149): following *VHL* loss, the alternative TSS was induced in 85 genes and repressed in 64 genes (Supplementary Data 2). To test the generality of these findings and to consider the mechanism, we performed similar analyses of alternative TSS usage among these 149 genes in sets of related conditions and compared the results (Extended Data Fig. 7). A strong correlation (Pearson's $r = 0.60$, $P < 1 \times 10^{-10}$) was observed with alternative TSS usage in 786-O VHL versus 786-O cells. In contrast, there was no correlation with the alternative TSS usage in 786-O VHL versus 786-O cells in which HIF transcription had been ablated by CRISPR–Cas9-mediated inactivation of *HIF1B* (Pearson's $r = -0.01$, $P = \sim0.9$) indicating that the effects were dependent on HIF. In keeping with this, a strong correlation was observed between changes mediated by loss of *VHL* in RCC4 and those induced by hypoxia in RCC4 VHL cells (Pearson's $r = 0.85$, $P < 1 \times 10^{-10}$).

We next sought to determine the effects of HIF-dependent altered TSS usage on mRNA translation by comparing the different isoforms of the same genes. Among the 129 genes whose CDS could be predicted for different isoforms, 71 (55%) have differences in predicted CDS (Supplementary Data 2). Among 117 genes whose different mRNA isoforms were expressed at sufficient levels for calculation of mean ribosome load, 75 (64%) have differences in translational efficiency (FDR < 0.1, Extended Data Fig. 8 and Supplementary Data 2). We again found an inverse relationship between the translational efficiency of mRNA isoforms and the number of the uORFs (see Extended Data Fig. 9 for overall analysis and examples). We then examined which of two modes of regulation contributes the most to VHL-dependent changes in translation of these genes: (1) the effect of VHL on translation is a direct consequence of the altered TSS usage, or (2) the effect of VHL on translation is observed across all transcripts associated with these genes,

irrespective of their TSS. To assess this, we recalculated changes in translational efficiency for each gene, omitting either the effect of (1) or (2) from the calculation and compared the results with the experimental measurement, as derived from both parameters. The correlation was much stronger using (1) than (2) (Pearson's $r = 0.83$ and $r = 0.54$, $P < 1 \times 10^{-10}$ and $P < 1 \times 10^{-5}$ respectively, Fig. 4a), indicating that the changes in translational efficiency of these genes were primary due to altered TSS usage.

Importantly, some of the largest effects on translation were associated with alternative TSS usage (*y* axis of Fig. 4a). Of these, Max-interacting protein 1 (*MXI1*), an antagonist of Myc proto-oncogene (MYC)[36], showed the most striking increase in translational efficiency upon *VHL* loss (Fig. 4a,b). 5′ end-seq identified the three most abundant *MXI1* mRNA isoforms, defined by alternative TSS usage (TSS1–TSS3, Fig. 4c), in RCC4 cells. TSS2 and TSS3 isoforms were the dominant isoforms in HIF-repressed RCC4 VHL cells. However, the TSS1 transcript (which has been reported to be HIF1A dependent[37] and bears a different CDS than the other isoforms) was strongly upregulated in *VHL*-defective RCC4 cells (Fig. 4d). Notably, TSS2 and TSS3 mRNA each contain an uORF that is excluded from TSS1 by alternative first exon usage (Fig. 4c). Consistent with the negative effects of uORFs on translation, the TSS1 mRNA isoform was much more efficiently translated than were the TSS2 and TSS3 isoforms (Fig. 4e). Thus, alternative TSS usage associated with *VHL* loss specifically upregulated the translationally more potent isoform, enhancing overall translation. Interestingly, the isoform that is orthologous to this transcript in mice has been reported to manifest stronger transcriptional repressor activity[38]. Taken together, these findings indicate that alternative TSS usage makes major contributions to altered translational efficiency among a subset of HIF-target genes.

**Sensitivity to mTOR among classes of HIF target gene.** Since concurrent dysregulation of HIF and mTOR pathways is frequently observed, we sought to determine how HIF-dependent transcriptional regulation and mTOR-dependent translational regulation interact. Comparison of changes in translational efficiency with mTOR inhibition in RCC4 VHL cells with those in RCC4 cells showed a strong correlation, with the slope of the regression line being slightly less than 1 (Pearson's $r = 0.89$, $P < 1 \times 10^{-10}$, slope = 0.85; Fig. 5a), indicating that mTOR inhibition regulates translation similarly, regardless of HIF status. The effect of mTOR inhibition was slightly weaker in *VHL*-defective cells, probably reflecting a small negative effect of HIF1A on mTOR-target mRNAs, as outlined above. We also analyzed the effect of mTOR inhibition on the expression of genes involved in the HIF signaling pathway. This revealed that two oxygen-sensitive 2-oxoglutrarate-dependent dioxygenases, FIH1 and PHD3 (ref. [39]), were more strongly downregulated than other HIF-pathway-related genes, indicating that mTOR has the potential to affect the cellular responses to hypoxia by several mechanisms (Extended Data Fig. 10a).

We then considered the relationship of HIF-dependent changes in transcription to mTOR-dependent changes in translation. Somewhat surprisingly, we observed no overall association between the two regulatory modes (Spearman's $\rho = 0.04$, $P < 1 \times 10^{-3}$; Fig. 5b). However, more detailed examination of the data revealed that distinct functional classes of mRNAs responded differently. Among transcripts that were induced in *VHL*-defective cells, those encoding glycolytic enzymes were hypersensitive to mTOR inhibition, whereas the translation of genes classified as involved in angiogenesis or vascular processes was much more resistant ($P < 1 \times 10^{-6}$, Mann–Whitney *U* test, Fig. 5c, Extended Data Fig. 10b,c and Supplementary Data 3). To confirm this, we re-analyzed published data using ribosome profiling[14,21] and observed a similar contrast (Extended Data Fig. 10d). Consistent with our overall findings that mRNAs with no uORF and/or a CDS around 1 kb in

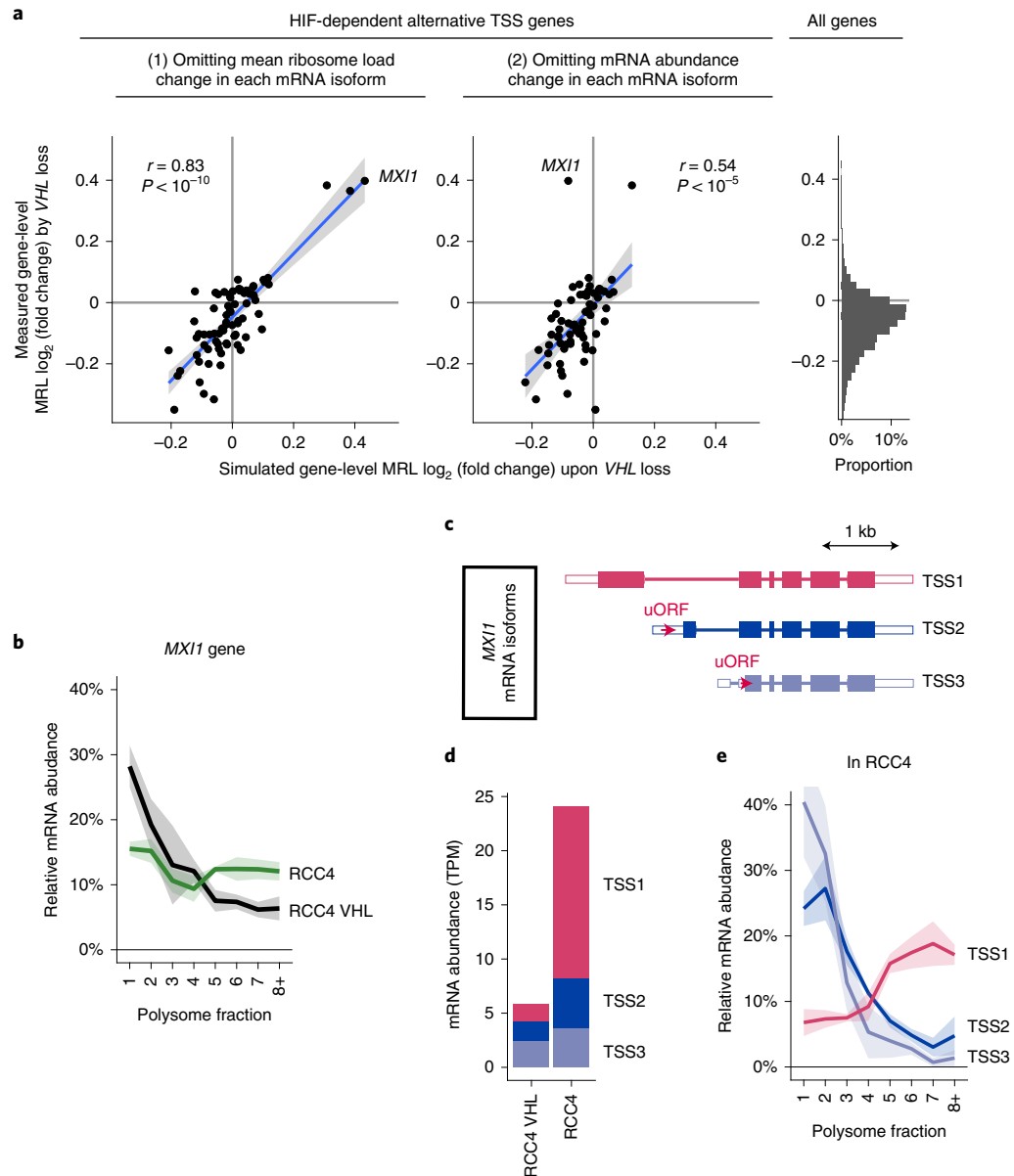

**Fig. 4 | Translational regulation by VHL-dependent alternative TSS usage. a**, Contribution of VHL-dependent alternative TSS usage to changes in translational efficiency of genes following *VHL* loss in RCC4 cells. The scatter plots show the correlations between measured changes in translational efficiency (log₂(fold change) in overall MRL for each gene, *y* axis) and that simulated when omitting one parameter (*x* axis). The analyses are of those genes manifesting an altered polysome distribution on their VHL-dependent alternative transcript. Pearson's product moment correlation coefficient was used to assess the association (*n* = 75 and 70 for (1) and (2), respectively). The blue line indicates the linear model fit by ordinary least squares, and the gray shade shows the standard error. Right panel (the same data as in the upper panel of Fig. 3a), is provided to reference the distribution of changes in translational efficiency amongst the subset of genes manifesting alternative TSS usage to all expressed genes. **b**, Proportion of *MXI1* mRNA distributed across polysome fractions; the line indicates the mean value, and the shaded area shows the s.d. of the data from the three independent clones. **c**, Schematics of the 3 most abundant mRNA TSS isoforms of *MXI1*; the 5′ and 3′ UTR are colored white, and the position of uORFs is indicated by red arrows. **d**, mRNA abundance of each *MXI1* mRNA TSS isoform estimated as transcript per million (TPM) from 5′ end-seq data. Data presented are the mean of the measurements of the three independent clones. **e**, Similar to **b**, but the proportion of each *MXI1* mRNA TSS isoform in RCC4 cells is shown separately.

length were hypersensitive to mTOR, a higher proportion of glycolytic genes were found to bear these features than of genes associated with angiogenesis or vascular processes (Extended Data Fig. 10e). Overall, these findings indicate that full upregulation of the glycolysis pathway requires both HIF and mTOR activity, as would be predicted to occur in *VHL*-defective kidney cancer with mTOR hyperactivation[2].

Of the two mTOR complexes, it is widely accepted that mTORC1 regulates translation[13]. Interestingly, the protein level of

HIF1A has been shown to be positively regulated by both mTORC1 and mTORC2, whereas HIF2A is dependent on only mTORC2 activity[40]. This raises the question of whether the HIF-induced, mTOR-resistant genes that function in angiogenesis or vascular processes might be principally regulated by HIF2A and hence transcriptionally, as well as translationally, resistant to mTORC1 inhibition. To this end, we interrogated pan-genomic data on HIF binding[41]. In agreement with previous studies showing that genes encoding glycolytic enzymes are induced specifically by HIF1A[42],

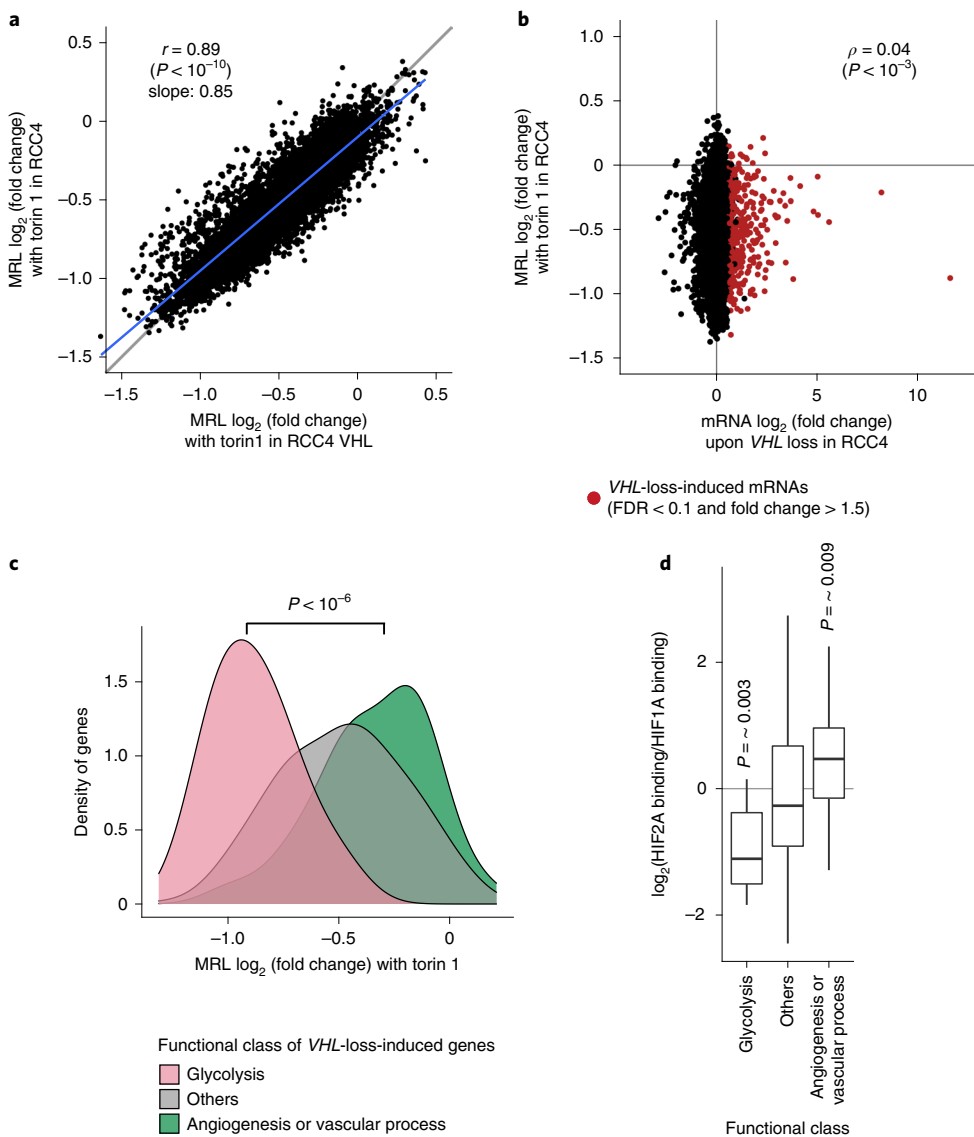

**Fig. 5 | Differential sensitivity to translational inhibition by mTOR among HIF-regulated transcripts encoding proteins with different functions.**
**a**, Comparison of effects of Torin 1 on translational efficiency of genes (expressed as $\log_2$(fold change) in MRL) in RCC4 VHL cells versus RCC4 cells. Pearson's product moment correlation coefficient was used to assess the association ($n=8,429$); the blue line indicates the linear model fit by ordinary least squares. **b**, Comparison of the effect of Torin 1 on translational efficiency with the effect of VHL on transcript abundance in RCC4 cells. Genes showing significant upregulation of mRNA abundance upon *VHL* loss are indicated in red (FDR < 0.1 and fold change > 1.5). Spearman's rank-order correlation coefficient was used to assess the association ($n=8,580$). **c**, Analysis of changes in translational efficiency of genes produced by Torin 1 among the specified functional classes of genes whose mRNAs were induced by *VHL* loss. Functional classes were defined by gene ontology and KEGG orthology. The distributions are shown using kernel density estimation, and compared using the two-sided Mann–Whitney *U* test ($n=12$ and 29 for glycolysis and angiogenesis or vascular-process genes respectively). **d**, Relative ratio of HIF2A and HIF1A binding at the nearest HIF-binding sites to genes induced by *VHL* loss, among the specified functional class of genes. HIF2A and HIF1A binding across the genome were analyzed by ChIP–seq. The ratios within a functional class were compared against the ratios for all other genes using the two-sided Mann–Whitney *U* test ($n=12$, 25, and 268 for glycolysis, angiogenesis or vascular process and others, respectively). Box plots show the median (horizontal lines), first to third quartile range (boxes), and 1.5× interquartile range from the box boundaries (whiskers).

HIF-binding sites near this class of genes had a lower HIF2A/HIF1A binding ratio than did other genes ($P = {\sim}0.003$, Mann–Whitney *U* test, Fig. 5d). This contrasted with a higher HIF2A/HIF1A binding ratio for angiogenesis or vascular-process genes induced in *VHL*-defective RCC4 cells ($P = {\sim}0.009$, Mann–Whitney *U* test, Fig. 5d). Consistent with this, mRNAs of HIF-target angiogenesis or vascular-process genes were also upregulated to a greater extent than other HIF-target genes upon *VHL* loss in 786-O cells, which express only HIF2A ($P = {\sim}0.007$, Mann–Whitney *U* test, Extended Data Fig. 10f). This suggests that they are primarily HIF2A targets, as well as resistant to effects of mTOR inhibition on translation, consistent with a role in correcting a hypoxic and nutrient-depleted environment.

## Discussion
Using a new technology to measure the ribosome load of mRNAs resolved by their TSS, we have characterized the pan-genomic interplay of HIF- and mTOR-dependent transcriptional and translational regulation in *VHL*-defective kidney cancer cells. Importantly, the increased throughput of the technology and use of external

normalization enabled us to directly compare translational effects across the genome for a larger number of interventions than most studies to date.

Our analysis revealed that mTOR inhibition heterogeneously downregulates translation of a very wide variety of mRNAs and demonstrated the hypersensitivity of many genes encoding metabolic enzymes. This suggests a greater role for translational alterations in gene expression and metabolism in mTOR-dysregulated cancer than previously thought.

Our findings confirmed that the HIF pathway primarily regulates transcription, but also revealed that HIF1A represses global translation moderately via mTOR and that HIF regulates the translation of a subset of genes bidirectionally through alternative TSS usage. HIF-dependent alternative TSS usage was often associated with altered translational efficiency and/or altered CDS. Apart from these transcripts, we were surprised to find little or no evidence for HIF-dependent upregulation of translation in *VHL*-defective cells, in contrast to previous reports of a major role for HIF2A in promoting EIF4E2-dependent translation. The original studies demonstrated this action of HIF2A in hypoxia and in *VHL*-defective cells (786-O)[9,10], as were used in this study, but the effect size of HIF2A-dependent translational regulation was not compared with other interventions, such as mTOR inhibition. Although we cannot exclude small effects on some targets, our findings indicate that, at least under the conditions of our experiments, the role of HIF2A–EIF4E2 in promoting translation is at best very limited, even for the genes reported to be regulated by this pathway[9,10].

Previous studies have reported that HIF inhibits mTOR activity through the transcriptional induction of antagonists of mTOR signaling[8,43], raising a question as to whether the use of mTOR inhibitors constitutes a rational approach to the treatment of *VHL*-defective cancer. Our comparative analysis of interventions revealed that the mTOR inhibition by HIF was very much weaker than that by pharmacological inhibition, offering a justification for this therapeutic approach.

To pursue this further, we compared transcriptional targets of HIF and translational targets of mTOR across the genome. Although little or no overall correlation was observed, these analyses revealed marked differences in mTOR sensitivity among HIF transcriptional targets, according to the functional classification of the encoded proteins. HIF1A-targeted genes encoding glycolytic enzymes were hypersensitive to mTOR, whereas HIF2A-targeted genes encoding proteins involved in angiogenesis and vascular process were resistant to mTOR inhibition. Clinically approved mTOR inhibitors primarily target mTORC1 (ref. [16]), and are therefore unlikely to affect HIF2A abundance[40]. Our results suggest that they are unlikely to affect the expression of these classes of HIF2A-target gene. Recently, a new class of drug that prevents HIF2A from dimerizing with HIF1B and hence blocks HIF transcriptional activity has shown promise in the therapy of *VHL*-defective kidney cancer[11,12,16]. Given that we observed few, if any, effects of HIF2A on translation, our results suggest that the combined use of these HIF2A transcriptional inhibitors, together with mTOR inhibitors, should therefore be considered as a rational therapeutic strategy for this type of cancer.

## Online content

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

## Methods

**Overview of the cell line and experimental conditions.** *VHL*-defective kidney cancer cell lines, RCC4 and 786-O, were from Cell Services at the Francis Crick Institute and were maintained in DMEM (high glucose, GlutaMAX Supplement, HEPES, Thermo Fisher Scientific, no. 32430100) with 1 mM sodium pyruvate (Thermo Fisher Scientific, 12539059) and 10% FBS at 37 °C in 5% $CO_2$. Cells were confirmed to be of the correct identity by STR profiling and to be free from mycoplasma contamination.

Hypoxic incubation was performed using an InvivO$_2$ workstation (Baker Ruskinn) in 1% $O_2$ and 5% $CO_2$ for 24 hours. To inhibit mTOR, cells were treated with 250 nM of Torin 1 (Cell Signaling Technology, no. 14379) for 2 hours.

An overview of the experimental interventions and analyses is provided in Supplementary Data 1. Biological replicates are individual experiments using different clones derived from the same cell line. All other replicates are defined as technical replicates.

**Genetic modification of cells.** *Lentiviral transduction.* Reintroduction of *VHL* or the empty vector control was performed using lentiviral transduction. The expression vector (pRRL-hPGK promoter-VHL-IRES-BSD) containing the coding sequence and the last 6 nucleotides of the 5′ UTR of *VHL* (RefSeq ID, NM_000551) and the empty vector control (pRRL-SFFV promoter-MCS-IRES-BSD) were constructed from pRRL-SFFV promoter-MCS-IRES-GFP (provided by K. R. Kranc, Queen Mary University of London). Lentiviruses were prepared from these plasmids, and RCC4 or 786-O cells were transduced with the viruses. Three or four clones each of *VHL*- or empty-vector-transduced RCC4 or 786-O cells were isolated using flow cytometry. These cells were maintained in DMEM (high glucose, GlutaMAX Supplement, HEPES) with 1 mM sodium pyruvate, 10% FBS and 5 μg/mL blasticidin (Thermo Fisher Scientific, A1113903) at 37 °C in 5% $CO_2$. Empty-vector-transduced RCC4 or 786-O cells are referred as RCC4 or 786-O, and *VHL*-transduced RCC4 or 786-O cells are referred as RCC4 VHL or 786-O VHL.

*CRISPR–Cas9-mediated HIF1B or EIF4E2 inactivation of 786-O cells.* CRISPR–Cas9-mediated inactivation of *HIF1B* or *EIF4E2* was performed using the electroporation of gRNA–Cas9 ribonucleoprotein (RNP). CRISPR RNAs (crRNAs) with the following sequences were synthesized by Integrated DNA Technologies (Alt-R CRISPR–Cas9 crRNA):

HIF1B, rGrArCrArUrCrArGrArUrGrUrArCrCrArUrCrArC
EIF4E2 (g1), rGrUrUrUrGrGrArArArGrGrArUrGrArUrGrArCrArGrU
EIF4E2 (g2), rGrGrUrCrCrCrCrArGrGrArCrGrUrArCrArUrG.

The *HIF1B* and *EIF4E2* (g2) gRNA sequences were designed by Integrated DNA Technologies (Hs.Cas9.ARNT.1.AD and Hs.Cas9.EIF4E2.1.AH, respectively), whereas the EIF4E2 (g1) gRNA sequence was designed using an online tool developed by F. Zhang's lab (https://crispr.mit.edu).

To prepare the gRNA, 100 μM of crRNA and 100 μM of tracrRNA (Integrated DNA Technologies, no. 14899756) were annealed in duplex buffer (Integrated DNA Technologies, 11-01-03-01) by incubation at 95 °C for 5 minutes, then at room temperature for 30 minutes. Cas9–gRNA RNP was formed by mixing 10 μM of the annealed tracrRNA–crRNA and 16.5 μg of TrueCut Cas9 protein (Thermo Fisher Scientific, A36498) in PBS, followed by incubation at room temperature for 30 minutes. The RNP was transfected into 786-O cells or 786-O VHL cells (pools of cells were used for HIF1B inactivation, whereas clone 1 of each sub-line was used for EIF4E2 inactivation). Transfections were performed using a 4D-Nucleofector System (Lonza) with a SF Cell Line 4D-Nucleofector X Kit L (Lonza, V4XC-2024) and the EW-113 transfection program. The transfected cells were cultured in DMEM (high glucose, GlutaMAX Supplement, HEPES) with 1 mM sodium pyruvate and 10% FBS at 37 °C in 5% $CO_2$ for at least 3 days, and single clones were isolated using flow cytometry. Inactivation of the target genes was confirmed by Sanger sequencing of the gRNA target region using TIDE analysis[78] and by immunoblotting.

**Immunoblotting.** *Protein extraction.* Cells were grown on 6-cm dishes. Cells were washed with 3 mL of ice-cold PBS and lysed by adding 150 μL of urea SDS lysis buffer (10 mM Tris-HCl pH 7.5, 6.7 M urea, 5 mM DTT, 10% glycerol, 1% SDS, 1× HALT protease and phosphatase inhibitor (Thermo Fisher Scientific, 78447), and 1/150 (v/v) of benzonase (Sigma-Aldrich, E1014-25KU)). The lysate was incubated at room temperature for 30 minutes before mixing with loading buffer (LI-COR Biosciences, 928-40004).

*Immunoblotting.* Proteins were separated by SDS–PAGE using a Mini-PROTEAN TGX Gel (4–15% or 8–16%, Bio-Rad Laboratories, nos. 4561086 and 4561106, respectively) and transferred to Immobilon-FL PVDF Membrane (Sigma-Aldrich, IPFL00010). Membranes were stained using a Revert 700 Total Protein Stain (LI-COR Biosciences, 926-11011). The data acquisition was performed using an Odyssey CLx system (LI-COR Biosciences), and the data were analyzed using Image Studio software (LI-COR Biosciences). The membrane was blocked by incubating in TBS (20 mM Tris-HCl pH 7.6 and 137 mM NaCl) with 5% fat-free milk for 1 hour with shaking at room temperature. The membrane was incubated in Odyssey Blocking Buffer (PBS, LI-COR Biosciences, no. 927-40000) with 0.2% Tween-20 and 1/1,000 (vol/vol) primary antibody (for anti-HIF2A antibody) or TBST

(TBS with 0.1% Tween-20) with 5% fat-free milk and 1/1,000 (vol/vol) primary antibody (for other primary antibodies), with shaking overnight at 4 °C. The membrane was washed three times with TBST and incubated in Odyssey Blocking Buffer (PBS for anti-HIF2A antibody and TBS (LI-COR Biosciences, no. 927-50000) for other primary antibodies) with 0.2 % Tween-20, 0.01% SDS, and 1/15,000 (vol/vol) secondary antibody, with shaking for 1 hour at room temperature. The membrane was washed three times with TBST and once with TBS.

*Antibodies.* The following antibodies were used for the western blotting analysis. Primary antibodies (used at 1/1,000 dilution): anti-VHL (Santa Cruz Biotechnology, sc-135657), anti-HIF1A (BD Biosciences, 610959), anti-HIF2A (Cell Signaling Technology, 7096), anti-HIF1B (Cell Signaling Technology, 5537), anti-EIF4E2 (Proteintech, 12227-1-AP), anti-NDRG1 (Cell Signaling Technology, 9485), anti-SLC2A1 (Cell Signaling Technology, 12939), anti-EGFR (Santa Cruz Biotechnology, sc-373746), and anti-CA9 (Cell Signaling Technology, 5649). Secondary antibodies (used at 1/15,000 dilution): anti-mouse-IgG DyLight 800 (Cell Signaling Technology, 5257) anti-mouse-IgG IRDye 680RD (LI-COR Biosciences, 925-68072), and anti-Rabbit-IgG IRDye 800CW (LI-COR Biosciences, 926-32213).

**Total RNA extraction.** Cells were grown on 6-well plates or 6-cm dishes. Total RNA used for the analysis of unfractionated mRNAs was extracted from the cells using the RNeasy Plus Mini Kit (QIAGEN, 74136), according to the manufacturer's instructions, except for technical replicate 2 of the samples from RCC4 cells (see Supplementary Data 1). For these samples, cells were lysed with 350 μL of Buffer RLT Plus (QIAGEN, 1053393), and total RNA was extracted from the lysate using an RNA clean and concentrator-25 kit (Zymo Research, R1018) with the following modification: 752.5 μL of preconditioned RNAbinding buffer (367.5 μL of RNA binding buffer (supplied with an RNA Clean & Concentrator-25 kit), 367.5 μL of absolute ethanol, and 17.5 μL of 20% SDS) was added to the cell lysate. After mixing, the material was loaded onto the column of an RNA Clean & Concentrator-25 kit, and the manufacturer's instructions were followed for the remaining steps.

**HP5 protocol (polysome profiling).** *Sucrose gradient preparation.* Sucrose gradients were prepared in polyallomer tubes (Beckman Coulter, 326819) by layering 2.25 mL 50% sucrose in 1× polysome gradient buffer (10 mM HEPES pH 7.5, 110 mM potassium acetate, 20 mM magnesium acetate, 40 U/mL RNasin plus (Promega, N2615), 20 U/mL SuperaseIn RNase Inhibitor (Thermo Fisher Scientific, AM2694) and 100 μg/mL cycloheximide (Sigma-Aldrich, C4859-1ML)) under 2.15 mL of 17% sucrose in 1× polysome gradient buffer. Each tube was sealed with parafilm, placed on its side, and kept in the horizontal position at 4 °C overnight to form the gradient[79].

*Cell lysis and fractionation.* Cells were grown on 15-cm dishes. To arrest mRNA translation, the cells (~80% confluency) were treated with 100 μg/mL cycloheximide for 3 minutes. The medium was removed, and the dish was placed on ice during the following steps. Cells were washed with 10 mL of ice-cold PBS with 100 μg/mL cycloheximide. Cells were then lysed by adding 800 μL of polysome lysis buffer (10 mM HEPES pH 7.5, 110 mM potassium acetate, 20 mM magnesium acetate, 100 mM potassium chloride, 10 mM magnesium chloride, 1% Triton X-100, 2 mM DTT, 40 U/mL RNase plus, 20 U/mL SuperaseIn RNase Inhibitor, 1× HALT Protease inhibitor (Thermo Fisher Scientific, 78438), and 100 μg/mL cycloheximide).

The cytoplasmic lysate was homogenized by passage through a 25-G syringe needle 5 times. To remove debris, the lysate was centrifuged at 1,200*g* for 10 minutes at 4 °C, and the supernatant was collected. This material was centrifuged again at 1,500*g* for 10 minutes at 4 °C, and the supernatant was collected. The protein and RNA concentrations were measured using 660-nm Protein Assay Reagent (Thermo Fisher Scientific, 22660) with Ionic Detergent Compatibility Reagent (Thermo Fisher Scientific, 22663) and Qubit RNA BR Assay Kit (Thermo Fisher Scientific, Q10210), respectively.

Lysate was then normalized according to the protein concentration, and 500 μL of the normalized lysate was overlaid on the sucrose gradient, as prepared above. The gradient was ultracentrifuged at 287,980*g* (average; 55,000 r.p.m.) for 55 minutes at 4 °C, with max acceleration and slow deceleration using an Optima LE-80K Ultracentrifuge and SW55Ti rotor (Beckman Coulter). The sucrose gradient was fractionated according to the number of associated ribosomes (from 1 to 8 ribosomes; material lower in the gradient was pooled with the 8 ribosome fraction), as determined by the profile of the absorbance at 254 nm using a Density Gradient Fractionation System (Brandel, Model BR-188). The fractionated samples were then snap-frozen on dry ice.

*External control RNA addition and RNA extraction.* Equal amounts of external control RNA were added to the polysome-fractionated samples after thawing the snap-frozen samples on ice. Commercially available external control RNA, including the ERCC RNA Spike-In Mix-1 kit (Thermo Fisher Scientific, 4456740) that we used, does not have a canonical mRNA cap. This can influence the template-switching reaction efficiency. Thus, the amount of external control RNA

added to the polysome-fractionated samples was determined by preliminary experiments, so as to result in a library containing around 0.1% of reads from the external control RNA.

RNA was extracted from 150 μL of the fractionated samples using an RNA Clean & Concentrator-5 kit (Zymo Research, R1016), using the same procedure to extract RNA from unfractionated cell lysate (described above), and was eluted into 10 μL of water. For a subset of samples, as indicated in Supplementary Data 1, half of the input volume was used, and RNA was eluted in 8 μL of water. The integrity of the purified RNA was confirmed using a Bioanalyzer (Agilent); the median value of RNA integrity number (RIN) for the samples from RCC4 VHL cells was 9.5, indicating that the RNA was largely intact.

**5′ end-seq protocol.** *Primer sequences.* The sequences of oligonucleotide primers used for 5′ end-seq are summarized in Supplementary Data 4. All the primers were synthesized and HPLC-purified by Integrated DNA Technologies.

The 5′ end-seq method involves the following steps.

*Step 1: reverse transcription and template switching.* cDNAs with adapter sequences at both the 5′ and 3′ ends were generated from full-length mRNAs using a combined reverse-transcription and template-switching reaction. The RT primers, containing an oligonucleotide (dT) sequence, were annealed to the poly A tail of mRNAs by incubating 4 μL reaction mix (1.9 μL extracted RNA, 1 μL 10 mM dNTP, 0.1 μL 20 U/μL SUPERaseIn RNase-Inhibitor, and 1 μL 10 μM RT primer) at 72 °C for 3 minutes and holding it at 25 °C. Then, 1 μL of 10 μM template-switching oligonucleotide (TSO), and 5 μL of RT reaction mix (2 μL of 5× RT buffer (supplied with Maxima H Minus Reverse Transcriptase), 2 μM of 5 M betaine, 0.25 μL of water, 0.25 μL of SUPERaseIn RNase-Inhibitor, and 0.5 μL of 200 U/μL Maxima H Minus Reverse Transcriptase (Thermo Fisher Scientific, EP0753)) were added to the reaction. The TSO contained an adapter sequence (the constant region annealed by the PCR primers), an index sequence (to identify the sample source of the cDNA), unique molecular identifiers (UMI), and three riboguanosines at the 3′ end (to facilitate template-switching reaction[80]). To perform the reverse-transcription and template-switching reactions, the mixture was kept at 25 °C for 45 minutes, 42 °C for 25 minutes, 47 °C for 10 minutes, 50 °C for 10 minutes, and 85 °C for 5 minutes, and held at 4 °C.

*Step 2: enzymatic degradation of primers and RNA.* Preliminary experiments indicated that the degradation of unused primers using a single-stranded DNA specific 3′–5′ exonuclease, Exonuclease I, reduced primer dimer artifacts in the subsequent PCR amplification, whereas the degradation of RNA by RNase H improved the yield of cDNA library. Furthermore, it is important to degrade TSO because, if unused TSO contaminates the cDNA library after multiplexing, it confounds the library indexing. Because we suspected that the TSO is resistant to Exonuclease I owing to the riboguanosines at the 3′ end, the TSO contains three deoxyuridines (after the adapter sequence, index sequence, and UMI) so that the TSO can be degraded by the combination of an enzyme-cleaving DNA at a deoxyuridine and Exonuclease I. Importantly, this degrades all the TSO except the adapter, which forms a high-melting-temperature duplex with the cDNA, protecting the cDNA from Exonuclease I. All these reactions were performed in a single step by adding 2 μL enzyme mix (1 μL of Thermolabile USER II (New England Biolabs, M5508L), 0.5 μL of Exonuclease I (New England Biolabs, M0293S), and 0.5 μL of RNase H (New England Biolabs, M0297S)) to the sample, which was incubated at 4 °C for 1 second, 37 °C for 1 hour, 80 °C for 20 minutes, and held at 4 °C.

*Step 3: limited-cycle PCR amplification.* Fifteen microliters of PCR reaction mix (1.25 μL of 10 μM of each PCR primer 1 forward/reverse, 12.5 μL of KAPA HiFi HotStart Uracil+ ReadyMix (Roche, KK2802), and 1.25 μL of water) was added to the RT reaction, and limited-cycle PCR amplification was performed by keeping the mixture at 98 °C for 3 minutes; 98 °C for 20 seconds, 67 °C for 15 seconds, and 72 °C for 6 minutes (4 cycles); and 72 °C for 5 minutes; and the mixture was then held at 4 °C.

*Step 4: multiplexing and optimized PCR cycle amplification.* After adding 37.5 μL ProNex beads (Promega, NG2002) to each sample, up to 16 samples were multiplexed. The cDNA library was purified according to the manufacturer's instructions, eluted into 42 μL of 10 mM Tris-HCl, pH 7.4, and then re-purified using ProNex beads (1.5:1 vol/vol ratio of beads to sample) and eluted into 45 μL of 10 mM Tris-HCl, pH 7.4. The library was reamplified by preparing PCR reaction mix (20 μL of cDNA library, 25 μL of KAPA HiFi HotStart Uracil+ ReadyMix, and 2.5 μL of 10 μM each PCR primer 1 forward/reverse), and the mixture was kept at 98 °C for 3 minutes; 98 °C for 20 seconds, 67 °C for 15 seconds, and 72 °C for 6 minutes (4–6 cycles (see beelow)); and 72 °C for 5 minutes; and the mixture was then held at 4 °C. The number of PCR cycles for each amplification was determined by a pilot experiment using quantitative PCR (qPCR) to ensure that the amplification was at the early linear phase. The amplified cDNA library was purified using ProNex beads, as above, and eluted into 26 μL of 10 mM Tris-HCl, pH 7.4. The purified cDNA library was quantified using a Qubit dsDNA HS Assay Kit (Thermo Fisher Scientific, Q32851).

*Step 5: tagmentation.* Tagmentation with Tn5 transposase was performed on 90-ng aliquots of the cDNA library using an Illumina DNA Prep kit (Illumina, 20018704), according to the manufacturer's instructions.

*Step 6: PCR amplification of mRNA 5′-end library.* The 'tagmented' library was attached to the beads of an Illumina DNA Prep kit. Limited-cycle PCR amplification was performed by adding 50 μL of the following reaction mix (2.5 μL of 10 μM each of the PCR primer 2 forward/reverse, 20 μL of Enhanced PCR Mix (supplied with an Illumina DNA Prep kit), and 27.5 μL of water) and using a program of 68 °C for 3 minutes; 98 °C for 3 minutes; 98 °C for 45 seconds, 62 °C for 30 seconds, and 68 °C for 2 minutes (3 cycles); and 68 °C for 1 minute; and it was then held at 10 °C. The PCR primers used here anneal to the TSO and an adapter added by tagmentation, and thus specifically amplify DNA fragments containing 5′ ends of mRNAs. The amplified mRNA 5′-end library was purified using ProNex beads, as above, and eluted into 25 μL of 10 mM Tris-HCl (pH 7.4).

The mRNA 5′-end library was reamplified by preparing a PCR reaction mix (10 μL of the mRNA 5′-end library, 25 μL KAPA HiFi HotStart ReadyMix, 2.5 μL of 10 μM each of PCR primer 3 forward/reverse (containing i5 and i7 index sequences), and 12.5 μL water), and the mixture was kept at 98 °C for 3 minutes; 98 °C for 20 seconds, 62 °C for 15 seconds, and 72 °C for 30 seconds (5 cycles (cycle number determined by a pilot experiment to define the early linear phase, as described above)); and 72 °C for 5 minutes; and it was then held at 4 °C. The mRNA 5′-end library was again purified using ProNex beads (1.4:1 vol/vol ratio of beads to sample) according to the manufacturer's instructions, and eluted into 20 μL of 10 mM Tris-HCl, pH 7.4. The purified mRNA 5′-end libraries were multiplexed again and then sequenced on HiSeq 4000 (Illumina) using paired-end (2×100 cycles) and dual-index mode.

**RT–qPCR.** RNAs extracted from polysome-fractionated samples were converted into cDNAs using the same protocol as the 5′ end-seq protocol described above, except that the anchored oligonucleotide dT primer (Integrated DNA Technologies, 51-01-15-08) was used, and the TSO was omitted from the reaction. The cDNA was purified using an RNA Clean and concentrator-5 kit (Zymo Research, R1016) according to the manufacturer's instructions. qPCR was performed using TaqMan Fast Advanced Master Mix (Thermo Fisher Scientific, 4444557) according to the manufacturer's instructions with the mRNA isoform-specific primers and Taqman probes summarized in Supplementary Data 4. All the primers were synthesized by Integrated DNA Technologies. Quantification of mRNAs in each fraction was normalized to the quantification of ERCC-0002 RNA in the same fraction.

**Overview of computational data analyses.** Data analyses were performed using R (4.0.0)[44] and the following packages (data.table (1.12.8)[45], dplyr (1.0.0)[46], stringr (1.4.0)[47], magrittr (1.5)[48], and ggplot2 (3.3.1)[49]) were used throughout.

The following reference data were used to annotate the data: human genome: hg38, obtained via BSgenome.Hsapiens.UCSC.hg38 (1.4.3)[50]; human transcripts: RefSeq[51] (GRCh38.p13) and GENCODE[52] (GENCODE version 34: gencode. v34.annotation.gtf) (these two reference data were combined and redundant GENCODE entries that have a corresponding RefSeq annotation were removed).

Prior to the high-throughput DNA-sequencing data analysis, sequencing data from the technical replicates were concatenated. Data are presented as the mean value of the biological replicates.

TSS boundaries and their associated mRNA isoforms were identified by 5′ end-seq of total (unfractionated) mRNAs. The TSSs assigned to a particular gene were those mapping within 50 base pairs of that gene locus, as specified by RefSeq and GENCODE. The abundance of the mRNA isoform associated with each TSS is the number of reads starting from that TSS. The gene-level mRNA abundance is the sum of these isoforms for the relevant gene.

**Statistics.** The correlation of two variables was analyzed with the cor.test function of R to calculate statistics on the basis of Pearson's product moment correlation coefficient or Spearman's rank correlation coefficient. The difference between two distributions was tested using the two-sided Mann–Whitney U test (for two independent samples) or the two-sided Wilcoxon signed-rank test (for paired samples). To analyze the effect size of the Wilcoxon signed-rank test, the matched-pairs rank biserial correlation coefficient[53] was calculated using the wilcoxPairedRC function of the rcompanion package (2.3.26)[54].

Kernel density estimation was performed using the geom_density function of the ggplot2 package with the parameter, bw = SJ.

**Sequencing read alignment.** *Read pre-processing.* The sequence at positions 1–22 of read 1 is derived from the TSO and was processed before mapping. First, the UMI located at positions 10–16 was extracted using UMI-tools (1.0.1)[55]. Note that the UMI was not used in the analyses because we found that the diversity of UMI was not sufficient to uniquely mark non-duplicated reads. Next, the library was demultiplexed using an index sequence located at positions 1–8, after which the constant regions of the TSO located at position 9 and positions 17–22 were removed using Cutadapt (2.10)[56] with the parameters, -e 0.2–discard-untrimmed.

*Read alignment.* The pre-processed reads were first mapped to cytoplasmic rRNAs (NR_023363.1 and NR_046235.1), mitochondrial ribosomal rRNAs (ENSG00000211459 and ENSG00000210082), and ERCC external control RNAs (https://www-s.nist.gov/srmors/certificates/documents/SRM2374_putative_T7_products_NoPolyA_v1.fasta) using Bowtie2 software (2.4.1)[57] with the following parameters: -N 1–un-conc-gz. The unmapped reads were then aligned to the human genome (hg38: sequence obtained via BSgenome.Hsapiens.UCSC.hg38[50]) with the annotation described above using STAR software (2.7.4a) in two-pass mode[58] with the following parameters: –outFilterType BySJout–outFilterMultimapNmax 1.

**Definition of TSS peaks and boundaries.** To define TSS clusters, we considered two widely used peak callers, paraclu (9)[59] and decomposition-based peak identification (dpi, beta3)[60] software. Our preliminary analysis indicated that paraclu software was more accurate in determining total peak area, whereas dpi was more accurate in resolving peaks within multimodal clusters. To obtain the most accurate resolution and quantification of TSS clusters, we therefore combined the strength of these programs and included information from existing large-scale database using the following four-step procedure.

*Step 1: definition of cluster areas.* Using the standard workflow of paraclu software on pooled data from normoxic cells, RCC4, RCC4 VHL, 786-O, and 786-O VHL, cluster areas of 5′ termini were identified.

*Step 2: definition of TSS clusters within cluster areas.* The cluster areas defined above were further resolved by combining above data with FANTOM5 data and using dpi software, as was originally used for FANTOM5, to resolve bona fide subclusters within the data. Internal sub-cluster boundaries were defined as the midpoint between adjacent dpi-identified peaks.

*Step 3: quality controls and filters.* Artifactual clusters of 5′ termini, potentially generated by internal TSO priming, were filtered on the basis of a low (<15%) proportion of reads bearing non-genomic G between the TSO and mRNA, as the template-switching reaction commonly introduces such bases at the mRNA cap but not following internal priming[4]. Since mitochondrial mRNAs are not capped, these transcripts were filtered if they did not overlap an annotated site.

A further filter was applied to remove TSS subclusters of low-abundance mRNA isoforms whose biological significance is unclear; low abundance was defined as ≤10% of the most abundant mRNA isoform for the relevant gene in any of the analyses.

*Step 4: final assignment of TSS boundaries.* To provide the most accurate identification of the TSS peaks and their boundaries, the resolved and filtered peaks from step 3 were mapped back onto the input cluster areas as defined in step 1, and boundaries were set at the midpoint between filtered peaks.

**Assignment of transcripts to TSS.** To identify mRNA features that might affect translational efficiency, we used base-specific information on 5′ termini and assembled paired-end reads starting from each TSS (StringTie software, 2.1.2 (ref. [61])) to define the primary structure of the 5′ portion of the transcript. We then used homology with this assembly to assign a full-length transcript from RefSeq and GENCODE. The CDS of the assigned transcript was then used for the analysis. In small number of cases, where this TSS was downstream of the start codon, we took the most upstream in-frame AUG sequence to redefine the CDS. The most abundant primary structure from each TSS and its CDS were then used for calculation of the association of mRNA features with mean ribosome load (see below). Details of this process are given in the computational pipeline.

**mRNA feature evaluation.** Features within the mRNA (for example TOP motif, structure near cap) were evaluated at base-specific resolution using the following formula:

$$\begin{pmatrix} RNA\ feature\ value \\ for\ an\ mRNA\ TSS\ isoform \end{pmatrix} = \sum_{i=1}^{n} \frac{mRNA\ abundance_i \times mRNA\ feature\ value_i}{\sum_{i=1}^{n} mRNA\ abundance_i}$$

where $i$ is a base position within the TSS, $n$ is the linear sequence extent of the TSS, *mRNA feature value$_i$* is the value of mRNA feature for the isoform transcribed from position $i$, and *mRNA abundance$_i$* is the mRNA abundance of the isoform transcribed from position $i$. The values were rounded to the nearest integer; a rounded value of 0 being taken as the absence of the feature.

All non-overlapping uORFs, starting from an AUG, were identified using the ORFik package (1.8.1)[62]. Kozak consensus score was calculated by the kozakSequenceScore function of the ORFik package. Using the mode including G-quadruplex formation, the minimum free energy (MFE) of predicted RNA structures was estimated using RNALfold (ViennaRNA package, 2.3.3)[63]. The MFE of RNA structures near the cap was that of the first 75 nucleotides. The MFE of the region distal to the cap was that of entire 5′ UTR minus the first 75 nucleotides. The position of a TOP motif was defined as the position of the 5′ most pyrimidine base, and its length was defined as that of the uninterrupted pyrimidine tract from that base.

The effect of HIF-dependent alternate TSS usage on CDS was defined by alteration in the genomic position of the start codon (Extended Data Fig. 8 and Supplementary Data 2). Expressed isoforms of a gene were defined as those with an abundance greater than 10% of that of the most highly expressed isoform of the same gene in either RCC4 VHL or RCC4 cells.

**Functional annotation of genes.** *Functions.* Functional classes of genes were defined by KEGG orthology[64], as indicated by the following KEGG IDs. Transcription factors: 03000, Transcription machinery: 03021, Messenger RNA biogenesis: 03019, Spliceosome: 03041, Cytoplasmic and mitochondrial ribosome: 03011 (genes with the name starting with MRP and DAP3 were categorized as mitochondrial ribosomes), Translation factors: 03012, Chaperones and folding catalysts: 03110, Membrane trafficking: 04131, Ubiquitin system: 04121, and Proteasome: 03051; Glycolysis: hsa00010, Pentose phosphate pathway: hsa00030, TCA cycle: hsa00020, Fatty acid biosynthesis: hsa00061 and hsa00062, Fatty acid degradation: hsa00071, Oxphos: hsa00190, Nucleotide metabolism: hsa00230 and hsa00240, and Amino acid metabolism: hsa00250, hsa00330, hsa00220, hsa00270, hsa00260, hsa00340, hsa00310, hsa00360, hsa00400, hsa00380, hsa00350, hsa00290, and hsa00280.

Genes associated with angiogenesis or vascular process were defined by referencing to gene ontology (GO)[65] database: GO:0003018, vascular process in circulatory system; GO:0001525, angiogenesis.

*Analysis of existing literatures describing mTOR targets.* In the analyses comparing HP5 data with previously published studies reporting the effects of mTOR inhibition[14,21,32,33], we followed the definition of mTOR hypersensitive genes in the original reports; for Hsieh et al. and Larsson et al., the genes showing changes in translation with PP242 were used; for Morita et al., genes described in Fig. 1b of the paper[33] were used. Since the data of Thoreen et al. were obtained using mouse cells, we mapped mouse genes to human genes using the gorth function of the gprofiler2 package (0.1.9)[66]. Since Hsieh et al. did not supply values for changes in translational efficiency for all genes, we took this data from Xiao et al.[67], who calculated the relevant values using the data from the original report.

To define known activities of mTOR via any mode of regulation except translational regulation (as indicated in Fig. 2c, first row), we considered review articles by Saxton et al.[13] and Morita et al.[68]. Known systematic translational downregulation by mTOR inhibition (as indicated in Fig. 2c, second row) was defined from previous genome-wide studies listed above[14,21,32]. A class of targets was defined as systematically regulated if ≥10% of genes in the class were identified as mTOR hypersensitive or resistant in any of these previous studies[21,32] or highlighted in the original report.

**Analyses of differential mRNA expression upon *VHL* loss.** The identification of differentially expressed genes and the calculation of log$_2$(fold change in mRNA abundance) upon *VHL* loss were performed using the DESeq2 package (1.28.0)[69]. Genes with an FDR < 0.1 and either log$_2$(fold change) > log$_2$(1.5) or < −log$_2$(1.5) were defined as upregulated or downregulated, respectively.

HIF-target genes (as considered in Extended Data Fig. 10f) were defined as those upregulated upon *VHL* loss in RCC4 cells. For this analysis, genes with very low expression in both 786-O and 786-O VHL cells, as identified by the DESeq2 package, were excluded from the analysis.

**Analysis of alternative TSS usage upon *VHL* loss.** Genes manifesting alternative TSS usage upon *VHL* loss were identified using the approach described by Love et al.[70]. Briefly, TSSs for mRNA isoforms with very low abundance were first filtered out using the dmFilter function of the DRIMSeq package (1.16.0)[71] with the parameters min_samps_feature_expr = 2, min_feature_expr = 5, min_samps_feature_prop = 2, min_feature_prop = 0.05, min_samps_gene_expr = 2, min_gene_expr = 20. The usage of a specific TSS relative to all TSSs was then calculated by DRIMSeq with the parameter add_uniform = TRUE.

The significance of changes in TSS usage upon VHL loss for a particular gene was analysed by the DEXSeq package[72]. The FDR was calculated using the stageR package (1.10.0)[73], with a target overall FDR < 0.1. For genes with significant changes in VHL-dependent TSS usage, a *VHL*-dependent alternative TSS was selected as that showing the largest fold change upon *VHL* loss (FDR < 0.1), and a base TSS was selected as that showing the highest expression in the presence of VHL. In these calculations, the DESeq2 and apeglm (1.10.0) package[74] were used to incorporate data variance to provide a conservative estimate of fold change and standard error.

To provide the highest stringency definition, genes manifesting *VHL*-dependent alternative TSS usage were further filtered by the proportional change > 5%, the absolute fold change > 1.5, and the significance of the difference in fold change between the alternate TSS and the base TSS (assessed by non-overlapping 95% confidence intervals).

For the comparative analysis of the *VHL*-dependent alternate TSS usage in various conditions (Extended Data Fig. 7), genes with very low expression that did not meet a criterion of 20 read counts in more than 1 sample were excluded.

**Calculation of mean ribosome load.** Mean ribosome load was calculated using the following formula:

$$\frac{\sum_{i=1}^{8} \left\{ \left( \begin{array}{c} associated\ ribosome\ number \\ for\ fraction\ i\ (= i) \end{array} \right) \times \left( \begin{array}{c} normalized\ read\ count\ of\ the\ mRNA \\ for\ fraction\ i \end{array} \right) \right\}}{\sum_{i=1}^{8} \left( \begin{array}{c} normalized\ read\ count\ of\ the\ mRNA \\ for\ fraction\ i \end{array} \right)}$$

The mRNA abundance values for each polysome fraction were normalized by the read count of the external control using the estimateSizeFactors fraction of the DESeq2 package. Very-low-abundance mRNAs that did not meet a criterion of six read counts in more than six samples were excluded.

**Statistical analysis of differences in polysome distribution.** *VHL-dependent alternative TSS mRNA isoforms.* To define VHL-dependent alternative mRNA isoforms with a different translational efficiency with reference to all other isoforms from the same gene, the significance of changes in their polysome profile was determined by considering the ratio of mRNA abundances as a function of polysome fraction using the DEXSeq package (1.34.0)[72]. The false-discovery rate (FDR) was calculated using the stageR package[73], with the target overall FDR < 0.1.

*Differentially translated mRNA isoforms from the same gene.* In analysis of two most differentially translated mRNA isoforms transcribed from the same gene (for Fig. 1f), each of these isoforms was censored for statistically significant differences from all other isoforms of the same gene using the same analysis as above.

*Changes in response to mTOR inhibition.* To identify genes that were hypersensitive or resistant to mTOR inhibition, genes manifesting a significant change in polysome distribution upon mTOR inhibition, compared to the population average, were first identified using the DESeq2 package[72] with the internal library size normalization and the likelihood ratio test. The genes with a significant change (FDR < 0.1) were classified as hypersensitive or resistant to mTOR inhibition if the log2 fold change of the mean ribosome load was lower or higher than the median of all expressed genes.

**Simulation of changes in translational efficiency with omitting a parameter.** $\text{Log}_2$(fold change) in mean ribosome load of a gene upon *VHL* loss can be expressed by the following formula:

$$log2 \left( \frac{\sum_{i=1}^{n} (MRL_{no\ VHL,\ i} \times \%\ mRNA\ abundance_{no\ VHL,i})}{\sum_{i=1}^{n} (MRL_{VHL,\ i} \times \%\ mRNA\ abundance_{VHL,i})} \right)$$

In this formula, *i* is mRNA isoform *i* (out of *n* mRNA isoforms), $MRL_{no\ VHL}$ or $MRL_{VHL,i}$ is the mean ribosome load of isoform i in RCC4 or RCC4 VHL cells, and % *mRNA abundance*$_{no\ VHL}$ or % *mRNA abundance*$_{VHL,i}$ is the percentage abundance of isoform *i* relative to that of all isoforms in RCC4 or RCC4 VHL cells.

To assess the contribution of alternative TSS usage to changes in mean ribosome load of a gene, we tested a simulation that omitted the VHL-dependent changes in translational efficiency within each mRNA isoform using the following formula:

$$log2 \left( \frac{\sum_{i=1}^{n} (MRL_{average,i} \times \%\ mRNA\ abundance_{no\ VHL,i})}{\sum_{i=1}^{n} (MRL_{average,i} \times \% mRNA\ abundance_{VHL,i})} \right)$$

In this formula, $MRL_{average,\ i}$ is the combined average of $MRL_{no\ VHL,\ i}$ and $MRL_{VHL,\ i}$ as defined above. When values for either of $MRL_{no\ VHL,\ i}$ and $MRL_{VHL,\ i}$ are missing, these values are excluded from the calculation of the average.

To assess the contribution of VHL-dependent changes in translational efficiency within each mRNA isoform to changes in mean ribosome load of a gene, we tested a simulation which omitted the VHL-dependent changes in TSS usage using the following formula:

$$log\ 2 \left( \frac{\sum_{i=1}^{n} (MRL_{no\ VHL,i} \times \%\ mRNA\ abundance_{average,i})}{\sum_{i=1}^{n} (MRL_{VHL,i} \times \%\ mRNA\ abundance_{average,i})} \right)$$

In this formula, % *mRNA abundance*$_{average,\ i}$ is the combined average of % *mRNA abundance*$_{no\ VHL,\ i}$ and % *mRNA abundance*$_{VHL,\ i}$ defined above. When values for either of $MRL_{no\ VHL,\ i}$ and $MRL_{VHL,\ i}$ are missing, these genes were excluded from the analysis.

**Generalized additive model to predict mean ribosome load.** A generalized additive model was used to predict mean ribosome load of mRNAs from the preselected mRNA features. To test the model, a cross-validation approach was deployed to predict the MRL of the top 50% expressed genes on 4 randomly selected chromosomes, which were excluded from the training data used to derive the model. To provide an accurate estimate of the model's performance, this process was repeated ten times, and the median value of the coefficient of determination ($R^2$) was calculated.

For model construction, the gam function of the mgcv package (1.8-31)[75] of R was used, deploying thin-plate regression splines with an additional shrinkage term (with the parameter, bs = 'ts') and restricted maximum likelihood for the selection of smoothness (with the parameter, method = 'REML'). The analysis was restricted to mRNAs with a 5′ UTR length longer than 0 nt and a CDS length longer than 100 nt; 5′ UTR and CDS length were $log_{10}$-transformed, and the MFE values of RNA structures were normalized by the segment length (nt).

**Principal component analysis.** Library-size normalization and a variance-stabilizing transformation were applied to the mRNA abundance data using the vst function of the DESeq2 package[69] with the parameter, blind = TRUE. Principal component analysis of the transformed data was performed for genes showing the most variance (top 25%) using the plotPCA function of the DESeq2 package.

**GO or KEGG orthology enrichment analysis.** GO or KEGG orthology enrichment analysis of the selected set of genes compared to all the expressed genes in the data was performed using the gost function of the gprofiler2 package[66].

**Analysis of HIF2A/HIF1A binding ratio near VHL-regulated genes.** HIF1A and HIF2A ChIP–seq data from Smythies et al.[41] were used to analyze HIF-binding sites across the genome. HIF1A- or HIF2A-binding sites were defined as the overlap of the peaks identified by ENCODE ChIP–seq pipeline (https://github.com/ENCODE-DCC/chip-seq-pipeline2) and those by MACS2 software (2.2.7.1)[76]. For this purpose, the ChIP–seq reads were aligned to the human genome using Bowtie2 software, and the aligned reads were analyzed by ENCODE ChIP–seq pipeline to identify the peaks. The blacklist filtered and pooled replicate data generated by the pipeline were analyzed by MACS2 software with the following parameters (callpeak -q 0.1–call-summits). The position of the binding sites was defined as the position of the hypoxia response element (HRE, RCGTG sequence) closest to the peak summits identified by MACS2 software. If the binding site did not contain an HRE within 50 bp of the peak summit, it was filtered out. Data on HIF1A and HIF2A binding, as defined above, were merged, and the HIF2A/HIF1A binding ratio was estimated using the DiffBind package (2.16.0)[77] with the parameters minMembers = 2 and bFullLibrarySize = FALSE.

**Reporting summary.** Further information on research design is available in the Nature Research Reporting Summary linked to this article.

## Data availability

Sequence data generated during this study are available from ArrayExpress (HP5: E-MTAB-10689, 5′ end-Seq of total mRNAs: E-MTAB-10688). Additional unprocessed data are provided as Source data. The following reference data were used; human genome: hg38, obtained via BSgenome.Hsapiens.UCSC.hg38 (1.4.3); human transcripts: RefSeq57 (GRCh38.p13) and GENCODE58 (GENCODE version 34: gencode.v34.annotation.gtf). Processed data files are provided as Supplementary Data and Source Data. The list of samples that were analyzed for this study is provided as Supplementary Data. Source data are provided with this paper.

## Code availability

The computational pipeline used for the data analysis is available on GitHub (https://github.com/YoichiroSugimoto/20211102_HP5_HIF_mTOR) and Zenodo (https://doi.org/10.5281/zenodo.6583247).

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

## Acknowledgements

We thank the advanced sequencing facility team at the Francis Crick Institute for Illumina HiSeq sequencing; Ratcliffe group members for support and discussion; and M. Cockman for critical reading of the manuscript. This work was supported by the Francis Crick Institute which receives its core funding from Cancer Research UK (CC2092), the UK Medical Research Council (CC2092), and the Wellcome Trust (CC2092). P. J. R. is also supported as a distinguished scholar of the Ludwig Institute for Cancer Research and by the Wellcome Trust (106241/Z/14/Z). For the purpose of Open Access, the authors have applied a CC BY public copyright licence to any Author Accepted Manuscript version arising from this submission.

## Author contributions

Y. S. and P. J. R. conceived the project. Y. S. performed experiments and data analysis. Y. S. and P. J. R. contributed to the interpretation of the data. Y. S. and P. J. R. wrote the manuscript.

## Competing interests

P. J. R. is a scientific co-founder and equity holder in ReOx Ltd. He is a non-executive director of Immunocore Ltd and holds a consultancy with IDP Discovery Pharma SL. Y. S. declares no competing interests.

## Additional information

**Extended data** is available for this paper at https://doi.org/10.1038/s41594-022-00819-2.

**Correspondence and requests for materials** should be addressed to Peter J. Ratcliffe.

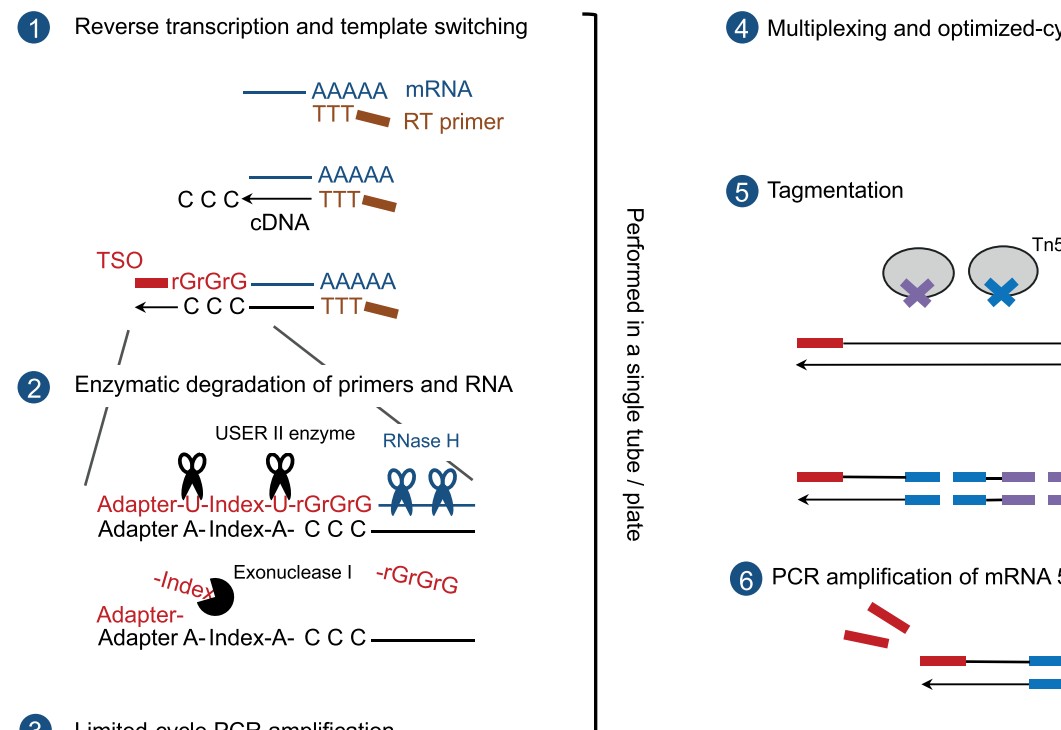

**Extended Data Fig. 1 | Overview of 5′ end-Seq protocol.** Schematic representation of the 5′ end-Seq protocol (see also Methods). 1. The reverse transcription is primed with an adapter containing an oligo (dT) sequence. The reverse transcriptase used for 5′ end-Seq adds additional non-templated cytidine residues beyond the cap, to the 3′ end of the cDNAs. This polycytidine sequence anneals to a polyriboguanosine sequence contained in the template switch oligo (TSO), and the reverse transcriptase switches the template from the mRNA to the TSO to add the complementary sequence of the TSO at the 3′ end of the cDNAs. An indexing sequence contained in the TSO to identify the sample source of the cDNAs is reverse transcribed in this process. 2. Unused primers and RNA are degraded using the combination of a single-stranded DNA specific exonuclease (Exonuclease I), an enzyme cleaving DNA at deoxyuridine (Thermolabile USER II enzyme), and RNase H. This step leaves the adapter sequence of the TSO (the constant region) annealing to the cDNA due to the high melting temperature of this duplex, which protects the cDNA from Exonuclease I. 3. The full-length cDNA library is amplified using limited cycle PCR amplification. 4. The libraries from different samples are multiplexed and the multiplexed libraries are amplified using PCR and an optimized cycle number. 5. Amplified libraries are fragmented and adapter tagged using tagmentation. 6. mRNA 5′ end library suitable for high-throughput DNA sequencing is generated using PCR amplification with primers annealing to the TSO and the appropriate tagmentation adapter.

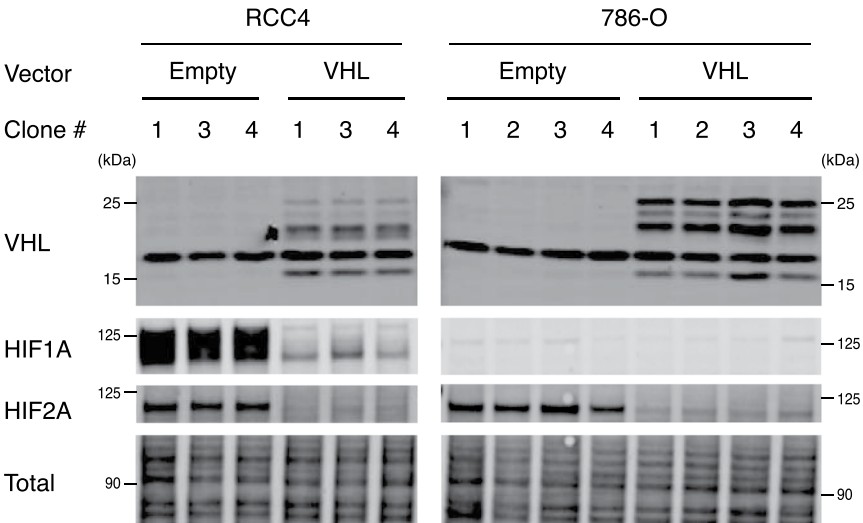

**Extended Data Fig. 2 | Establishment of cell lines.** Immunoblotting analysis of RCC4 or 786-O cells re-expressing either wild type VHL or empty vector alone (n = 3 or 4 experiments in independent clones of RCC-4 and 786-O cells). The successful reintroduction of VHL was confirmed by the expression of VHL protein and degradation of HIF1A and/or HIF2A protein. Similar protein loading across lanes was confirmed by total protein staining. Note that multiple species of VHL were observed consistent with previous studies. In part, they arise from an internal start codon in VHL that produces an 18 kDa isoform[81], but the precise origin of additional species has not been established[82].

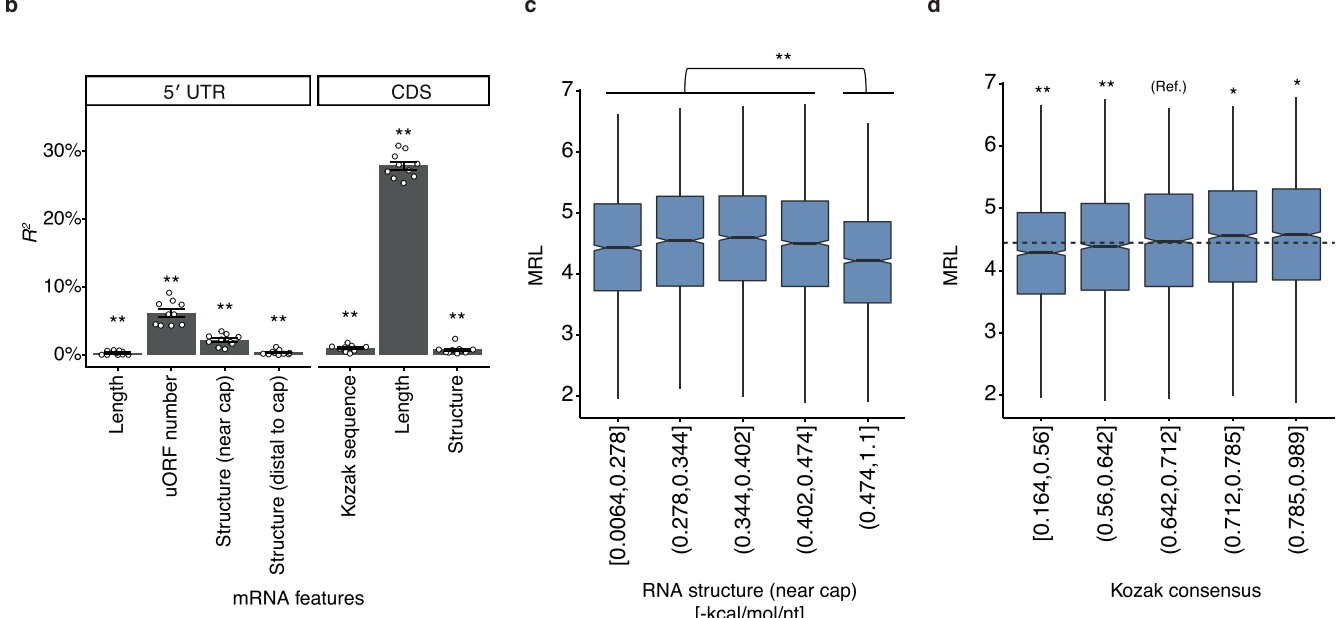

**a**

**b**

**c**

**d**

Extended Data Fig. 3 | See next page for caption.

**Extended Data Fig. 3 | mRNA features predicting mean ribosome load. (a)** Proportion of mRNA isoforms in relation to the total mRNA across polysome fractions for selected genes, as measured by HP5 (upper panel) or RT-qPCR (lower panel). The line indicates the mean value while the shaded area shows the standard deviation of assays using 3 independent clones for the HP5 data, or 2 technical replicates for the RT-qPCR data, respectively. The examples have been selected to compare data on genes where HP5 defined different mRNA isoforms (the schematics are shown below the line plots). In some cases, the resolution provided by RT-qPCR was less than HP5, in which case integration of the HP5 data was performed to permit quantitative comparisons between HP5 and RT-qPCR. *Different upstream or downstream mRNA isoforms not resolved by RT-qPCR and are grouped. **Downstream mRNA isoform not separately resolved by RT-qPCR therefore resolved species comprise upstream and upstream+downstream mRNAs.* **(b)** Proportion of variance in mean ribosome load (MRL) between mRNAs that is explained by a single mRNA feature (expressed as $R^2$) using a generalized additive model (mean ± s.e.m of 10 iterations of cross validation). The significance of mRNA features in predicting MRL was determined by the Wald test. Length, log10 sequence length (nucleotides, nts); Structure (near cap, first 75 nts; distal to cap, rest of the 5′ UTR), inverse of minimum free energy per nucleotide of predicted RNA structures; Kozak consensus, match score to the consensus sequence. The analysis identified that CDS length, uORF number, stability of RNA structures near cap, and Kozak consensus score were the four most predictive features. **(c)** MRL as a function of the stability of RNA structures near cap. mRNAs were ranked by their RNA structural stability, and split into 5 groups according to the rank; the intervals of the stability are indicated on the x-axis. MRL for mRNAs with less stable structures was compared with the most stable group using the two-sided Mann-Whitney *U* test. **(d)** Similar to **c**, but MRL as a function of Kozak consensus score. The median value of MRL for all mRNAs is shown by a dashed line. MRL for mRNAs with the indicated Kozak consensus score was compared to that with the score of 0.642 to 0.712, using the two-sided Mann-Whitney *U* test. **(a-d)** Data are for RCC4 VHL cells. Boxplots show the median (horizontal lines), first to third quartile range (boxes), and 1.5× interquartile range from the box boundaries (whiskers). **(b-d)** * $p < 0.05$, ** $p < 0.005$. *P* values were adjusted for multiple comparisons using Holm's method. Details of the sample sizes and exact *p* values are summarized in Supplementary Information.

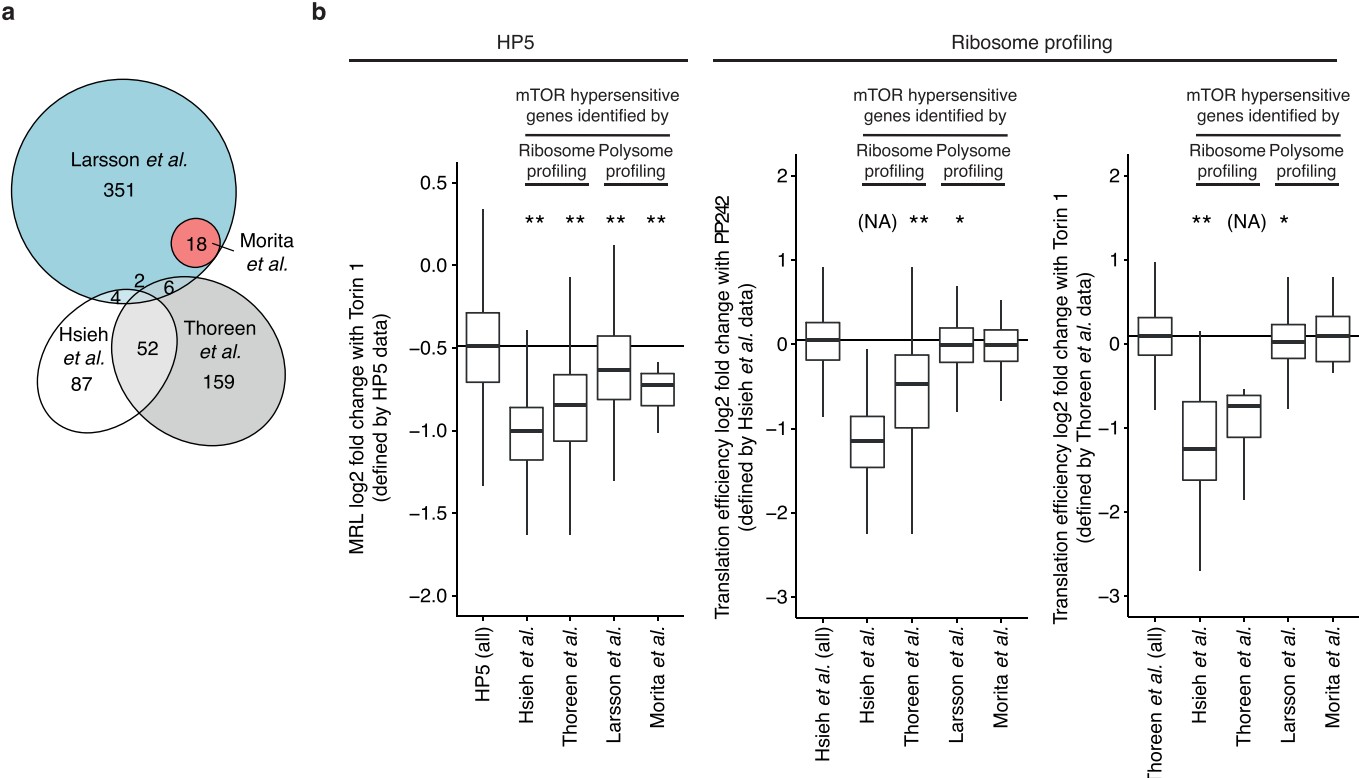

**Extended Data Fig. 4 | Comparison of the HP5 data with the previous reports. (a)** Venn diagram showing the numbers and overlap of mTOR hypersensitive genes identified by previous studies[14,21,32,33]. **(b)** Boxplots showing changes in translation upon mTOR inhibition as measured by the indicated study (HP5, left-hand panel; ribosome profiling, centre and right-hand panels) for the genes identified as mTOR hypersensitive in each of the previous studies[14,21,32,33]. In each panel, the left-hand boxplot shows the changes in mean ribosome load (MRL) or translational efficiency of all expressed genes in that study; horizontal line, median value. Responses of mTOR hypersensitive genes identified by the indicated study were compared against responses for all expressed genes using the two-sided Mann-Whitney $U$ test. * $p < 0.05$, ** $p < 0.005$. $p$ values were adjusted for multiple comparisons using Holm's method. Details of the sample sizes and exact $p$ values for **(b)** are summarized in Supplementary Information. Boxplots show the median (horizontal lines), first to third quartile range (boxes), and 1.5× interquartile range from the box boundaries (whiskers).

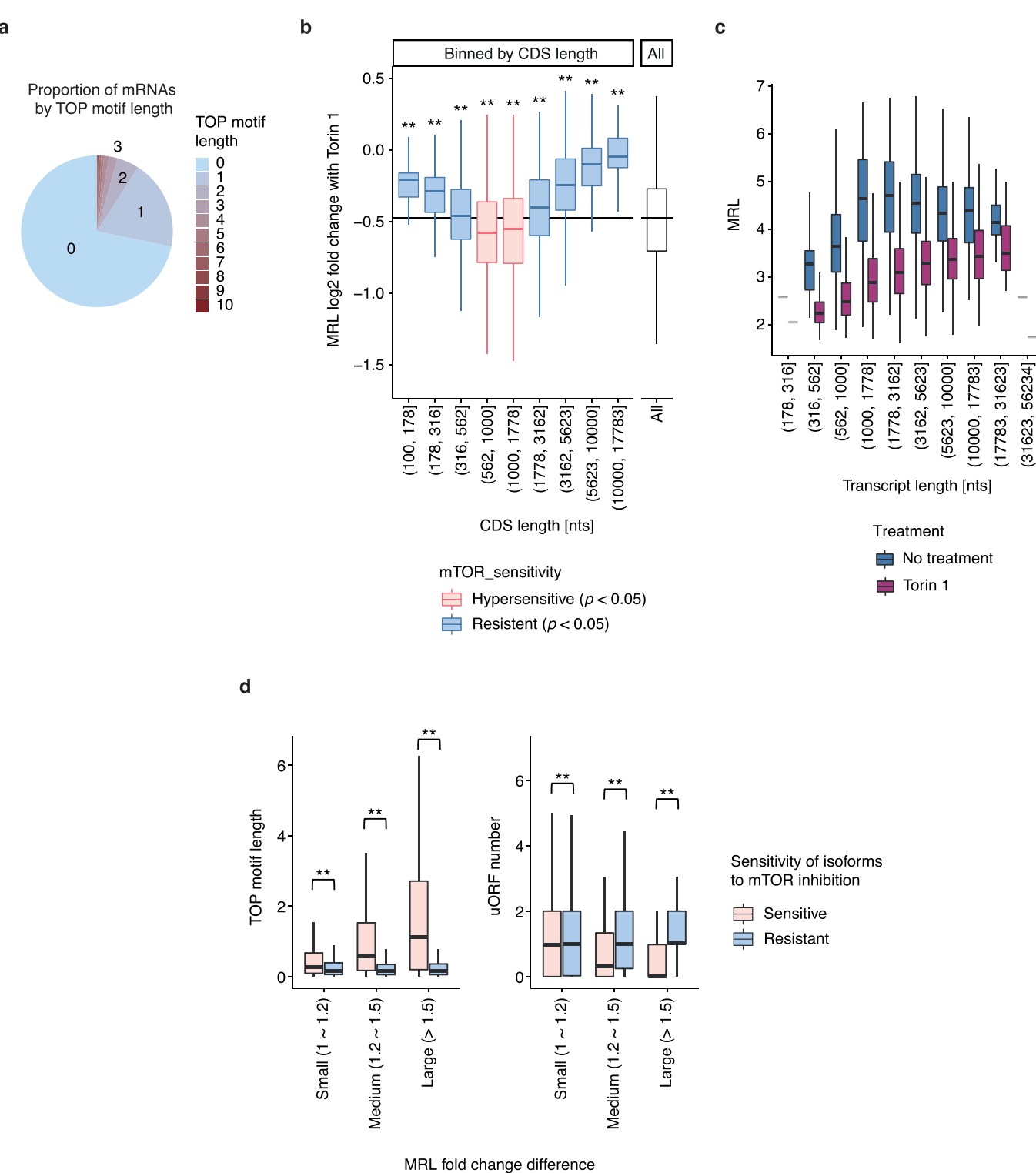

**Extended Data Fig. 5 | See next page for caption.**

**Extended Data Fig. 5 | HP5 refined mRNA features influencing the mTOR sensitivity of mRNAs. (a)** Proportion of mRNAs as a function of the TOP motif length (n = 9,589). **(b)** Boxplots showing changes in translational efficiency of mRNAs with Torin 1 (log2 fold change in mean ribosome load, MRL) as a function of CDS length. Responses of mRNAs with the indicated CDS length were compared against responses of all other mRNAs using the two-sided Mann-Whitney $U$ test; classes more downregulated or less downregulated compared to all other mRNAs (that is hypersensitive or resistant to mTOR inhibition, $p < 0.05$) are colored red or blue respectively. **(c)** Boxplots showing MRL as a function of transcript length, in the presence (purple) or absence (blue) of Torin 1. **(d)** Boxplots illustrating associations between mRNA features (length of TOP motif, left panel; number of uORFs, right panel) and sensitivity to mTOR inhibition, for alternate TSS mRNA isoforms of same gene. The mRNA isoforms are classified as sensitive or resistant based on their sensitivity to mTOR inhibition (the isoform with a larger or smaller mean ribosome load, MRL, log2 fold change with Torin 1, respectively). When more than two isoforms were expressed from the same gene, the isoforms with the largest and smallest MRL log2 fold change were selected for the analysis. The comparison was performed by the groups binned by their difference in MRL fold change of the two isoforms (x-axis). Distributions of the length of TOP motifs or the number of uORFs were compared using the two-sided Wilcoxon signed rank test. **(a-d)** Data are for RCC4 VHL cells. Boxplots show the median (horizontal lines), first to third quartile range (boxes), and 1.5× interquartile range from the box boundaries (whiskers). \* $p < 0.05$, \*\* $p < 0.005$. **(b and d)** $P$ values were adjusted for multiple comparisons using Holm's method. Details of the sample sizes and exact $p$ values for **(b-d)** are summarized in Supplementary Information.

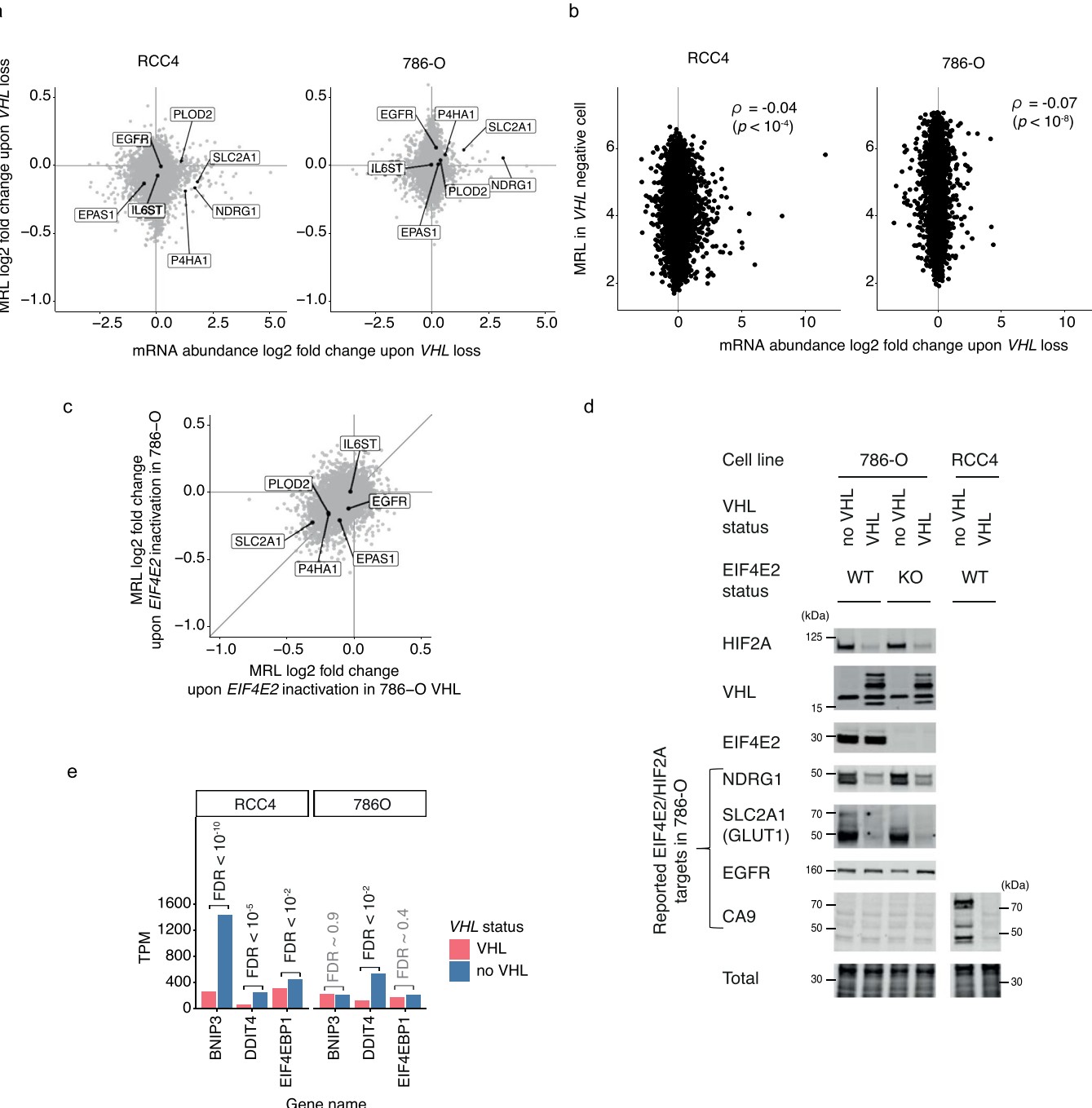

**Extended Data Fig. 6 | Effects of VHL on the efficiency of translation. (a)** Scatter plots showing the data from Fig. 3b with the genes reported to be translationally regulated by HIF2A–EIF4E2 labeled[9,10]. **(b)** Scatter plots comparing changes in mRNA abundance of genes upon *VHL* loss with the mean ribosome load (MRL) in RCC4 and 786-O cells. Spearman's rank-order correlation was used to assess the association (n = 9,493 and 8,065 for RCC4 and 786-O respectively). The absence of correlation indicates that genes induced by *VHL* loss were not preferentially translated upon induction compared to genes that are not induced by VHL. **(c)** Scatter plot comparing changes in MRL upon *EIF4E2* inactivation in 786-O cells with or without VHL (n = 7,753). **(d)** Immunoblotting analysis of four previously reported EIF4E2–HIF2A target genes in 786-O cells[9,10] as a function of VHL and EIF4E2. HIF2A induction did not alter EGFR protein abundance. *EIF4E2* inactivation did not alter NDRG1 and SLC2A1 protein abundance in the presence of HIF2A. Although CA9 was reported to be an EIF4E2–HIF2A target in 786-O cells[10], CA9 protein expression could not be detected in 786-O cells in agreement with previous studies showing that CA9 is transcriptionally induced by HIF1A but not by HIF2A[83]. Identical results were obtained using a second targeting gRNA for EIF4E2 in 786-O cells. **(e)** Analysis of changes in mRNA abundance of negative regulators of mTOR pathway upon *VHL* loss. mRNA abundance was measured as transcripts per million (TPM) and the significance of differential expression was assessed using the Wald test (n = 3 and 4 for each condition in RCC4 and 786-O respectively).

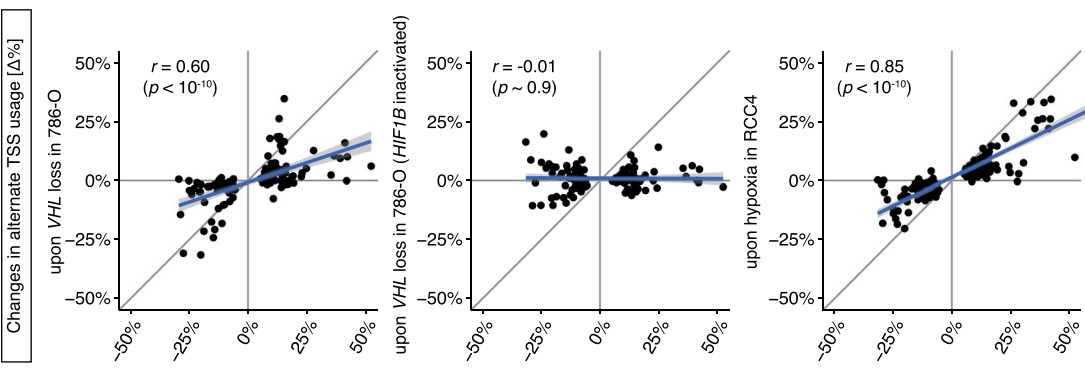

**Extended Data Fig. 7 | Alternate TSS usage in *VHL*-defective and hypoxic cells.** Correlations between VHL-dependent changes in alternative TSS usage in RCC4 (x-axis) and such changes in other cells or conditions (y-axes); panels show correlations with *VHL* loss in 786-O cells (left); with *VHL* loss in *HIF1B* inactivated 786-O cells (middle) and with hypoxia (1% O$_2$ for 24 h) in RCC4 VHL cells (right). Genes with too little mRNA expression for quantitative analysis were excluded from the analyses (see Methods). Pearson's product moment correlation coefficient was used to assess the associations (n = 124, 126, and 148 for the respective comparisons).

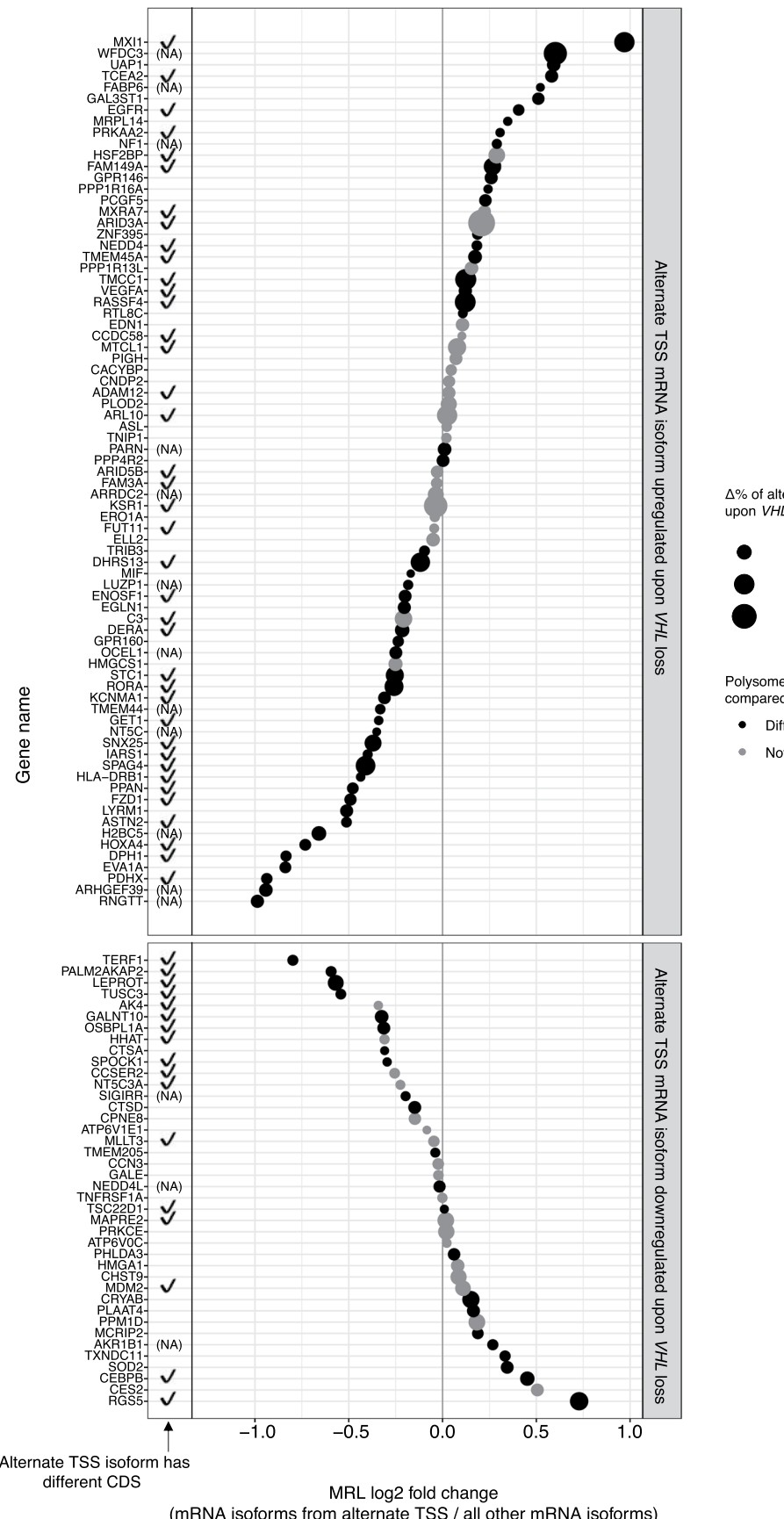

**Extended Data Fig. 8 | See next page for caption.**

**Extended Data Fig. 8 | VHL dependent alternate TSS usage generates mRNAs with an altered translational efficiency.** The plot shows the differences in translational efficiency (expressed as mean ribosome load, MRL) between mRNA isoforms that are generated from VHL-dependent alternative TSSs and all other isoforms transcribed from the same gene. Data are for RCC4 cells. Genes are sorted by log2 fold difference in MRL; significant differences in polysome distribution (FDR < 0.1) are indicated by black colouring. The magnitude of changes in alternative TSS usage is shown by the size of point. Genes whose alternate TSS isoform contains a different predicted CDS are indicated with a check mark; NA, CDS could not be predicted. Genes with too little alternate TSS isoform expression for MRL calculation were excluded from the analysis (see Methods).

a

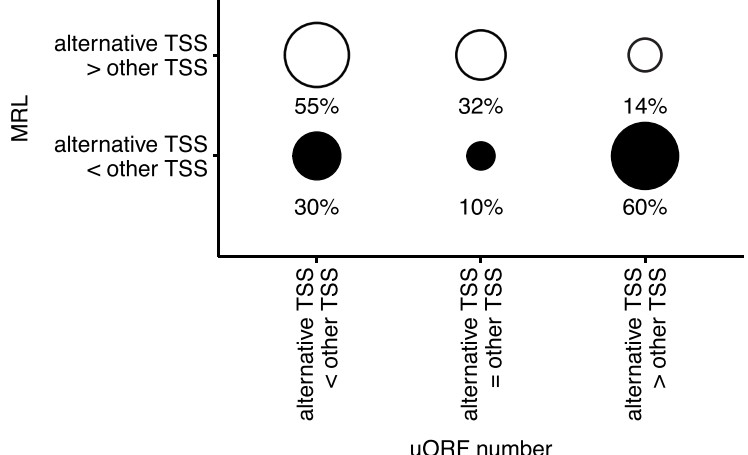

b

**Extended Data Fig. 9 | See next page for caption.**

**Extended Data Fig. 9 | Difference in translation of VHL-dependent isoform from that of other isoforms. (a)** Bubble chart showing the proportion of genes manifesting greater (white bubbles, n = 22) or lesser (black bubbles, n = 20) MRL on alterative versus other mRNA isoforms, grouped by the difference in the number of uORF between those isoforms (x-axis). The analysis was performed for 42 genes for which 5′ UTR sequences of the alternate TSS and the majority of other isoforms could be predicted, among the 75 genes whose VHL-dependent alternate TSS isoforms were translated differently to other isoforms of same gene. **(b)** Examples of the effect of VHL-dependent alternate TSS usage on translation. Bar charts (left panels) show the abundance of UDP-N-Acetylglucosamine pyrophosphorylase 1 (*UAP1*) and deoxyribose-phosphate aldolase (*DERA*) mRNA isoforms defined by TSS, estimated as transcript per million (TPM) from 5′ end-Seq data. Data presented are the mean of measurements for three independent clones of RCC4 and RCC4 VHL cells. Line charts (middle panels) show the proportion of each TSS-defined mRNA isoform of *UAP1* or *DERA* distributed across polysome fractions in RCC4 cells; the line indicates the mean value while the shaded area shows the standard deviation of the data from the three independent clones. The number of uORFs in the relevant 5′ UTR is indicated in the schematics (right panels). Note that *VHL* inactivation induced the TSS3 isoform of *UAP1* and *DERA* with less or more uORFs than other isoforms. In both cases, the isoform with less uORFs was better translated.

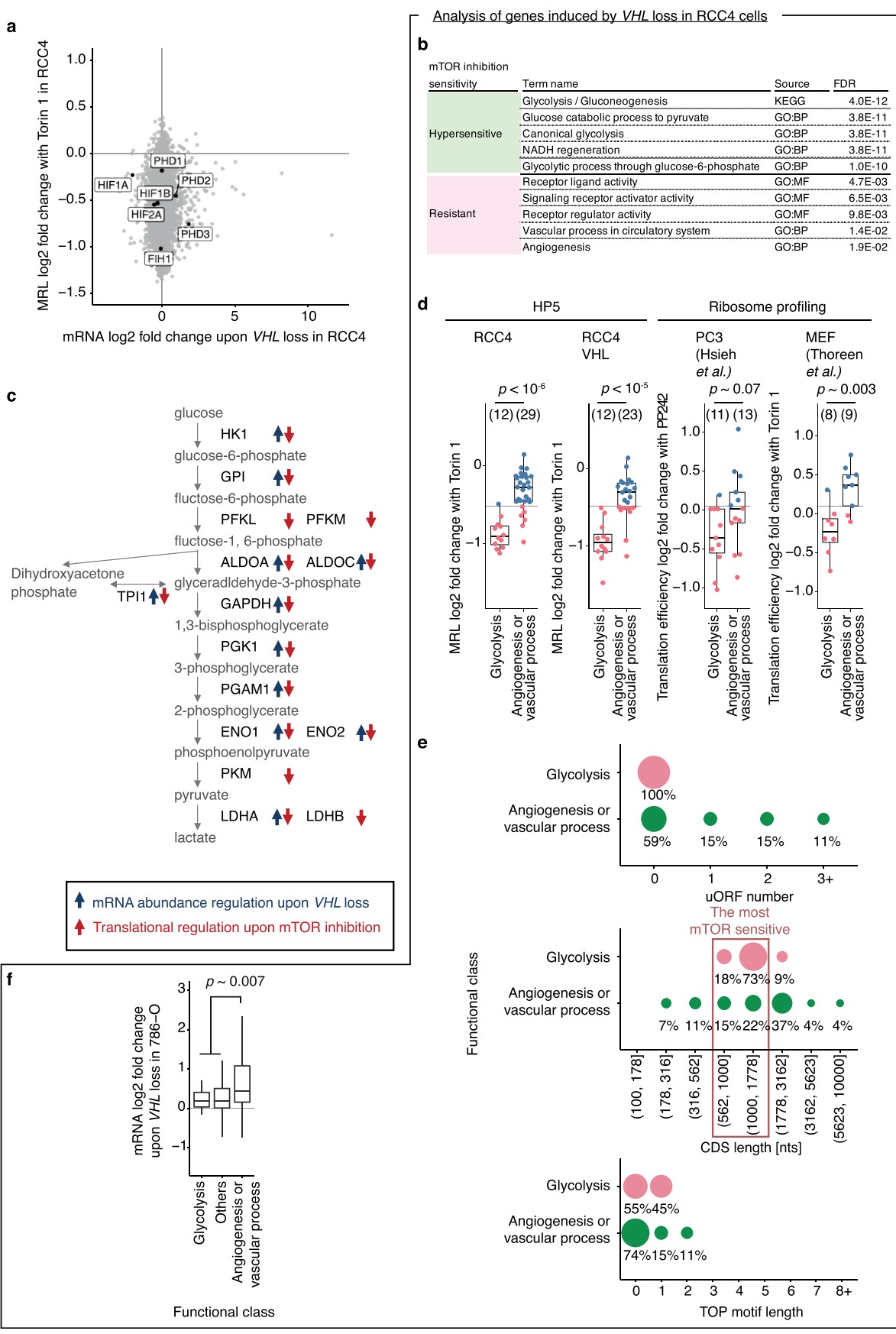

Extended Data Fig. 10 | See next page for caption.

**Extended Data Fig. 10 | Differential sensitivity to mTOR inhibition amongst functional groups of transcripts induced upon *VHL* loss. (a)** The figure is identical to Fig. 5b, but genes involved in the HIF signaling pathway are labelled. **(b)** Gene set enrichment analysis among genes induced upon *VHL* loss (FDR < 0.1 and mRNA fold change > 1.5) and either hypersensitive (green) or resistant (pink) to mTOR inhibition in RCC4 cells (see Methods for definition). The top 5 most enriched gene ontology or KEGG orthology terms are shown. n = 124 and 122 mTOR hypersensitive or resistant genes were considered for the analysis. **(c)** Schematic showing inverse changes in mRNA abundance upon *VHL* loss to changes in translational efficiency upon mTOR inhibition for genes encoding glycolytic enzymes in RCC4 cells. **(d)** Boxplots comparing changes in translational efficiency for the indicated classes of HIF-target gene in response to treatment with an mTOR inhibitor in this and previous published studies[14,21]. Data are expressed as log2 fold change in mean ribosome load (MRL) for the HP5 data and as translational efficiency for the ribosome profiling data. The distributions in each data set were compared using the two-sided Mann-Whitney *U* test; the number of genes in each class is indicated in parenthesis. The horizontal lines show the median changes in MRL or translational efficiency of all expressed genes. The colours indicate whether the changes are above or below the median value of all genes (blue and red respectively). PC3: human prostate cancer cells; MEF: mouse embryonic fibroblasts. **(e)** Proportion of genes with the indicated number of uORFs (top panel), CDS length (middle panel), and the length of TOP motif (bottom panel), amongst the specified functional class of HIF-target genes. For the plot of the CDS length, the two most mTOR sensitive groups are indicated by a red box. These analyses were performed for genes for which the 5′ UTR sequence could be predicted for the majority of expressed mRNAs (n = 11 and 27 for glycolysis and angiogenesis or vascular process genes respectively). **(f)** Boxplots showing changes in mRNA abundance of HIF-target genes upon *VHL* loss in 786-O cells (that is induction of HIF2A) as a function of the encoded protein class. The changes in mRNA abundance of angiogenesis or vascular process genes were compared against those of all other HIF-target genes using the two-sided Mann-Whitney *U* test (n = 27 and 305 respectively). Boxplots show the median (horizontal lines), first to third quartile range (boxes), and 1.5× interquartile range from the box boundaries (whiskers).

# Reporting Summary

Nature Research wishes to improve the reproducibility of the work that we publish. This form provides structure for consistency and transparency in reporting. For further information on Nature Research policies, see our Editorial Policies and the Editorial Policy Checklist.

## Statistics

For all statistical analyses, confirm that the following items are present in the figure legend, table legend, main text, or Methods section.

| n/a | Confirmed | |
|---|---|---|
| ☐ | ☒ | The exact sample size (*n*) for each experimental group/condition, given as a discrete number and unit of measurement |
| ☐ | ☒ | A statement on whether measurements were taken from distinct samples or whether the same sample was measured repeatedly |
| ☐ | ☒ | The statistical test(s) used AND whether they are one- or two-sided<br>*Only common tests should be described solely by name; describe more complex techniques in the Methods section.* |
| ☒ | ☐ | A description of all covariates tested |
| ☐ | ☒ | A description of any assumptions or corrections, such as tests of normality and adjustment for multiple comparisons |
| ☐ | ☒ | A full description of the statistical parameters including central tendency (e.g. means) or other basic estimates (e.g. regression coefficient) AND variation (e.g. standard deviation) or associated estimates of uncertainty (e.g. confidence intervals) |
| ☐ | ☒ | For null hypothesis testing, the test statistic (e.g. *F*, *t*, *r*) with confidence intervals, effect sizes, degrees of freedom and *P* value noted<br>*Give P values as exact values whenever suitable.* |
| ☒ | ☐ | For Bayesian analysis, information on the choice of priors and Markov chain Monte Carlo settings |
| ☒ | ☐ | For hierarchical and complex designs, identification of the appropriate level for tests and full reporting of outcomes |
| ☐ | ☒ | Estimates of effect sizes (e.g. Cohen's *d*, Pearson's *r*), indicating how they were calculated |

*Our web collection on statistics for biologists contains articles on many of the points above.*

## Software and code

Policy information about availability of computer code

| Data collection | High-throughput DNA sequencing: HiSeq 4000 (illumina); Density gradient fractionation system (Brandel, BR-188); Western blotting: Odyssey CLx system (LI-COR Biosciences); RT-qPCR: StepOnePlus Real-Time PCR System (Thermo Fisher Scientific, 4376600) |
|---|---|
| Data analysis | The computational pipeline that was used to analyze high-throughput DNA sequencing and RT-qPCR data is available on GitHub (https://github.com/YoichiroSugimoto/20211102_HP5_HIF_mTOR) and Zenodo (https://doi.org/10.5281/zenodo.6583247).<br><br>In the pipeline, the following software was used: R (4.0.0), data.table (1.12.8), dplyr (1.0.0), stringr (1.4.0), magrittr (1.5), ggplot2 (3.3.1), rcompanion (2.3.26), UMI-tools (1.0.1), Cutadapt (2.10), Bowtie2 (2.4.1), STAR (2.7.4a), paraclu (9), dpi (beta3), StringTie (2.1.2), ORFik (1.8.1), ViennaRNA Package (for RNALfold, 2.3.3), gprofiler2 (0.1.9), DESeq2 (1.28.0), DRIMSeq (1.16.0), apeglm (1.10.0), stageR (1.10.0), DEXSeq (1.34.0), mgcv (1.8-31), MACS2 (2.2.7.1), and DiffBind (2.16.0).<br><br>For RT-qPCR data collection, StepOne Software (2.3) was used. For western blotting data analysis, Image Studio (5.2) was used. |

For manuscripts utilizing custom algorithms or software that are central to the research but not yet described in published literature, software must be made available to editors and reviewers. We strongly encourage code deposition in a community repository (e.g. GitHub). See the Nature Research guidelines for submitting code & software for further information.

## Data

Policy information about availability of data

All manuscripts must include a data availability statement. This statement should provide the following information, where applicable:

- Accession codes, unique identifiers, or web links for publicly available datasets
- A list of figures that have associated raw data
- A description of any restrictions on data availability

The HP5 and 5' end-Seq of total mRNA data are available on ArrayExpress with the accession numbers E-MTAB-10689 and E-MTAB-10688 respectively. The Ct values for RT-qPCR analysis that were analyzed for Extended Data Fig. 3a are provided as Source data. mTOR hypersensitive genes identified by previous studies are provided as Source data. The ChIP-Seq data from Smythies et al. that were analyzed for Fig. 5d are available on GEO with the accession number GSE120885. The following reference data were used for this study; human genome: hg38, obtained via BSgenome.Hsapiens.UCSC.hg38 (1.4.3); human transcripts: RefSeq57 (GRCh38.p13) and GENCODE58 (GENCODE version 34: gencode.v34.annotation.gtf). Processed data files are provided as Supplementary Data and Source Data. The list of samples that were analyzed for this study is provided as Supplementary Data.

# Field-specific reporting

Please select the one below that is the best fit for your research. If you are not sure, read the appropriate sections before making your selection.

☒ Life sciences ☐ Behavioural & social sciences ☐ Ecological, evolutionary & environmental sciences

For a reference copy of the document with all sections, see nature.com/documents/nr-reporting-summary-flat.pdf

# Life sciences study design

All studies must disclose on these points even when the disclosure is negative.

| | |
|---|---|
| Sample size | No statistical method was employed to predetermine sample size. The sample sizes (numbers of repetitions) were determined based on previous publications that used similar methodologies. |
| Data exclusions | Samples that failed to produce high-throughput DNA sequencing libraries within a multiplexed reaction (defined as < 25% of median number of reads) were excluded. Although this criterion was not pre-established, such samples were clearly outlying, implying technical failure. Other data exclusions are described in the manuscript. |
| Replication | The reproducibility of findings was assessed through biological replicate experiments. Only individual experiments using different clones derived from same cell line were treated as biological replicates. High-throughput DNA sequencing and western blotting experiments were performed with 2-4 biological replicates as reported in the figure legends or Supplementary Note. RT-qPCR was performed with 2 technical replicates since the purpose of the experiments was to show that the HP5 method has an equivalent quantitative power to RT-qPCR, and not to test a biological hypothesis. All attempts at replication were successful. |
| Randomization | Randomization is not relevant to this study because the samples were not allocated into separate experimental groups. |
| Blinding | Investigator was not blinded to group allocation. However, all the samples were processed and analyzed in the same/parallel manner, and the analyses were internally controlled (i.e. not subject to the bias of the investigator). |

# Reporting for specific materials, systems and methods

We require information from authors about some types of materials, experimental systems and methods used in many studies. Here, indicate whether each material, system or method listed is relevant to your study. If you are not sure if a list item applies to your research, read the appropriate section before selecting a response.

### Materials & experimental systems

| n/a | Involved in the study |
|---|---|
| ☐ | ☒ Antibodies |
| ☐ | ☒ Eukaryotic cell lines |
| ☒ | ☐ Palaeontology and archaeology |
| ☒ | ☐ Animals and other organisms |
| ☒ | ☐ Human research participants |
| ☒ | ☐ Clinical data |
| ☒ | ☐ Dual use research of concern |

### Methods

| n/a | Involved in the study |
|---|---|
| ☐ | ☒ ChIP-seq |
| ☒ | ☐ Flow cytometry |
| ☒ | ☐ MRI-based neuroimaging |

## Antibodies

| | |
|---|---|
| Antibodies used | (Primary antibodies; used at 1/1,000 dilution) anti-VHL (Santa Cruz Biotechnology, sc-135657, clone: VHL40), anti-HIF1A (BD |

| Antibodies used | Biosciences, 610959, clone: 54), anti-HIF2A (Cell Signaling Technology, #7096, clone: D9E3), anti-HIF1B (Cell Signaling Technology, #5537, clone: D28F3), anti-EIF4E2 (Proteintech, 12227-1-AP, polyclonal), anti-NDRG1 (Cell Signaling Technology, #9485, clone: D8G9), anti-SLC2A1 (Cell Signaling Technology, #12939, clone: D3J3A), anti-EGFR (Santa Cruz Biotechnology, sc-373746, clone: A-10), and anti-CA9 (Cell Signaling Technology, #5649, clone: D47G3); (secondary antibodies; used at 1/15,000 dilution) anti-mouse IgG DyLight 800 (Cell Signaling Technology, #5257) anti-mouse IgG IRDye 680RD (LI-COR Biosciences, 925-68072), and anti-Rabbit IgG IRDye 800CW (LI-COR Biosciences, 926-32213). |
|---|---|
| Validation | Primary antibodies that were used for this study are supported for use in immunoblotting of human proteins as follows.

Antibodies supplied by Cell Signaling Technology (anti-HIF2A, anti-HIF1B, anti-NDRG1, anti-SLC2A1, and anti-CA9 antibody) were validated for the recommended applications by the company as follows: (https://www.cellsignal.co.uk/about-us/cst-antibody-validation-principles). Anti-HIF1A antibody was validated for the immunoblotting of the human HIF1A according to the manufacturer (BD Biosciences). Anti-EGFR antibody was validated for the immunoblotting of human EGFR using a siRNA targeting EGFR in A-431 cells according to the manufacturer's website (Santa Cruz Biotechnology).

In addition, anti-VHL, anti-HIF1A, and anti-HIF2A antibody were validated by the comparison of human VHL-defective kidney cancer cells (RCC4 and 786-O cells) against cell lines with human VHL reintroduction, using immunoblotting. The validation experiments were performed using established cell lines produced in the laboratory, and consistent results were obtained for the cell lines generated in this study (Extended Data Fig. 2).

The antibodies for well established HIF target genes (NDRG1 and SLC2A1) were validated by comparing human VHL-defective kidney cancer cells (RCC4 and 786-O cells) with those with VHL reintroduced, using immunoblotting (Extended Data Fig. 6d).

Anti-EIF4E2 antibody was validated by comparing multiple clones of 786-O cells with intact EIF4E2 with those with CRISPR/Cas9 mediated inactivation of EIF4E2 (which was independently confirmed by the sequencing of the target genomic region.) |

# Eukaryotic cell lines

Policy information about cell lines

| Cell line source(s) | RCC4 and 786-O cells were obtained from the Cell Services at the Francis Crick Institute. |
|---|---|
| Authentication | RCC4 and 786-O cells were authenticated by STR profiling by the Cell Services at the Francis Crick Institute. |
| Mycoplasma contamination | Cells were confirmed to be free from mycoplasma contamination by the Cell Services at the Francis Crick Institute. |
| Commonly misidentified lines (See ICLAC register) | Both RCC4 and 786-O cells used for this study were not included in the list. |

# ChIP-seq

## Data deposition

☒ Confirm that both raw and final processed data have been deposited in a public database such as GEO.

☒ Confirm that you have deposited or provided access to graph files (e.g. BED files) for the called peaks.

| Data access links *May remain private before publication.* | Published ChIP-Seq data (GEO accession ID of GSE120885) were re-analyzed and the experimental procedure is described in the original article (Smythies et al., EMBO Rep (2019)20:e46401). The reprocessed peak data are available on GitHub (https://github.com/YoichiroSugimoto/20211102_HP5_HIF_mTOR). |
|---|---|
| Files in database submission | Raw data: GSM3417826 RCC4_Normoxia_HIF-1a (PM14)_Rep 1 GSM3417827 RCC4_Normoxia_HIF-1a (PM14)_Rep 2 GSM3417828 RCC4_Normoxia_HIF-2a (PM9)_Rep 1 GSM3417829 RCC4_Normoxia_HIF-2a (PM9)_Rep 2 GSM3417830 RCC4_Normoxia_Input_Rep 1 GSM3417831 RCC4_Normoxia_Input_Rep 2

Processed data: HIF-binding-site.bed |
| Genome browser session (e.g. UCSC) | The peak data in the bed format are available on GitHub (https://github.com/YoichiroSugimoto/20211102_HP5_HIF_mTOR). |

## Methodology

| Replicates | Duplicate experiments were performed for HIF1A, HIF2A, and input ChIP-Seq experiments (Smythies et al., EMBO Rep (2019)20:e46401) |
|---|---|
| Sequencing depth | All the sequence data are paired-end with the read length of 75 bases. The followings are the sequence depth: RCC4 Normoxia HIF1A Rep 1: (total) 27493998 (uniquely aligned) 21955115 RCC4 Normoxia HIF1A Rep 2: (total) 27838028 (uniquely aligned) 22291592 RCC4 Normoxia HIF2A Rep 1: (total) 29535774 (uniquely aligned) 23752981 |

RCC4 Normoxia HIF2A Rep 2: (total) 29807598 (uniquely aligned) 23972781
RCC4 Normoxia input Rep 1: (total) 38705153 (uniquely aligned) 31431115
RCC4 Normoxia input Rep 2: (total) 42199628 (uniquely aligned) 34381727

Antibodies

Anti-HIF1A rabbit polyclonal, PM14 and anti-HIF-2α rabbit polyclonal, PM9. (Smythies et al., EMBO Rep (2019)20:e46401). The procedure to raise these two antibodies is described by Lau et al. (Br J Cancer. 2007 Apr 23; 96(8): 1284–1292.).

Peak calling parameters

The intersection of ChIP-Seq peaks identified by the ENCODE ChIP-Seq pipeline and MACS2 software was used. The ENCODE ChIP-Seq pipeline was used with the default parameters while the MACS2 was used with the following parameters (-q 0.1 --call-summits)

Data quality

The high quality of the data was demonstrated in the original study by Smythies et al. (EMBO Rep (2019)20:e46401).

Software

The Bowtie2, ENCODE ChIP-Seq pipeline (https://github.com/ENCODE-DCC/chip-seq-pipeline2) and MACS2 software. The pipeline to process the ChIP-Seq data is available on GitHub (https://github.com/YoichiroSugimoto/20211102_HP5_HIF_mTOR).

