## [Peer Review File · Nature Structural & Molecular Biology]

Peer Review Information

Journal: Nature Structural and Molecular Biology

Manuscript Title: Isoform-resolved mRNA profiling of ribosome load defines interplay of HIF and mTOR dysregulation in kidney cancer

Corresponding author name(s): Professor Peter Ratcliffe

Editorial Notes:

Reviewer Comments & Decisions:

Decision Letter, initial version:
--

Dear Dr. Ratcliffe,

Thank you for submitting your manuscript "Isoform resolved measurements of absolute translational efficiency define interplay of HIF and mTOR dysregulation in kidney cancer". I sincerely apologize for the delay in processing your manuscript, which resulted from difficulties in obtaining referees' reports. We have received comments from the 3 reviewers who have evaluated your manuscript are below. Unfortunately, after carefully considering their comments, we cannot offer to publish your manuscript in Nature Structural & Molecular Biology.

You will see that while the referees find the work of potentially interesting, they raise significant

concerns both regarding the need to benchmark your method to other previously developed methods and provide more data to support that it can determine 'absolute translation efficiency (referees #1 and #3), and also concerns that the conclusions regarding HIF2 have to be reproduced in other cell lines to be fully convincing (all three referees). Together, these cast doubt on the strength of the novel conclusions that can be drawn at this stage.

However, if further experimentation, analysis, and revisions allow you to address the referees concerns in full, we would be prepared to consider an appeal of our decision, on the condition that no related work is published in the interim or has been accepted in our journal. Please contact me to discuss an appeal and potential revision. Please note that, until we have the opportunity to read the revised manuscript in its entirety, we cannot promise that it will be sent back for peer review.

I am sorry we could not be more positive on this occasion, especially after such a long revision. I hope that you find the referees' comments useful in deciding how best to proceed.

Sincerely,

Carolina

Carolina Perdigoto, PhD
Chief Editor
Nature Structural & Molecular Biology
orcid.org/0000-0002-5783-7106

Referee expertise:

Referee #1: HIF and mTOR signalling, hypoxia response in cancer

Referee #2: HIF and mTOR signalling, hypoxia response in cancer

Referee #3: translation, ribosome profiling

Reviewers' Comments:

Reviewer #1:

Remarks to the Author:

This manuscript describes a method to quantify individual mRNA translation efficiency on a global level. The authors apply this method to the situations of mTOR inhibition (Torin) and +/-VHL. Some interesting findings emerge from this exercise. First of all, mTOR appears to negatively affect the translation efficiency of the vast majority of transcripts, but enhance the translation of a subset of transcription factors and components of the ubiquitination machinery (Fig.2c). In comparison, and as expected, VHL has minimal overall effect on translation efficiencies (Fig.3). Instead, VHL (and by

inference HIF) regulate the transcript levels of some specific mRNA isoforms that are well translated, such as an isoform of MXI1 (Fig.4). HIF targets, like the complete battery of mRNAs, experience variable sensitivities to Torin (Fig.5).

In essence, none of these discoveries may be particularly surprising, but rather the paper provides valuable datasets and analysis to support current understandings of translational regulations (e.g. importance of TOP sequences for regulation by mTOR), and various implications that the group and others' could build upon.

Authors claim that an important result is that translational control by mTOR is greater than previously appreciated. It would strengthen the paper to provide specific comparisons to other studies to substantiate this claim, and to justify why the current approach allowed resolution of this effect.

Perhaps most thought-provoking is the finding that HIF probably has little effect on translation despite previous papers demonstrating a specific role for HIF2 in translation. As authors acknowledge, the dataset does not rule out general regulation of a smaller magnitude. Importantly however, the paper does not address the specific translational regulation of mRNAs previously show to be regulated by HIF2. It is therefore unclear if this regulation could not be reproduced, or whether these specific transcripts are simply not expressed in the analyzed cells. This is a very important issue that could and should be addressed with the existing dataset.

In addition to these issues pertaining to the major conclusions, I have several concerns that need to be addressed:

1) The introduction could benefit from motivating why there is a need for this new method, and should reference similar approaches applied by others. Authors do refer to ribosome protection assays. However, an important body of work previously describing in detail and applying very similar methods of calculating mean ribosome occupancy by polysome fractionation (e.g. PMID 17998058, 16098621, 16467844) are not referenced.

2) An important limitation of the approach appears to be the lack of quantification of mRNAs with 0 or >8 ribosomes. Authors need to address this:

a) The total mRNA abundance needs to be accounted for. With the current approach, the assigned MRL of mRNAs may be substantially skewed due to the omission of "0"s. Authors argue in the discussion that normalizing by total mRNA introduces variability, which is probably true. However, ignoring this issue could substantially affect conclusions. Some mRNAs may have poor general translation initiation efficiency, but effective re-initiation due to mRNA circularization. This results in two mRNA pools: One associated with ribosomes that could be efficiently translated, and one sequestered from ribosomes that is not translated at all. The relative abundance in these pools could be affected by mTOR and other translation regulators. In fact, others have previously employed methods to quantify the fraction of mRNAs associated with ribosomes, in addition to the average MRL of ribosome-bearing mRNAs (e.g. PMID 16467844).

b) As figure 1a indicates, resolution of the polysome peaks are generally lost after at least 8. Is material lower in the gradient discarded or pooled with the 8r fraction? Do most individual mRNAs have a ~normal distribution across measurable values (i.e. 1-8) or are there mRNAs that cluster heavily towards 8+, indicating that their actual ribosome load is higher?

3) It is a weakness that none of the major conclusions are validated by probing translation efficiency

of individual transcripts via independent methods such as radiolabelling and immune precipitation normalized by mRNA abundance. For example, demonstrating translational regulation of previously unknown targets in the 'transcription', 'ubiquitination', 'glycolysis' and 'angiogenesis' groups would be of great value and provide confidence in the approach and analysis.

4) All effects of VHL presence/absence are ascribed to HIF. This is clearly inappropriate as VHL has other targets. Only once do investigators experimentally attribute an effect to HIF by knocking down HIF1b. This issue needs to be addressed in the way data are interpreted throughout, and the limitation acknowledged.

Minor:

1) Similarly, there is no data to corroborate that effects of Torin are attributable only to mTOR. This is less of an issue, but at least some references to the specificity of the inhibitor and potential off-target effects needs to be included.

2) Analysis performed in Fig4a and resulting conclusions are unclear and should be better explained. What is the purpose of 'omitting' certain parameters and what does 'simulated MRL' mean?

Reviewer #2:

Remarks to the Author:

The manuscript by Sugimoto and Ratcliffe entitled "Isoform resolved measurements of absolute translational efficiency define interplay of HIF and mTOR dysregulation in kidney cancer" describes a comprehensive effort to identify the relationship of transcriptional reprogramming by HIF to translational reprogramming by mTOR in the setting of kidney cancer. The authors describe a new workflow for measurement of absolute translational efficiency of mRNAs resolved by their transcription start sites. They determined a remarkable translational reprogramming by mTOR, especially with metabolic enzymes and pathways. In contrast global effects of HIF on translation are limited and not observed at all by HIF-2 α . Instead, these are mediated by HIF-1 α in a subset of genes. A specific class of HIF-1 α transcriptional target genes exhibit sensitivity to mTOR effects indicating that inhibitors of HIF-2 α and mTOR could be efficacious in the treatment of kidney cancer. Overall, the data is extremely convincing and the manuscript well written. I strongly recommend acceptance upon the resolution of the following concerns.

1. The western blot pattern describing pVHL protein abundance in Figure S2 reveals a complex pattern of bands. Can the authors explain why the species are so numerous? Is this due to post translational modifications, or distinct pVHL isoforms, etc.?

2. Based on comparing results in RCC4 cells (expressing both HIF-1 α and HIF-2 α) to 786-O cells (expressing HIF-2 α only), the authors conclude that HIF-2 α has no real effect on the translational efficiency of kidney cancer mRNAs. However, to make this claim the authors really need to remove HIF-1 α from RCC4 cells genetically or add HIF-1 α to 786-O cells so that isogenic cell lines are compared.

3. Minor points: on page 9 "particular striking" should read "particularly striking".

4. The authors call into question previous findings by Uniacke et al. (Nature 2012) and provide some explanations for the discrepancy in their findings. This should be elaborated on to improve readability of the manuscript.

5. Finally, the first paper to demonstrate hypoxic influences on mTOR was by Arsham et al. (JBC 2003). This paper convincingly demonstrated that the influence of hypoxia on mTOR is rapid and independent of the HIF-1 α /REDD 1 pathway described by Brugarolas et al in Genes and Development. The paper should be cited.

Reviewer #3:

Remarks to the Author:

The manuscript "Isoform resolved measurements of absolute translational efficiency define interplay of HIF and mTOR dysregulation in kidney cancer" is devoted to investigation of MTOR and HIF dependent changes in gene expression for transcripts with alternative transcription start sites in kidney cancer cells. The authors developed a clever modification of polysome profiling in order to define translation efficiencies of mRNA isoforms that differ in their 5'ends. Using this approach, the authors demonstrated that in contrast to MTOR, HIF signaling does not directly affect translation of mRNAs as it was reported before. Instead, HIF activation results in alternative transcription start site usage for a number of genes, which results in alterations in 5'leaders of mRNAs and in differential translation. Finally, the authors found that HIF1A and HIF2A induced mRNAs have differential sensitivity to mTOR inhibition.

To my opinion, this is definitely an interesting and important study; however, I have several comments regarding the method and clarity of presentation and discussion of results. Also, I have a number of suggestions related to additional analysis of already available data which may result in new interesting findings.

1. Regarding the methodology - first of all, while the method which the authors called HP5 is definitely useful, I have a feeling that the authors seriously overestimate their approach, as it certainly can't be applied for estimation of "absolute translation efficiency". In addition, this approach is not really new - there are several important manuscripts related to isoform specific translation which currently are not properly discussed.

Most importantly, this method does not discriminate which open reading frame is translated in particular mRNA. A simple example - for mRNA with one uORF and median ribosome load 3 it is equally possible that all 3 ribosomes occupy either uORF or CDS. Given that nearly half of mammalian mRNAs possess at least one uORF, one can't reliably predict translation efficiency from mRNA position in the gradient. For instance, well known ATF4 mRNA possess several regulatory uORFs, and according to riboseq data, under normal conditions most ribosomes translate uORFs but not ATF4 CDS.

In addition, the authors ignored alternative splicing (especially important - in 5'leaders) and alternative 3'UTRs utilization - in this case mRNAs with the same 5'end may have different translation efficiencies. Such cases can't be detected as the method relies only on sequencing of mRNA 5'ends. Finally, ribosome occupancy does not always correlate with translation efficiency, given that local elongation rate may vary for different mRNAs. A good example can be found in a recent manuscript (PMID: 32589965). It was demonstrated that depletion of one of eIF3 subunits resulted in increased ribosome occupancy on certain mRNAs, but these mRNAs were not activated but actually repressed at translation level.

To conclude, the method is useful for estimation of translation efficiency of mRNA isoforms, but can't provide absolute translation efficiency. The authors must discuss the limitations of their approach and,

where necessary, implement alternative methods to validate HP5 results (e.g. – RT-PCR of specific mRNA isoforms with primers specific to unique parts of alternative 5'leader in polysome gradient fractions).

2. There is one concern related to the experimental approach: the authors used oligo-dT primer for reverse transcription - this means that for long mRNAs there are less chances that reverse transcriptase will reach its 5'end. Moreover, in case of limited mRNA degradation, which is impossible to completely avoid during sample preparation, there are more chances for long mRNAs to be cleaved at internal positions, which will prevent detection of 5'end. How do the authors take this into account? Can it be that this potential artefact can account for unusual dependence of MRL on CDS length (Fig 1 D)? Perhaps utilization of a random reverse primer instead of oligo-dT for reverse transcription may be a good alternative.

3. As I mentioned before, the method is not entirely new. There are several studies about mRNA isoform specific translation, which are very similar in methodology (e.g. Wang et al. Pervasive isoform-specific translational regulation via alternative transcription start sites in mammals PMID: 27430939; Floor et al "Tunable protein synthesis by transcript isoforms in human cells" PMID: 26735365; Tamarkin-Ben-Harush et al. "Cap-proximal nucleotides via differential eIF4E binding and alternative promoter usage mediate translational response to energy stress", PMID: 28177284; Gandin et al., "nanoCAGE reveals 5' UTR features that define specific modes of translation of functionally related MTOR-sensitive mRNAs", PMID: 26984228) – and these papers are not properly discussed and cited here. The last one is especially relevant, as it is devoted to the same issue - interplay between TSS usage and response to mTOR inhibition at translational level.

4. Then studying response to mTOR inhibition, the authors found a lot of hypersensitive mRNAs. It would be interesting to compare these results with results of other studies performed with riboseq and polysome profiling. What mRNAs are commonly downregulated at translation level, except 5'TOP mRNAs, and what is new? It is known that 5'TOP-containing mRNAs are not the only class of mRNAs that are "hypersensitive" to mTOR at translation level (see Gandin et al. – they really carefully examined the impact of 5'leader variants on MTOR mediated translation repression using the nanoCAGE technique).

5. It may be interesting to show how translation of mRNAs encoding proteins directly implicated in HIF signaling pathway (VHL, HIF subunits, HIF-prolyl hydroxylases etc) respond to mTOR inhibition itself?

6. Are there any mRNAs with alternative TSS that are differentially regulated by mTOR? If there are such cases – it will be very interesting to discuss it.

7. The finding that HIF2A does not regulate translation is very important. In addition to western blots (supplementary fig 5b) I suggest to show the MRL values for previously reported "HIF2A/eIF4E2 axis - regulated mRNAs" in cells with and without HIF2A and eIF4E2– this would strengthen conclusions and convince readers that HIF2A is not translation factor.

8. The part "HIF promotes alternate TSS usage to regulate translation" is written in such a way that it is really hard to follow and understand. In my opinion, it is necessary to rewrite this part in order to make it clear what exactly was found. HIF alters TSS usage, and a number of mRNAs with alternative TISs are differentially translated – is it correct? What is this number? MXI1 is a really good example, and I suggest adding more examples (in the main text or in the supplementary). It is interesting to see whether it is common that differentially translated mRNA variants differ in uORFs? Are there any cases where mRNA variant with longer 5'leader is translated better than that with shorter 5'leader? There may be particularly interesting explanation that long 5'UTR variants can bear some cis-acting element which stimulates translation.

9. Given the potentially high importance of HIF-dependent MXI1 transcription regulation, it is important to mention that HIF-mediated switch of TSS usage not only alters 5'leaders but also CDSs – three different transcripts encode different MXI1 proteoforms. Is anything known about functional

interplay of these proteoforms? In general, it may be interesting to highlight those genes where alternative TSS result in changes of CDS, similar to the MXI1 case.

10. Are there any specific features (e.g. length, secondary structures, uORFs) or sequence motifs in 5'leaders of mRNAs from "glycolysis" and "angiogenesis or vascular process" subsets which may help to explain differential sensitivity of translation to mTOR inhibition?

11. I suggest generating a supplementary table which will include all HIF dependent transcripts and effects of MTOR inhibition – currently it contains only those that are related to glycolysis and angiogenesis.

Author Rebuttal to Initial comments

Referees' Comments:

Referee #1:

Remarks to the Author: This manuscript describes a method to quantify individual mRNA translation efficiency on a global level. The authors apply this method to the situations of mTOR inhibition (Torin) and +/-VHL. Some interesting findings emerge from this exercise. First of all, mTOR appears to negatively affect the translation efficiency of the vast majority of transcripts, but enhance the translation of a subset of transcription factors and components of the ubiquitination machinery (Fig.2c). In comparison, and as expected, VHL has minimal overall effect on translation efficiencies (Fig.3). Instead, VHL (and by inference HIF) regulate the transcript levels of some specific mRNA isoforms that are well translated, such as an isoform of MXI1 (Fig.4). HIF targets, like the complete battery of mRNAs, experience variable sensitivities to Torin (Fig.5).

In essence, none of these discoveries may be particularly surprising, but rather the paper provides valuable datasets and analysis to support current understandings of translational regulations (e.g. importance of TOP sequences for regulation by mTOR), and various implications that the group and others' could build upon. Authors claim that an important result is that translational control by mTOR is greater than previously appreciated. It would strengthen the paper to provide specific comparisons to other studies to substantiate this claim, and to justify why the current approach allowed resolution of this effect.

As suggested we have expanded the analysis described in Fig. 2c of the original manuscript to provide direct comparisons of our data with those in four highly cited studies that reported mTOR hypersensitive genes: Hsieh *et al.* (PMID: 22367541; deploying ribosome profiling), Thoreen *et al.* (PMID: 22552098; deploying ribosome profiling), Larsson *et al.* (PMID: 22611195; deploying polysome profiling), and Morita *et al.* (PMID: 24206664; a subset of targets identified by Larsson *et al.*).

Our analyses first revealed that the mTOR hypersensitive genes identified by these studies are not entirely overlapping, overlap being small between studies using ribosome profiling and those using polysome profiling (Supplementary Fig. 4a).

We then examined the extent to which HP5 could identify each of these groups of reported mTOR targets, and compared its sensitivity with that of the two ribosome profiling studies (Supplementary Fig. 4b). This figure (reproduced below) directly compares the changes in translation (y-axes) as measured by HP5 (left panel) or by each of the two large published ribosomal profiling studies (centre and right panels) for those genes reported as being significantly downregulated in the study indicated below each boxplot. These analyses showed that HP5 could identify translational downregulation of mTOR targets identified by all of the four studies (left panel). By contrast, ribosome profiling in either published study was less powerful at identifying translational downregulation of mTOR targets that had been identified using polysome profiling (centre and right panels).

We believe that the following improvements contributed to the greater sensitivity of HP5. First, HP5 produces a higher proportion of sequence reads which can be uniquely mapped to mRNAs than ribosome profiling; the libraries of ribosome profiling typically contain a large proportion of contaminated reads from rRNAs (PMID: 32503920) and consist of much shorter sequencing reads which are more challenging to map uniquely. Second, HP5 provides greater resolution of the polysome profile, whilst providing higher sequence depth for each fraction than previous polysome profiling-based studies. Finally, HP5 uses RNA standards to accurately calculate mean ribosome load of mRNAs. We have included these analyses and a brief discussion thereof in the revised manuscript in Supplementary Fig. 4a and b and Supplementary Notes 1.

Supplementary Fig. 4

(a) Venn diagram showing the numbers and overlap of mTOR hypersensitive genes identified by previous studies. **(b)** Box plots showing changes in translation upon mTOR inhibition as measured by the indicated study (HP5, left-hand panel; ribosome profiling, centre and right-hand panels) for the genes identified as mTOR hypersensitive in each of the previous studies. In each panel, the left-hand boxplot shows the changes in MRL or translational efficiency of all expressed genes in that study; horizontal line, median value. Responses of mTOR hypersensitive genes identified by the indicated study were compared against responses for all expressed genes using the Mann-Whitney U test. *: $p < 0.05$, **: $p < 0.005$. p values were adjusted for multiple comparisons using Holm's method. Note that HP5 identified mTOR responsive genes reported in all four studies, whereas ribosome profiling was less sensitive in the identification of mTOR hypersensitive genes defined by the studies using polysome profiling.

Perhaps most thought-provoking is the finding that HIF probably has little effect on translation despite previous papers demonstrating a specific role for HIF2 in translation. As authors acknowledge, the dataset does not rule out general regulation of a smaller magnitude. Importantly however, the paper does not address the specific translational regulation of mRNAs previously show to be regulated by HIF2. It is therefore unclear if this regulation could not be reproduced, or whether these specific transcripts are simply not expressed in the analyzed cells. This is a very important issue that could and should be addressed with the existing dataset.

The referee asks that we show data on specific transcripts previously reported to be translationally regulated by HIF2A. The referee is correct that this data is present within the existing datasets, and we did in fact provide immunoblots illustrating the absence of HIF2A- and/or EIF4E2-dependency of a number of these reported HIF2A/EIF4E2 target genes (Uniacke *et al.* PMID: 22678294 and Ho *et al.* PMID: 26854219). This data was illustrated in Supplementary Fig. 5b of the original manuscript (new Supplementary Fig. 6d). This analysis was in fact a direct attempt to reproduce Fig. 4J of PMID: 26854219 and Supplementary Fig. 25a of PMID: 22678294.

In addition, in the revised manuscript, we have highlighted the proposed HIF2A/EIF4E2 target genes in new Supplementary Fig. 6a. This data is reproduced in the figure below (in which the relevant genes have been annotated on Fig 3b of the original manuscript). It can be seen that there is little, or no, translational regulation of any of these reported targets despite major changes in HIF2A upon *VHL*-inactivation in each of the cell lines (Supplementary Fig. 2).

To further support this, we have expanded the analysis shown in Fig 3c of the original manuscript (Supplementary Fig. 6c). This figure shows that any effects of *EIF4E2* inactivation are small and observed similarly in both *VHL*-defective cells (high levels of HIF2A) and *VHL*-complemented cells (low levels of HIF2A) – and therefore not HIF2A-dependent.

In summary, the analyses indicate that the reported regulation of specific transcripts by HIF2A/EIF4E2 (Uniacke *et al.* PMID: 22678294 and Ho *et al.* PMID: 26854219) could not be reproduced, at least under the conditions of our experiments.

We have elaborated and clarified our position on this point and included these additional analyses in the manuscript.

Supplementary Fig. 6a and c

(a) Scatter plots showing the data from Fig. 3b in the original manuscript, with genes reportedly translationally regulated by HIF2A/EIF4E2 now labelled. Note that the reported HIF2A/EIF4E2 targets show essentially no change in mean ribosome load (MRL) upon *VHL* loss which greatly increases HIF2A level in each cell type. **(c)** A scatter plot comparing changes in MRL upon *EIF4E2* inactivation in 786-O cells with or without *VHL*.

In addition to these issues pertaining to the major conclusions, I have several concerns that need to be addressed:

1) The introduction could benefit from motivating why there is a need for this new method, and should reference similar approaches applied by others. Authors do refer to ribosome protection assays. However, an important body of work previously describing in detail and applying very similar methods of calculating mean ribosome occupancy by polysome fractionation (e.g. PMID 17998058, 16098621, 16467844) are not referenced.

Thank you very much for pointing this out. We apologize for these omissions, and we have discussed these studies in the revised manuscript.

2) An important limitation of the approach appears to be the lack of quantification of mRNAs with 0 or >8 ribosomes. Authors need to address this:

a) The total mRNA abundance needs to be accounted for. With the current approach, the assigned MRL of mRNAs may be substantially skewed due to the omission of “0”s. Authors argue in the discussion that normalizing by total mRNA introduces variability, which is probably true. However, ignoring this issue could substantially affect conclusions. Some mRNAs may have poor general translation initiation efficiency, but effective re-initiation due to mRNA circularization. This results in two mRNA pools: One associated with ribosomes that could be efficiently translated, and one sequestered from ribosomes that is not translated at all. The relative abundance in these pools could be affected by mTOR and other translation regulators. In fact, others have previously employed methods to quantify the fraction of mRNAs associated with ribosomes, in addition to the average MRL of ribosome-bearing mRNAs (e.g. PMID 16467844).

The referee is correct in pointing out that mRNAs not included in the 1 – 8+ ribosome fractions could, in theory, affect the results. We elected not to include the ‘0’ ribosome fraction due to the technical challenge of collecting this fraction accurately. This is because the flow of samples from the fractionator can be unstable for the first fraction. Thus, our decision on this point was determined by the potential to introduce inaccuracy. In order to demonstrate that the omission of ‘0’ ribosome fraction does not affect the conclusions, we have performed additional experiments to provide a direct comparison of data with and without the inclusion of the ‘0’ ribosome fraction with the careful collection and analysis. The data show that the mRNA content of the ‘0’ ribosome fraction was similar to that of ‘1’ ribosome fraction (Supplementary Notes Fig. 2a) and that the values of mean ribosome load were very similar when calculated with or without including ‘0’ ribosome fraction (Supplementary Notes Fig. 2b). In addition, consideration of the ‘0’ ribosome fraction has a little effect on the calculation of the changes in translation upon Torin 1 treatment (Supplementary Notes Fig. 2c). These results demonstrate that omission of ‘0’ ribosome fraction had a negligible effect on the analysis of translation under the conditions of our experiments. We have included these analyses in Supplementary Notes 1 to the revised manuscript.

Supplementary Notes Fig. 2

(a) Principal component analysis of HP5 data by polysome fraction. The data is identical to Fig. 1b, but the analysis includes mRNAs from the '0' ribosome fractions. **(b)** Scatter plot comparing the mean ribosome load (MRL) for each gene in RCC4 VHL cells calculated with and without inclusion of mRNAs in the '0' ribosome fraction. **(c)** Scatter plot comparing the changes in translational efficiency, expressed as log2 fold change in MRL for each gene with Torin 1, calculated with and without inclusion of mRNAs in the '0' ribosome fraction. Note in **b**, the sharp demarcation above the line of identity is created because inclusion of any value for the '0' ribosome fraction cannot increase MRL; **c** demonstrates that any changes in the calculated value of MRL have no discernible effect on the analysis of changes in MRL in response to treatment of cells with Torin 1.

b) As figure 1a indicates, resolution of the polysome peaks are generally lost after at least 8. Is material lower in the gradient discarded or pooled with the 8r fraction? Do most individual mRNAs have a ~normal distribution across measurable values (i.e. 1-8) or are there mRNAs that cluster heavily towards 8+, indicating that their actual ribosome load is higher?

Material lower in the gradient was pooled with the 8-ribosome fraction, and thus these mRNAs were considered in the estimate of mean ribosome load. We have clarified this point in the revised manuscript.

The referee asked whether the majority of mRNAs were distributed normally within the measurable range. The PCA analysis of mRNAs in each fraction (Fig. 1b) showed that mRNA contents in neighbouring fractions were more similar than other fractions, indicating that the mRNAs were in general broadly distributed across multiple neighbouring fractions. To support this, we analysed how the mRNAs distributed across 8 polysome fractions (Fig. A, below). The analysis further demonstrated that few mRNAs were heavily clustered in 8+ fraction. We believe the PCA analysis substantiates our claim on this point. On grounds of simplicity, we would prefer not to include the heatmap below, which is provided for review, unless the editor or referees felt strongly that it should be included.

Fig. A for referees | Distribution of mRNAs across polysome fractions

Heatmap showing the distribution of mRNAs across polysome fractions. Each row represents a single gene, and the colour indicates the amount of mRNAs in the indicated fraction related to the total amount of that mRNA.

3) It is a weakness that none of the major conclusions are validated by probing translation efficiency of individual transcripts via independent methods such as radiolabelling and immune precipitation normalized by mRNA abundance. For example, demonstrating translational regulation of previously unknown targets in the 'transcription', 'ubiquitination', 'glycolysis' and 'angiogenesis' groups would be of great value and provide confidence in the approach and analysis.

We agree that comparison with independent methods for probing translational efficiency could strengthen the paper. However, probing the translation of multiple individual transcripts across multiple classes of gene by labelling of newly synthesized proteins and immunoprecipitation is not completely straightforward to perform or interpret, particularly as such measurements may be confounded if proteins are secreted, or their stability is also regulated by mTOR (PMID: 26669439 and PMID: 25043031). Given that our analysis was able to identify previously reported mTOR sensitive targets, as defined by earlier studies of both polysome profiling and ribosomal profiling (see response to point 1) we argued that such data should contain evidence of the effect we describe even if the signal was weaker and had not been securely identified.

We therefore set out to validate our finding that "*HIF1A-targeted genes encoding glycolytic enzymes were hypersensitive to mTOR whereas HIF2A-targeted genes encoding proteins involved in angiogenesis and vascular process were resistant to mTOR inhibition.*" by referencing the datasets using an orthogonal method, ribosome profiling (Hsieh *et al.*, PMID: 22367541 and Thoreen *et al.*, PMID: 22552098). We chose to validate this finding because it is one of our significant new findings on the interplay of HIF and mTOR.

We selected the data from the published ribosome profiling studies as this method has been regarded as the gold standard to identify the translational targets of mTOR (Hsieh *et al.*, PMID: 22367541 and Thoreen *et al.*, PMID: 22552098). In addition, ribosome profiling has an advantage over metabolic labelling of newly synthesized proteins as (1) (like polysome profiling) it directly measures translational efficiency of mRNAs at a specific time point, and (2) its measurement is not affected by the changes in the rate of protein degradation or secretion (that affects many angiogenesis or vascular process related genes).

We tested whether the same trend in our findings could be found in the ribosome profiling data. Consistent with this, the analysis of ribosome profiling data revealed that the relative translational efficiency of mRNAs encoding glycolytic genes was more strongly downregulated upon mTOR inhibitor treatment than those for genes encoding angiogenesis or vascular process related proteins (Supplementary Fig. 10d). HP5 data showed a greater difference in changes in translation for these two different groups. This is likely to be due to somewhat increased power to measure changes in translation, as demonstrated above. This analysis has been included in the revised manuscript as Supplementary Fig. 10d.

d

Supplementary Fig. 10d

Boxplots comparing changes in translational efficiency for the indicated classes of gene in response to treatment with an mTOR inhibitor. Data are expressed as log₂ fold change in mean ribosome load (MRL) for the HP5 data or in translational efficiency for the ribosome profiling data. The horizontal lines show the median changes in MRL or translational efficiency of all expressed genes. The colours indicate whether the changes are above or below the median value of all genes (blue and red respectively). PC3: human prostate cancer cells; MEF: mouse embryonic fibroblasts; the number of genes in each class is indicated in parenthesis.

4) All effects of VHL presence/absence are ascribed to HIF. This is clearly inappropriate as VHL has other targets. Only once do investigators experimentally attribute an effect to HIF by knocking down HIF1b. This issue needs to be addressed in the way data are interpreted throughout, and the limitation acknowledged.

The referee is correct that we sometimes ascribed the effects on gene expression of intervention on VHL as being due to HIF without directly testing in all cases. Although the regulation of HIF is the best understood (and best reproduced) action of VHL, the referee is also correct that other targets have been reported. As suggested, we have acknowledged the limitation in the manuscript.

Minor:

1) Similarly, there is no data to corroborate that effects of Torin are attributable only to mTOR. This is less of an issue, but at least some references to the specificity of the inhibitor and potential off-target effects needs to be included.

Thank you. Although, as the referee suggests, Torin 1 is a relatively specific inhibitor we take the point that off-target effects could arise. In the revised manuscript, we have cited a study that reported the discovery of Torin 1 and examined the specificity (PMID: 19150980).

2) Analysis performed in Fig4a and resulting conclusions are unclear and should be better explained. What is the purpose of 'omitting' certain parameters and what does 'simulated MRL' mean?

The purpose of the analyses in Figure 4a is to define which of two potential modes of regulation contribute most to VHL-dependent changes in translational efficiency of mRNAs associated with genes whose transcriptional start-site (TSS) usage is altered by VHL-status.

That is:

- (i) translational efficiency could be generally regulated by VHL across this class of gene (i.e. across all the transcripts irrespective of their TSS)
- (ii) translational efficiency could be changed only as a consequence of the altered TSS.

The plots show the correlations between measured translational efficiency of all mRNAs derived from these genes and that calculated from transcript resolved data missing out each of these terms (i.e. either on the assumption that there are no VHL-dependent alterations of TSS; or on the assumption that there are no VHL-dependent changes in translation within a TSS-defined transcript).

The analysis is important in demonstrating that VHL-dependent altered translational efficiency of mRNAs derived from these genes is largely due to the altered TSS. We have done our best to explain this with greater clarity in the revised manuscript.

Referee #2:

Remarks to the Author: The manuscript by Sugimoto and Ratcliffe entitled "Isoform resolved measurements of absolute translational efficiency define interplay of HIF and mTOR dysregulation in kidney cancer" describes a comprehensive effort to identify the relationship of transcriptional reprogramming by HIF to translational reprogramming by mTOR in the setting of kidney cancer. The authors describe a new workflow for measurement of absolute translational efficiency of mRNAs resolved by their transcription start sites. They determined a remarkable translational reprogramming by mTOR, especially with metabolic enzymes and pathways. In contrast global effects of HIF on translation are limited and not observed at all by HIF-2 α . Instead, these are mediated by HIF-1 α in a subset of genes. A specific class of HIF-1 α transcriptional target genes exhibit sensitivity to mTOR effects indicating that inhibitors of HIF-2 α and mTOR could be efficacious in the treatment of kidney cancer. Overall, the data is extremely convincing and the manuscript well written. I strongly recommend acceptance upon the resolution of the following concerns.

1. The western blot pattern describing pVHL protein abundance in Figure S2 reveals a complex pattern of bands. Can the authors explain why the species are so numerous? Is this due to post translational modifications, or distinct pVHL isoforms, etc.?

Thank you. Multiple species of pVHL have been observed consistently in many previous studies. In part, they arise from internal initiation; an internal start codon in VHL produces an 18 kDa isoform in addition to the full length 24 kDa isoform (PMID: 10102622). However, we agree there are further species. These additional species of VHL have also been observed in previous studies, and possibly represent post-translational modifications, but to our knowledge their precise origin has not been

established (PMID: 9671762). We have included a brief explanation on this in the figure legend of Supplementary Fig. 2.

2. Based on comparing results in RCC4 cells (expressing both HIF-1 α and HIF-2 α) to 786-O cells (expressing HIF-2 α only), the authors conclude that HIF-2 α has no real effect on the translational efficiency of kidney cancer mRNAs. However, to make this claim the authors really need to remove HIF-1 α from RCC4 cells genetically or add HIF-1 α to 786-O cells so that isogenic cell lines are compared.

The referee's comment concerns our conclusion that HIF2A has no major ('real') effect on translation efficiency (of mRNAs expressed in kidney cancer cells) which is in fact based on the absence of changes in translational efficiency when HIF2A is suppressed by reintroduction of VHL in 786-O cells. Since the previous studies (Uniake *et al.* PMID: 22678294 and Ho *et al.* PMID: 26854219) encompassed data in (the same) 786-O cells, our choice of 786-O cells is valid, and we clearly did not reproduce the reported translational effects of HIF2A (see the responses to referee 1).

Even so, we agree that examination of another cell with an intervention on HIF2A alone would reinforce this point. The referee is correct that in theory we could engineer RCC4 cells to inactivate HIF1A (so that VHL would regulate HIF2A alone). However, for the proposed work, this would need to be a two-stage procedure i.e. RCC4 cells would need to be engineered to remove HIF1A and then clones of these cells would need to be stably transfected with a plasmid expressing VHL or empty vector and re-cloned to permit the desired comparison. Apart from the time to complete this, perform the polysome profiling, and obtain sequencing data (we estimate 9-12 months), there are uncertainties in whether manipulation of HIF1A would be tolerated in this setting. We therefore think that this would fall outside the time-scale for timely and efficient revision, particularly as we in fact have orthogonal unpublished data on this point in an immortalized renal cell line HKC8.

The work was performed using ribosome profiling. Comparison of translational efficiency in CRISPR/Cas9 engineered cells in which *HIF2A* had or had not been inactivated manifest closely similar translational efficiencies under hypoxia, including for HIF2A/EIF4E2 target genes (Uniake *et al.* PMID: 22678294 and Ho *et al.* PMID: 26854219) whose translation was reported to be upregulated by HIF2A (Fig. B). We are providing this data to the referee to provide reassurance on the point raised. However, unless the reviewer or editor felt strongly we would prefer not to include it in the manuscript as it would mean introducing a new set of reagents and technologies that could potentially be distracting from the main messages.

Fig. B for referees | Changes in translational efficiency, as measured by ribosomal profiling.

The graph shows the log₂ fold change in translational efficiency upon hypoxia in wild type HKC8 cells and in HKC8 cells engineered to inactivate *HIF2A*. There is no evidence of a HIF2A-dependent increase in translational efficiency either across all genes or for reported HIF2A/EIF4E2 targets (PMID: 22678294 and PMID: 26854219) as labelled. Note that in this analysis genes whose translation is positively regulated by HIF2A, in hypoxia should generate values beyond 0 on the x-axis, and near 0 on the y-axis, which is not the case.

3. Minor points: on page 9 “particular striking” should read “particularly striking”.

Thank you. We have corrected the mistake.

4. The authors call into question previous findings by Uniacke et al. (Nature 2012) and provide some explanations for the discrepancy in their findings. This should be elaborated on to improve readability of the manuscript.

We have elaborated and made our position overt in the revised manuscript that our data are in conflict with the findings of Uniacke *et al.* and Ho *et al.* (PMID: 22678294 and PMID: 26854219). We also note that although no other contradictory findings have so far been published, no confirmatory findings have been published by any other group either. This is surprising given the high profile of that work. However, we have no explanation other than the one we give – that in the absence of any comparison to a known intervention on translation, the effects may have been exaggerated. As our manuscript contains many other important points, we did not wish this contradiction to dominate too much.

5. Finally, the first paper to demonstrate hypoxic influences on mTOR was by Arsham et al. (JBC 2003). This paper convincingly demonstrated that the influence of hypoxia on mTOR is rapid and independent of the HIF-1 α /REDD 1 pathway described by Brugarolas et al in Genes and Development. The paper should be cited.

Thank you very much for pointing out this missing reference. We have cited this paper in our revised manuscript.

Referee #3:

Remarks to the Author: The manuscript “Isoform resolved measurements of absolute translational efficiency define interplay of HIF and mTOR dysregulation in kidney cancer” is devoted to investigation of MTOR and HIF dependent changes in gene expression for transcripts with alternative transcription start sites in kidney cancer cells. The authors developed a clever modification of polysome profiling in order to define translation efficiencies of mRNA isoforms that differ in their 5’ends. Using this approach, the authors demonstrated that in contrast to MTOR, HIF signaling does not directly affect translation of mRNAs as it was reported before. Instead, HIF activation results in alternative transcription start site usage for a number of genes, which results in alterations in 5’leaders of mRNAs and in differential translation. Finally, the authors found that HIF1A and HIF2A induced mRNAs have differential sensitivity to mTOR inhibition. To my opinion, this is definitely an interesting and important study; however, I have several comments regarding the method and clarity of presentation and discussion of results. Also, I have a number of suggestions related to additional analysis of already available data which may result in new interesting findings.

1. Regarding the methodology - first of all, while the method which the authors called HP5 is definitely useful, I have a feeling that the authors seriously overestimate their approach, as it certainly can’t be applied for estimation of “absolute translation efficiency”. In addition, this approach is not really new – there are several important manuscripts related to isoform specific translation which currently are not properly discussed. Most importantly, this method does not discriminate which open reading frame is translated in particular mRNA. A simple example – for mRNA with one uORF and median ribosome load 3 it is equally possible that all 3 ribosomes occupy either uORF or CDS. Given that nearly half of mammalian mRNAs possess at least one uORF, one can’t reliably predict translation efficiency from mRNA position in the gradient. For instance, well known ATF4 mRNA possess several regulatory uORFs, and according to riboseq data, under normal conditions most ribosomes translate uORFs but not ATF4 CDS. In addition, the authors ignored alternative splicing (especially important - in 5’leaders) and alternative 3’UTRs utilization - in this case mRNAs with the same 5’end may have different translation efficiencies. Such cases can’t be detected as the method relies only on sequencing of mRNA 5’ends. Finally, ribosome occupancy does not always correlate with translation efficiency, given that local elongation rate may vary for different mRNAs. A good example can be found in a recent manuscript (PMID: 32589965). It was demonstrated that depletion of one of eIF3 subunits resulted in increased ribosome occupancy on certain mRNAs, but these mRNAs were not activated but actually repressed at translation level. To conclude, the method is useful for estimation of translation efficiency of mRNA isoforms, but can’t provide absolute translation efficiency. The authors must discuss the limitations of their approach and, where necessary, implement alternative methods to validate HP5 results (e.g. – RT-PCR of specific mRNA isoforms with primers specific to unique parts of alternative 5’leader in polysome gradient fractions).

We thank the reviewer for their positive comments on the overall value of the manuscript, but note the critique, which is useful. We have made revisions that address this, both by discussing limitations and by adding new data to address the referee’s final point.

By the isolation, high depth sequencing, and normalization of sequence reads for each individual polysome fraction to an external standard, the method does provide greater accuracy and greater ability to compare samples. Importantly, it enables distinction as to whether a change in translational efficiency is ‘absolutely’ present for a specific transcript, as opposed to being only defined relative to some normalization process, which is itself affected by the intervention (e.g. a change in total protein synthesis level in the cell). This is, of course, particularly important when comparing responses to one intervention with those to another intervention, which is the subject of the manuscript. It is in that sense that we used the term ‘absolute’.

However, in the light of the referee’s critique we can see that our use of the term ‘absolute’ can cause confusion. The referee is correct in stating that we cannot correct for changes in translational elongation or determine where ribosomes are situated on the transcript. So, the measurement of translational efficiency is not ‘absolute’ in the sense of always reflecting the (quantitative) rate of translation of every mRNA. We have clarified this in revision using the term ‘externally normalized’ where appropriate to avoid ambiguity, and discussed the limitations raised by the referee. As suggested, we have also discussed the relationship of our work to other work analysing isoform specific translation.

Furthermore, as suggested in the referee’s final paragraph, we have also corroborated resolution of mRNA isoforms with different 5’ leader sequences in different parts of the polysome gradient using RT-qPCR. We have provided this data in the revised manuscript in Supplementary Notes Fig. 1 (reproduced below).

Supplementary Notes Fig. 1

Comparison of data generated by HP5 (upper panels) with that generated by RT-qPCR (lower panels) on the same polysome fractions. The figure shows the proportion of mRNAs in relation to the total mRNAs across all eight polysome fractions for selected genes, as indicated above the panels. The lines indicate the mean values while the shaded area shows the standard deviation of 3 biological replicates for the HP5 data, or 2 technical replicates for the RT-qPCR data, respectively. The analyses were performed on RCC4 VHL cells. The examples have been selected to compare data on genes where HP5 defined different mRNA isoforms. These are illustrated in the schematics below the line plots and termed ‘upstream TSS’ and ‘downstream TSS’ isoforms according to the relative positions of the TSS. In some cases, the resolution provided by RT-qPCR was less than HP5, in which case appropriate

integration of the HP5 data was performed to permit exact quantitative comparisons between HP5 and RT-qPCR.

Overall, closely similar patterns were observed using RT-PCR and HP5.

**Different upstream or downstream mRNA isoforms not resolved by RT-qPCR and are grouped for comparison with HP5.*

***The downstream mRNA isoform not separately resolved by RT-qPCR therefore resolved species comprise upstream and upstream + downstream mRNAs.*

2. There is one concern related to the experimental approach: the authors used oligo-dT primer for reverse transcription - this means that for long mRNAs there are less chances that reverse transcriptase will reach its 5' end. Moreover, in case of limited mRNA degradation, which is impossible to completely avoid during sample preparation, there are more chances for long mRNAs to be cleaved at internal positions, which will prevent detection of 5' end. How do the authors take this into account? Can it be that this potential artefact can account for unusual dependence of MRL on CDS length (Fig 1 D)? Perhaps utilization of a random reverse primer instead of oligo-dT for reverse transcription may be a good alternative.

Whilst we accept that it is not possible to entirely avoid degradation, we believe that our protocol extracted largely intact RNAs as evidenced by Bioanalyzer analysis (Fig. C). We have referred to this results in the revised Methods section.

Fig. C for referees

Boxplot showing the distribution of the RNA integrity (RIN) number, from Bioanalyzer analyses, for the RNA extracted after polysome fractionation according to the HP5 protocol. The data were for RCC-4 VHL cells. A RIN number above 8 (highlighted in the figure) is typically considered to represent high-quality RNA (PMID: 24060121).

Notwithstanding that, we do not think that any such degradation, or indeed inefficient reverse transcription, of long mRNAs could explain the observed dependence of mean ribosome load on CDS

length. Such a bias would be introduced only if the library from different fractions were differently affected by these processes. There is no obvious reason why this should be, since we processed samples from all the fractions in parallel, and these biases should be equally introduced to all the fractions.

Use of oligo dT priming enabled us to produce mRNA-Seq libraries without an mRNA purification step from total RNA, which is typically required when random primers are used. Thus, it enables an increase in efficiency and sensitivity that is important to the protocol.

3. As I mentioned before, the method is not entirely new. There are several studies about mRNA isoform specific translation, which are very similar in methodology (e.g. Wang et al. Pervasive isoform-specific translational regulation via alternative transcription start sites in mammals PMID: 27430939; Floor et al “Tunable protein synthesis by transcript isoforms in human cells” PMID: 26735365; Tamarkin-Ben-Harush et al. “Cap-proximal nucleotides via differential eIF4E binding and alternative promoter usage mediate translational response to energy stress”, PMID: 28177284; Gandin et al., “nanoCAGE reveals 5' UTR features that define specific modes of translation of functionally related MTOR-sensitive mRNAs”, PMID: 26984228) – and these papers are not properly discussed and cited here. The last one is especially relevant, as it is devoted to the same issue - interplay between TSS usage and response to mTOR inhibition at translational level.

Thank you. We have provided a more detailed description of these related methodologies in the revised manuscript as suggested. Though these studies are important, we consider present work to be a substantial advance, both in providing increased resolution and in enabling multiple comparisons across different conditions.

4. Then studying response to mTOR inhibition, the authors found a lot of hypersensitive mRNAs. It would be interesting to compare these results with results of other studies performed with riboseq and polysome profiling. What mRNAs are commonly downregulated at translation level, except 5'TOP mRNAs, and what is new? It is known that 5'TOP-containing mRNAs are not the only class of mRNAs that are “hypersensitive” to mTOR at translation level (see Gandin et al. – they really carefully examined the impact of 5'leader variants on MTOR mediated translation repression using the nanoCAGE technique).

The referee's first point here is very similar to that of referee 1 (point 1) and as suggested, we have analysed the ability of HP5 to identify mTOR hypersensitive genes reported by four very highly cited previous studies: Hsieh *et al.* (PMID: 22367541; deploying ribosome profiling), Thoreen *et al.* (PMID: 22552098; deploying ribosome profiling), Larsson *et al.* (PMID: 22611195; deploying polysome profiling), and Morita *et al.* (PMID: 24206664; a subset of targets identified by Larsson *et al.*). Note that Gandin *et al.* (PMID: 26984228) reanalysed the data from Larsson *et al.* (PMID: 22611195).

These analyses revealed that mTOR hypersensitive genes identified by previous studies are not entirely overlapping, overlap being small between studies using ribosome profiling and those using polysome profiling (new Supplementary Fig. 4a). The analyses further demonstrated that whilst HP5 data faithfully identified the strong downregulation of mTOR hypersensitive genes identified by all four studies upon an mTOR inhibitor treatment, ribosome profiling was less powerful at identifying translational downregulation of mTOR hypersensitive genes previously identified by polysome profiling (new Supplementary Fig. 4b). For details, please see also our response to referee 1.

Thus, we consider that HP5 resolves changes in translation with greater sensitivity and accuracy than previous studies. This underpins the comprehensive analyses of the interplay between the VHL-dependent transcriptional response and mTOR inhibition that we describe. Apart from this the analyses also provided new insights into control by mTOR inhibition itself, including the sensitivity of a wide range of functional classes of gene, the importance of the precise positioning of the TOP motif and interactions between the mTOR sensitivity of mRNAs and the number of uORFs and CDS length.

5. It may be interesting to show how translation of mRNAs encoding proteins directly implicated in HIF signaling pathway (VHL, HIF subunits, HIF-prolyl hydroxylases etc) respond to mTOR inhibition itself?

Thank you. As suggested, we have performed an additional analysis to investigate the effect of mTOR inhibition on genes involved in the HIF signalling pathway. The analysis revealed that translation of two of the oxygen sensitive 2-oxoglutarate-dependent dioxygenases, FIH1 and PHD3 were more strongly downregulated than other genes in this group indicating that mTOR inhibition alters how the cell responds to hypoxia. We have provided this data as Supplementary Fig. 10a in the revised manuscript.

Supplementary Fig. 10a

The figure is identical to Fig. 5b, but genes involved in the HIF signalling pathway have been labelled.

6. Are there any mRNAs with alternative TSS that are differentially regulated by mTOR? If there are such cases – it will be very interesting to discuss it.

Thank you. Our data do indeed permit the analysis of the difference in mTOR sensitivity between alternate TSS mRNA isoforms of same gene. Consistent with the overall findings in the manuscript, these analyses show that alternate TSS isoforms that are more sensitive to mTOR inhibition do have longer TOP motifs and a lesser number of uORFs when compared to other isoforms of same gene that are less mTOR sensitive (Supplementary Fig. 5c).

We have provided this data in the revised manuscript in Supplementary Fig. 5c.

Supplementary Fig. 5c

Boxplots illustrating associations between mRNA features (length of TOP motif, left panel; number of uORFs, right panel) and sensitivity to mTOR inhibition, for alternate TSS mRNA isoforms of same gene. The mRNA isoforms are classified as sensitive or resistant based on their sensitivity to mTOR inhibition (the isoform with a larger or smaller mean ribosome load, MRL, log₂ fold change with Torin 1, respectively). When more than two isoforms were expressed from the same gene, the isoforms with the largest and smallest MRL log₂ fold change were selected for the analysis. The comparison was performed by the groups binned by their difference in MRL log₂ fold change of the two isoforms (x-axis). Distributions of the length of TOP motifs or the number of uORFs were compared using the Wilcoxon signed rank test. *: $p < 0.05$, **: $p < 0.005$. P values were adjusted for multiple comparisons using Holm's method. Data are for RCC4 VHL cells.

7. The finding that HIF2A does not regulate translation is very important. In addition to western blots (supplementary fig 5b) I suggest to show the MRL values for previously reported "HIF2A/eIF4E2 axis - regulated mRNAs" in cells with and without HIF2A and eIF4E2– this would strengthen conclusions and convince readers that HIF2A is not translation factor.

Thank you and yes, we agree. Our analyses did not reveal any major changes in mean ribosome load that were dependent on HIF2A or EIF4E2 (Fig. 3b). As outlined in our response to referee 1, we have included data that highlights all genes reported as 'HIF2A/eIF4E axis regulated mRNAs' (i.e. whose translational efficiency was proposed to be regulated by this axis by Uniacke *et al.* PMID: 22678294 and Ho *et al.* PMID: 26854219) in Supplementary Fig. 6a and c. This demonstrates that HIF2A and eIF4E2 do not regulate the translation of these genes under conditions of our experiments (which as far as we can ascertain from the publications were the same as those used in the studies originally reported the role of HIF2A in translational regulation).

8. The part "HIF promotes alternate TSS usage to regulate translation" is written in such a way that it is really hard to follow and understand. In my opinion, it is necessary to rewrite this part in order to

make it clear what exactly was found. HIF alters TSS usage, and a number of mRNAs with alternative TSSs are differentially translated – is it correct? What is this number?

Yes, the referee's interpretation is correct. We apologize if this was difficult to understand. We found that the alternate TSS isoforms of 75 genes (64% of 117 genes; this analysis was restricted to genes with sufficient mRNA isoform expression for the MRL calculation, amongst 149 genes that manifested VHL-dependent alternative TSS usage) were differentially translated from other isoforms of the same gene. Note the data for each gene are displayed in Supplementary Fig. 8 (Supplementary Fig. 7 of the original manuscript), and we have added the data on the numbers in the revised manuscript.

We see that referee 1 had similar difficulties with this description and we have rewritten the relevant text to improve clarity.

MXI1 is a really good example, and I suggest adding more examples (in the main text or in the supplementary). It is interesting to see whether it is common that differentially translated mRNA variants differ in uORFs? Are there any cases where mRNA variant with longer 5' leader is translated better than that with shorter 5' leader? There may be particularly interesting explanation that long 5'UTR variants can bear some cis-acting element which stimulates translation.

We have first addressed the question of whether it is common that differentially translated (VHL-dependent) transcripts differ in their uORFs. Although our data can readily resolve the difference in translation of VHL-dependent alternate TSS isoforms from other isoforms of same gene, identification of sequence features that can explain the difference is more challenging, in part because uncertainties in predicting the precise anatomy of the 5' UTR (e.g. as arising from splicing – as correctly raised by this referee in point one). This is particularly a problem for this dataset where HIF activation could potentially increase uncertainties in annotation. However, we considered that using our algorithm, we were able to predict that sequence of the entire 5' UTR of both the alternate TSS and other expressed isoforms, for 42 genes. This is out of 75 genes whose VHL-dependent alternate TSS isoforms were translated differently from other isoforms of same gene.

Using this data, we have examined the relationship between differential translation and a different number of uORFs. This analysis reveals that there is indeed an inverse relationship across this group of 42 genes. We also provide data for two examples: UDP-N-Acetylglucosamine pyrophosphorylase 1 (*UAP1*) and deoxyribose-phosphate aldolase (*DERA*) where VHL-regulated differentially translated mRNAs were associated with different numbers of uORFs. This data (together with the above caveats) is illustrated in Supplementary Fig. 9 in the revised manuscript.

Supplementary Fig. 9a

Bubble chart showing the proportion of genes by the difference in the number of uORF between alternate TSS and other isoforms (x-axis). The proportion was calculated in relation to the total number of genes by the difference in translational efficiency, as expressed as mean ribosome load (MRL), between alternate TSS and other isoforms (y-axis). $n = 20$ and 22 for genes whose alternate TSS isoform had a higher or lower MRL than the other isoforms respectively. The analysis was performed for 42 genes for which 5' UTR sequence of alternate TSS and majority of other isoforms could be predicted, out of 75 genes whose VHL-dependent alternate TSS isoforms were translated differently to other isoforms of same gene.

Supplementary Fig. 9b

Similar to **Fig. 4c-e** in the main manuscript but giving data for *UAP1* and *DERA*. The number of uORFs in their 5' UTR is indicated in the schematic. Note that *VHL* inactivation induced the TSS3 isoform of *UAP1* and *DERA* with less or more uORFs than other isoforms. In both cases, isoforms with less uORFs were better translated.

The referee also asked whether, in addition to *MXI1*, there were *VHL*-dependent mRNA isoforms in which a longer 5'UTR was better translated and whether this might imply a cis-acting element that positively regulates translation. We could not find a specific cis-element, but this may reflect the limited sample size, particularly after excluding genes where the presence of uORF could explain the translational differences.

9. Given the potentially high importance of HIF-dependent *MXI1* transcription regulation, it is important to mention that HIF-mediated switch of TSS usage not only alters 5'leaders but also CDSs – three different transcripts encode different *MXI1* proteoforms. Is anything known about functional interplay of these proteoforms? In general, it may be interesting to highlight those genes where alternative TSS result in changes of CDS, similar to the *MXI1* case.

The referee raises an important point. We have clarified that that the HIF-dependent switch in TSS usage alters the CDS of *MXI1* in the revised manuscript. This pattern is conserved, in that the mouse

Mxi1 gene has similar alternative TSS isoforms. Furthermore, the isoform equivalent to that induced by *VHL* inactivation in humans has an additional domain that binds to the Sin3/histone deacetylase and manifests stronger transcriptional repressor activity (PMID: 15467743), strongly suggesting a downstream action of the switch in TSS usage. We have included a brief comment on this in the revised manuscript.

As suggested we have also analysed the prevalence of altered CDS consequent on altered TSS usage. This revealed that 55% of the genes (71 out of 129 genes; this analysis was restricted for genes whose CDS of the mRNA isoforms could be predicted, amongst 149 genes) that manifest a VHL dependent alternate TSS had a different predicted CDS. We have added this data in the revised Supplementary Fig. 7 (original manuscript, now Supplementary Fig. 8) and Supplementary Table 2.

(Revised) Supplementary Fig. 7 | VHL dependent alternate TSS usage generates mRNAs with an altered translational efficiency

The plot shows the differences in translational efficiency (expressed as mean ribosome load, MRL) between mRNA isoforms that are generated from VHL-dependent alternative TSSs and all other isoforms transcribed from the same gene. Data are for RCC4 cells. Genes are sorted by log2 fold difference in MRL; significant differences in polysome distribution (FDR < 0.1) are indicated by black colouring. The magnitude of changes in alternative TSS usage is shown by the size of point. Genes whose alternate TSS isoform contains a different predicted CDS are indicated with a check mark; NA, CDS could not be predicted. Genes with too little alternate TSS isoform expression for MRL calculation were excluded from the analysis (see Methods).

10. Are there any specific features (e.g. length, secondary structures, uORFs) or sequence motifs in 5' leaders of mRNAs from "glycolysis" and "angiogenesis or vascular process" subsets which may help to explain differential sensitivity of translation to mTOR inhibition?

We have performed comparative analyses of these classes of genes for features that we found to be associated with greater or lesser sensitivity to mTOR in the overall dataset. These analyses revealed that a higher proportion of mRNAs for HIF-target glycolytic genes, which were hypersensitive to mTOR, possessed no uORF and CDS with a length in the region of 1,000 nts, as compared to those for HIF-target angiogenesis or vascular process related genes (Supplementary Fig. 10e). In contrast, we did not find a difference in the length of TOP motif between these two groups. We have provided this data as Supplementary Fig. 10e in the revised manuscript.

Supplementary Fig. 10e

Proportion of genes with the indicated number of uORFs (top panel), CDS length (middle panel), and the length of TOP motif (bottom panel), amongst the specified functional class of HIF-target genes. For the plot of the CDS length, the two most mTOR sensitive groups are indicated by a red box. These analyses were performed for genes for which the 5' UTR sequence could be predicted for the majority of expressed mRNAs (n = 11 and 27 for glycolysis and angiogenesis or vascular process genes respectively).

11. I suggest generating a supplementary table which will include all HIF dependent transcripts and effects of MTOR inhibition – currently it contains only those that are related to glycolysis and angiogenesis.

As suggested by the referee, we have provided data on the effects of mTOR inhibition on all HIF dependent transcripts in Supplementary Table 3.

Decision Letter, first revision:

Our ref: NSMB-A45126A-Z

4th Apr 2022

Dear Dr. Ratcliffe,

Thank you for submitting your revised manuscript "Isoform-resolved mRNA profiling of ribosome load defines interplay of HIF and mTOR dysregulation in kidney cancer" (NSMB-A45126A-Z). It has now been seen by the original referees and their comments are below. The reviewers find that the paper has improved in revision, and therefore we'll be happy in principle to publish it in Nature Structural & Molecular Biology, pending minor revisions to satisfy the referees' final requests and to comply with our editorial and formatting guidelines.

To facilitate our work at this stage, we would appreciate if you could send us the main text as a word file. Please make sure to copy the NSMB account (cc'ed above).

Sincerely,

Carolina

Carolina Perdigoto, PhD
Chief Editor
Nature Structural & Molecular Biology
orcid.org/0000-0002-5783-7106

Reviewer #1 (Remarks to the Author):

The authors of this manuscript have provided thoughtful and detailed responses to reviewers' concerns and revised the manuscript accordingly. This revised version contains several new data panels of great value, a clearer description of approaches, and more comprehensive referencing of the existing literature.

Authors have chosen not to pursue further validation of findings regarding gene sets (or MX11) by

independent methods as suggested by this reviewer. I am still of the opinion that this would strengthen the paper. I recommend that in lieu of such validation, authors acknowledge that ribosome loading is not always directly proportional to translation rate, and that this should be considered in the interpretations of the work.

In its present form, this manuscript represents a highly valuable addition to literature, both in terms of the new method presented and in the scientific conclusions derived from its application.

Reviewer #2 (Remarks to the Author):

The authors have done an excellent job revising the paper and it is now appropriate for acceptance by the journal.

Reviewer #3 (Remarks to the Author):

In my opinion, the revised manuscript is substantially improved and I will be happy to recommend it to publication. I still have some minor suggestions and recommendations and I believe that addressing these issues will improve the manuscript.

1. New Supplementary figure 4 shows the comparison of mTOR hypersensitive genes identified in previous studies. I understand that comparison of ribosome profiling and polysome profiling data can be challenging, but I was quite surprised to find such a small overlap between both ribosome profiling data and Larsson et al. What about 5'TOP mRNAs which are believed to be hypersensitive to mTOR inhibition at translation level? It is unclear why there is such a small overlap, less than 10 genes? I may recommend the authors to double check this with regard to 5'TOP mRNAs.
2. I found the figure Supplementary Notes Fig. 1 really convincing and straightforward. I highly recommend the authors to place it somewhere among the supplementary figures, or even among the main figures, as it highlights superior performance of their HP5 approach. Of course, the authors may wish to place this figure where they find it suitable.
3. One of the most interesting and provocative findings regarding mTOR mediated translational response is dependence of CDS length on translation. I think that the authors interpretation through closed loop formation may be correct, however, it should be noted that the closed loop is formed between the ends of mRNA, not CDS. Therefore, if the authors are correct, I would expect an even stronger relationship between mRNA length (including both 5'UTR, CDS, and importantly - 3'UTRs) and mTOR inhibition. It is well known that mammalian 3'UTRs are often quite long, and sometimes are much longer than CDS. The author may wish to perform the same analysis of mRNA as it was done for CDS length - I believe that it will either support their hypothesis, or provide some interesting ideas for future investigations.
4. Page 13, line 303 - please correct "protein translation" to mRNA translation or protein synthesis.

Decision Letter, final checks

Our ref: NSMB-A45126A-Z

4th May 2022

Dear Dr. Ratcliffe,

Thank you for your patience as we've prepared the guidelines for final submission of your Nature Structural & Molecular Biology manuscript, "Isoform-resolved mRNA profiling of ribosome load defines interplay of HIF and mTOR dysregulation in kidney cancer" (NSMB-A45126A-Z). Please carefully follow the step-by-step instructions provided in the attached file, and add a response in each row of the table to indicate the changes that you have made. Ensuring that each point is addressed will help to ensure that your revised manuscript can be swiftly handed over to our production team.

In recognition of the time and expertise our reviewers provide to Nature Structural & Molecular Biology's editorial process, we would like to formally acknowledge their contribution to the external peer review of your manuscript entitled "Isoform-resolved mRNA profiling of ribosome load defines interplay of HIF and mTOR dysregulation in kidney cancer". For those reviewers who give their assent, we will be publishing their names alongside the published article.

Nature Structural & Molecular Biology offers a Transparent Peer Review option for new original research manuscripts submitted after December 1st, 2019. As part of this initiative, we encourage our authors to support increased transparency into the peer review process by agreeing to have the reviewer comments, author rebuttal letters, and editorial decision letters published as a Supplementary item. When you submit your final files please clearly state in your cover letter whether or not you would like to participate in this initiative. Please note that failure to state your preference will result in delays in accepting your manuscript for publication.

Cover suggestions

As you prepare your final files we encourage you to consider whether you have any images or illustrations that may be appropriate for use on the cover of Nature Structural & Molecular Biology.

Nature Structural & Molecular Biology has now transitioned to a unified Rights Collection system which will allow our Author Services team to quickly and easily collect the rights and permissions required to publish your work. Approximately 10 days after your paper is formally accepted, you will receive an email in providing you with a link to complete the grant of rights. If your paper is eligible for Open Access, our Author Services team will also be in touch regarding any additional information that may be required to arrange payment for your article.

Please note that *Nature Structural & Molecular Biology* is a Transformative Journal (TJ). Authors may publish their research with us through the traditional subscription access route or make their paper immediately open access through payment of an article-processing charge (APC). Authors will not be required to make a final decision about access to their article until it has been accepted. [Find out more about Transformative Journals](https://www.springernature.com/gp/open-research/transformative-journals)

Please use the following link for uploading these materials:
[Redacted]

Best regards,

Sophia Frank
Editorial Assistant
Nature Structural & Molecular Biology
nsmb@us.nature.com

On behalf of

Carolina Perdigoto, PhD
Chief Editor
Nature Structural & Molecular Biology
orcid.org/0000-0002-5783-7106

Reviewer #1:

Remarks to the Author:

The authors of this manuscript have provided thoughtful and detailed responses to reviewers' concerns and revised the manuscript accordingly. This revised version contains several new data panels of great value, a clearer description of approaches, and more comprehensive referencing of the existing literature.

Authors have chosen not to pursue further validation of findings regarding gene sets (or MXI1) by independent methods as suggested by this reviewer. I am still of the opinion that this would strengthen the paper. I recommend that in lieu of such validation, authors acknowledge that ribosome loading is not always directly proportional to translation rate, and that this should be considered in the interpretations of the work.

In its present form, this manuscript represents a highly valuable addition to literature, both in terms of the new method presented and in the scientific conclusions derived from its application.

Reviewer #2:

Remarks to the Author:

The authors have done an excellent job revising the paper and it is now appropriate for acceptance by the journal.

Reviewer #3:

Remarks to the Author:

In my opinion, the revised manuscript is substantially improved and I will be happy to recommend it to publication. I still have some minor suggestions and recommendations and I believe that addressing these issues will improve the manuscript.

1. New Supplementary figure 4 shows the comparison of mTOR hypersensitive genes identified in previous studies. I understand that comparison of ribosome profiling and polysome profiling data can

be challenging, but I was quite surprised to find such a small overlap between both ribosome profiling data and Larsson et al. What about 5'TOP mRNAs which are believed to be hypersensitive to mTOR inhibition at translation level? It is unclear why there is such a small overlap, less than 10 genes? I may recommend the authors to double check this with regard to 5'TOP mRNAs.

2. I found the figure Supplementary Notes Fig. 1 really convincing and straightforward. I highly recommend the authors to place it somewhere among the supplementary figures, or even among the main figures, as it highlights superior performance of their HP5 approach. Of course, the authors may wish to place this figure where they find it suitable.

3. One of the most interesting and provocative findings regarding mTOR mediated translational response is dependence of CDS length on translation. I think that the authors interpretation through closed loop formation may be correct, however, it should be noted that the closed loop is formed between the ends of mRNA, not CDS. Therefore, if the authors are correct, I would expect an even stronger relationship between mRNA length (including both 5'UTR, CDS, and importantly – 3'UTRs) and mTOR inhibition. It is well known that mammalian 3'UTRs are often quite long, and sometimes are much longer than CDS. The author may wish to perform the same analysis of mRNA as it was done for CDS length - I believe that it will either support their hypothesis, or provide some interesting ideas for future investigations.

4. Page 13, lane 303 – please correct “protein translation” to mRNA translation or protein synthesis.

Final Decision Letter:

Dear Dr. Ratcliffe,

We are now happy to accept your revised paper "Isoform-resolved mRNA profiling of ribosome load defines interplay of HIF and mTOR dysregulation in kidney cancer" for publication as a Article in Nature Structural & Molecular Biology.

To assist our authors in disseminating their research to the broader community, our SharedIt initiative

provides all co-authors with the ability to generate a unique shareable link that will allow anyone (with or without a subscription) to read the published article. Recipients of the link with a subscription will also be able to download and print the PDF.

As soon as your article is published, you can generate your shareable link by entering the DOI of your article here: http://authors.springernature.com/share.

Corresponding authors will also receive an automated email with the shareable link

Note the policy of the journal on data deposition:

<http://www.nature.com/authors/policies/availability.html>.

Your paper will be published online soon after we receive proof corrections and will appear in print in the next available issue. You can find out your date of online publication by contacting the production team shortly after sending your proof corrections. Content is published online weekly on Mondays and Thursdays, and the embargo is set at 16:00 London time (GMT)/11:00 am US Eastern time (EST) on the day of publication. Now is the time to inform your Public Relations or Press Office about your paper, as they might be interested in promoting its publication. This will allow them time to prepare an accurate and satisfactory press release. Include your manuscript tracking number (NSMB-A45126B) and our journal name, which they will need when they contact our press office.

About one week before your paper is published online, we shall be distributing a press release to news organizations worldwide, which may very well include details of your work. We are happy for your institution or funding agency to prepare its own press release, but it must mention the embargo date and Nature Structural & Molecular Biology. If you or your Press Office have any enquiries in the meantime, please contact press@nature.com.

An online order form for reprints of your paper is available at https://www.nature.com/reprints/author-reprints.html. Please let your coauthors and your institutions' public affairs office know that they are also welcome to order reprints by this method.

Please note that *Nature Structural & Molecular Biology* is a Transformative Journal (TJ). Authors may publish their research with us through the traditional subscription access route or make their paper immediately open access through payment of an article-processing charge (APC). Authors will not be required to make a final decision about access to their article until it has been accepted. ](https://www.springernature.com/gp/open-research/transformative-journals) Find out more about Transformative Journals

Authors may need to take specific actions to achieve compliance with funder and institutional open access mandates. If your research is supported by a funder that requires immediate open access (e.g. according to Plan S principles) then you should select the gold OA route, and we will direct you to the compliant route where possible. For authors selecting the subscription publication route, the journal's standard licensing terms will need to be accepted, including self-archiving policies. Those licensing terms will supersede any other terms that the author or any third party may assert apply to any version of the manuscript.

Sincerely,

Carolina Perdigoto, PhD
Chief Editor
Nature Structural & Molecular Biology
orcid.org/0000-0002-5783-7106